# A Generic Family of Graphical Models:
# Diversity, Efficiency, and Heterogeneity

Yufei Huang [* 1]   Changhu Wang [* 2 3]   Junjie Tang [2 3]   Weichi Wu [4]   Ruibin Xi [2 3]

## Abstract

Traditional network inference methods, such as Gaussian Graphical Models, which are built on continuity and homogeneity, face challenges when modeling discrete data and heterogeneous frameworks. Furthermore, under high-dimensionality, the parameter estimation of such models can be hindered by the notorious intractability of high-dimensional integrals. In this paper, we introduce a new and flexible device for graphical models, which accommodates diverse data types, including Gaussian, Poisson log-normal, and latent Gaussian copula models. The new device is driven by a new marginally recoverable parametric family, which can be effectively estimated without evaluating the high-dimensional integration in high-dimensional settings thanks to the marginal recoverability. We further introduce a mixture of marginally recoverable models to capture ubiquitous heterogeneous structures. We show the validity of the desirable properties of the models and the effective estimation methods, and demonstrate their advantages over the state-of-the-art network inference methods via extensive simulation studies and a gene regulatory network analysis of real single-cell RNA sequencing data.

## 1. Introduction

Graphical models (Lauritzen, 1996) are widely used to explore network structures and identify complex interactions between random variables. Recent research has increasingly focused on high-dimensional settings, commonly encountered in applications such as microarray experiments (Amaratunga et al., 2014) and single-cell RNA sequencing (scRNA-seq) studies. However, parameter estimation in such settings poses significant challenges due to the computational intractability of high-dimensional integrals, limiting the efficiency of existing methods. This highlights the critical need for developing efficient estimation frameworks tailored to high-dimensional graphical models.

Due to well-established theoretical properties and high interpretability, Gaussian graphical model (GGM) (Meinshausen & Bühlmann, 2006) has been extensively studied in recent years. Notable methods include $L_1$-penalized log-likelihood maximization (Yuan & Lin, 2007; Banerjee et al., 2008; Friedman et al., 2008), penalized regression (Meinshausen & Bühlmann, 2006; Peng et al., 2009), adaptive thresholding (Cai & Liu, 2011), and D-trace loss (a smooth convex loss function) (Zhang & Zou, 2014), among others. To address the limitations of the Gaussian assumption, semiparametric Gaussian copula models (Liu et al., 2009; Xue & Zou, 2012; Liu et al., 2012) have been developed to handle continuous data, using monotonic univariate transformations. *These methods, based on Gaussian and Gaussian copula models, are usually efficient but are only applicable to continuous data and cannot handle discrete data.*

In many applications, particularly in genomics studies, discrete data are ubiquitous. The observed data are considered to be generated from the discretization of underlying latent variables (Skrondal & Rabe-Hesketh, 2007), which naturally leads to the use of hierarchical models for data modeling. For example, in scRNA-seq data, gene expression levels are count data, often featuring numerous zero values (Islam et al., 2014; Zheng et al., 2017). To effectively model the underlying dependencies in count data, the multivariate Poisson log-normal (PLN) distribution (Aitchison & Ho, 1989) is plausible and popular, since it can capture the conditional dependencies while accommodating overdispersion in the marginal distributions. However, compared to GGM, maximizing the likelihood of the PLN model is more challenging due to the high-dimensional integrals involved, without a known closed-form solution. To address this issue, Choi et al. (2017) used Laplace's method for

---

[*]Equal contribution [1]Center for Data Science, Peking University, Beijing, China [2]School of Mathematical Sciences, Peking University, Beijing, China [3]Center for Statistical Science, Peking University, Beijing, China [4]Department of Statistics and Data Science, Tsinghua University, Beijing, China. Correspondence to: Ruibin Xi <ruibinxi@math.pku.edu.cn>, Weichi Wu <wuweichi@mail.tsinghua.edu.cn>.

*Proceedings of the 42nd International Conference on Machine Learning*, Vancouver, Canada. PMLR 267, 2025. Copyright 2025 by the author(s).

likelihood approximation and applied Alternating Direction Method of Multipliers (ADMM) to compute the penalized maximum likelihood estimator. Chiquet et al.(2019) introduced a variational approximation method to infer the network of the PLN model. *However, these methods rely on approximations of the likelihood function and lack theoretical guarantees.*

In practice, samples often arise from mixed populations exhibiting heterogeneous patterns. Traditional graphical models usually assume that samples come from a single population with a shared network, which *limits their ability to capture data heterogeneity.* To address this, mixture models are widely utilized. For example, in scRNA-seq data, samples comprise single cells from different cell types, each with its own gene regulatory network. One of the arguably most important tools for studying the mixture and count data with different reglatory networks is the mixture Poisson lognormal (MPLN). Silva et al.(2019) employed the MPLN model for clustering count data and used Expectation-Maximization (EM) algorithms with MCMC steps to maximize the computationally intractable log-likelihood. Tang et al.(2024) adopted a variational inference approach for inferring cell-type-specific gene regulatory networks. However, both EM algorithms with MCMC and variational inference are computationally expensive.

To address the aforementioned challenges, this paper makes the following contributions with theoretical guarantees.

1) To accommodate diverse data types, we propose a novel class of distributions, the marginally recoverable parametric family, which unifies existing models and is not restricted to a single data type. This family includes the Gaussian, PLN, and latent Gaussian copula models introduced by Fan et al. (2017) for binary data.

2) To overcome the computational intractability of high-dimensional integration, we develop an efficient parameter estimation framework for this family based on the Maximum Marginal Likelihood Estimator (MMLE). Thanks to marginal recoverability, this method simplifies high-dimensional integrals into multiple low-dimensional integral computations.

3) To capture heterogeneous structures, we extend the framework to a mixture of marginal recoverable models by integrating the EM algorithm to update the MMLE to EM-MMLE.

We establish the consistency of MMLE for covariance matrices and networks under mild conditions. Simulations show our method outperforms existing ones for mixture count and binary data. Furthermore, we apply the EM-MMLE to real scRNA-seq data to infer cell-type-specific gene regulatory networks, showcasing its practical utility. All code is available at https://github.com/XiDsLab/EMMMLE.

Our work balances diversity, efficiency, and heterogeneity, whereas existing methods typically address at most two of these aspects. For instance, Fan et al. (2017) proposed a latent Gaussian copula model for mixed data, combining continuous and binary variables, and introduced a generalized rank-based method. While this approach ensures diversity and efficiency, it does not account for heterogeneity as it cannot handle mixture models.

## 2. Flexible Marginally Recoverable Family

In this section, we propose the marginally recoverable parametric family to address diversity. First, we introduce some notations. For a $p$-dimensional vector $\mathbf{a}$ with the $i$-th entry $a_i$, $\mathbf{a}_{[j,k]} = (a_j, a_k)^\top$ represent the two-dimensional subvector for $1 \le j < k \le p$. For a $p \times p$ symmetric matrix $\mathbf{A}$ with the $(i,j)$-th entry $a_{ij}$, the submatrix $\mathbf{A}_{[j,k]}$ is defined as:

$$\mathbf{A}_{[j,k]} = \left( \begin{array}{cc} a_{jj} & a_{jk} \\ a_{jk} & a_{kk} \end{array} \right) \tag{1}$$

where $1 \le j < k \le p$. This submatrix is constructed by selecting the entries in the $j$-th and $k$-th rows and columns of $\mathbf{A}$. Let $\mathcal{M}_p$ denote the set of $p \times p$ symmetric positive semi-definite matrices.

Our work is inspired by the properties of the Gaussian distribution (Lauritzen, 1996). For a $p$-dimensional random vector $\mathbf{X} \sim N_p(\boldsymbol{\mu}, \boldsymbol{\Sigma})$, any of its two-dimensional marginal distributions $\mathbf{X}_{[j,k]} \sim N_2(\boldsymbol{\mu}_{[j,k]}, \boldsymbol{\Sigma}_{[j,k]})$ for $1 \le j < k \le p$. The parameters of the $p$-dimensional distribution can be fully characterized by the parameters of all two-dimensional marginal distributions. Consequently, the model parameters can be estimated independently through marginal likelihoods. Inspired by this property, we define the marginally recoverable parametric family as follows.

**Definition 2.1** (Marginally recoverable parametric family)**.** Let $\{H_d\}_{d=1}^\infty$ be a sequence of distribution functions, where for each $d \ge 1$, any $p$-dimensional marginal distribution ($1 \le p \le d$) of $H_d$ belongs to the family

$$\mathcal{H}_p = \{H_p(\boldsymbol{\mu}, \boldsymbol{\Sigma}) : \boldsymbol{\mu} \in \mathbb{R}^p, \boldsymbol{\Sigma} \in \mathcal{M}_p\}.$$

For any $p$-dimensional random vector $\mathbf{X} = (X_1, \ldots, X_p)^\top$ with $p \ge 2$ such that $\mathbf{X} \sim H_p(\boldsymbol{\mu}, \boldsymbol{\Sigma})$ where $\boldsymbol{\mu} = (\mu_1, \ldots, \mu_p)^\top$ and $\boldsymbol{\Sigma} = [\sigma_{jk}]_{1 \le j,k \le p}$, we say that $\mathbf{X}$ *or $H_p$ is marginally recoverable* if the following conditions hold:

- For $1 \le j \le p$, $X_j \sim H_1(\mu_j, \sigma_{jj})$.

- For $1 \le j < k \le p$, $\mathbf{X}_{[j,k]} \sim H_2(\boldsymbol{\mu}_{[j,k]}, \boldsymbol{\Sigma}_{[j,k]})$.

The marginally recoverable parametric family includes many of the most common distributions. For instance, we have the following remark.

*Remark* 2.2. Elliptical distributions, including the Gaussian distribution and the multivariate $t$-distribution, are marginally recoverable.

The following proposition elucidates that hierarchical models with marginally recoverable inner layers also satisfy the marginally recoverable condition in Definition 2.1.

**Proposition 2.3.** *Let $Q(\lambda)$ be a distribution function characterized by a single parameter $\lambda$. If $\mathbf{X}$ is marginally recoverable and $\mathbf{Y} \mid \mathbf{X} \sim \prod_{j=1}^{p} Q(X_j)$, then $\mathbf{Y}$ is also marginally recoverable.*

Gaussian copula is a widely used semiparametric model, overcoming the drawback of the Gaussian model's reliance on exact normality (Liu et al., 2009). It is easy to verify that the Gaussian copula model is marginally recoverable.

**Definition 2.4** (Gaussian copula model). Let $\mathbf{X}$ be a random $p$-vector. $\mathbf{X}$ is sampled from the Gaussian copula model, if there exists a monotonic transformation $f$ such that $f(\mathbf{X}) = (f(X_1), \dots, f(X_p))^{\top} \sim N_p(\boldsymbol{\mu}, \boldsymbol{\Sigma})$. Then we denote $\mathbf{X} \sim \mathrm{NPN}(\boldsymbol{\mu}, \boldsymbol{\Sigma}, f)$.

Despite its flexibility, the Gaussian copula model cannot be directly applied to discrete data. For practical applications in network inference, we introduce two hierarchical models designed for count data, including binary data, both of which are marginally recoverable according to Proposition 2.3.

**Example 2.5** (Latent Gaussian copula model for count data). *Let $\mathbf{X} = (X_1, \dots, X_p)^{\top}$ and $\mathbf{Y} = (Y_1, \dots, Y_p)^{\top}$ be two random $p$-vectors. $\mathbf{Y}$ is sampled from the latent Gaussian copula model for count data, if*

$$
\begin{aligned}
\mathbf{Y} \mid \mathbf{X} &\sim \prod_{j=1}^{p} \mathrm{Poisson}\left(S \exp\left(X_j\right)\right), \\
\mathbf{X} &\sim \mathrm{NPN}(\boldsymbol{\mu}, \boldsymbol{\Sigma}, f).
\end{aligned}
\tag{2}
$$

This distribution is commonly used to model genomic data (Sarkar & Stephens, 2021; Sinclair & Hooker, 2019). When $\mathbf{X} \sim N_p(\boldsymbol{\mu}, \boldsymbol{\Sigma})$, we say that $\mathbf{Y}$ follows the PLN distribution, denoted as $\mathrm{PLN}\left(S; \boldsymbol{\mu}, \boldsymbol{\Sigma}\right)$. The PLN model is widely used for single-cell RNA sequencing data, influenza-like illness dataset and purchase frequency counts (Silva et al., 2019; Wu et al., 2018; Trinh et al., 2014). In scRNA-seq data, $\mathbf{X}$ denotes the underlying expression levels of genes and $S$ represents the sequencing depth of the cell, which can be estimated by the sum of UMI counts across all genes (Sarkar & Stephens, 2021; Hafemeister & Satija, 2019).

The binary data type is an important special class of count data and is often observed in genetic and genomic studies. A prominent example is DNA nucleotide data (Abbasy et al., 2012). More concretely, genes that exhibit higher levels of expression are represented as 1, whereas genes with lower levels of expression are represented as 0. The latent Gaussian copula model for binary data is proposed by Fan et al. (2017).

**Example 2.6** (Latent Gaussian copula model for binary data). *Let $\mathbf{X} = (X_1, \dots, X_p)^{\top}$ be a random $p$-vector and $\mathbf{Y} = (Y_1, \dots, Y_p)^{\top}$ represents $p$-dimensional binary variables. $\mathbf{Y}$ is sampled from the latent Gaussian copula model for binary data, if*

$$
\begin{aligned}
Y_j &= I\left(X_j > C_j\right), \\
\mathbf{X} &\sim \mathrm{NPN}(\mathbf{0}, \boldsymbol{\Sigma}, f),
\end{aligned}
\tag{3}
$$

*where $C_j$ is a constant, $I(\cdot)$ is the indicator function, and $\sigma_{jj} = 1$ for any $1 \le j \le p$.*

In the latent Gaussian copula model for binary data, each binary component $Y_j$, which takes values of 0 or 1, is generated from a latent continuous random variable $X_j$ truncated at an unknown threshold value $C_j$.

Examples 2.5 and 2.6 can be regarded as two specific instances of marginally recoverable distributions. In both cases, $\boldsymbol{\Sigma}$ represents the covariance matrix of latent variables. In Example 2.5, $\boldsymbol{\mu}$ is the mean of the Gaussian copula distributions in the inner layer, while in Example 2.6, $\boldsymbol{\mu}$ represents the threshold values. The inverse of the covariance matrix $\boldsymbol{\Sigma}$, denoted as $\boldsymbol{\Theta}$, reveals the network. Specifically, in Example 2.5 and 2.6, $X_i$ and $X_j$ are independent given the remaining variables if and only if $\Theta_{ij} = 0$. Consequently, inferring the graph structure can be achieved by estimating $\boldsymbol{\Theta}$. However, the likelihood function in hierarchical models, such as Example 2.5 and Example 2.6, involves high-dimensional integrations that pose significant computational challenges, since these integrations seldom have closed-form solutions, making the computation of maximum likelihood estimation infeasible. Fortunately, the computational issue can be circumvented by leveraging the properties of marginally recoverable distributions and simplifying the problem of estimating the parameters of a high-dimensional distribution into multiple lower-dimensional parameter estimation problems.

## 3. Efficient Estimation

In this section, we introduce an efficient estimation framework designed to estimate parameters, particularly $\boldsymbol{\Sigma}$, which is associated with the network structure in $H_p(\boldsymbol{\mu}, \boldsymbol{\Sigma})$. Then, we extend the estimation framework to accommodate mixture distributions for inferring networks.

## 3.1. The Maximum Marginal Likelihood Estimator

Let $h_p$ denote the density function of the distribution $H_p$. Suppose that $\mathbf{Y}_i$, for $i = 1, \ldots, n$, are $n$ random $p$-dimensional vectors sampled from $H_p(\boldsymbol{\mu}, \boldsymbol{\Sigma})$. When $p$ is large, computing the maximum likelihood estimator is infeasible due to the intractable high-dimensional integration. By leveraging the properties of the marginally recoverable component as defined in Definition 2.1, we propose an efficient estimation framework based on MMLE for parameter estimation. This framework is inspired by the pairwise likelihood method (Cox & Reid, 2004; Varin, 2008), which computes only the likelihoods of pairs of observations. Therefore, this method significantly reduces the computational cost compared to conventional likelihood. This reduction in computation relies on evaluating a limited number of sets of two-dimensional integrals instead of computing the full high-dimensional likelihood integral, i.e.,

$$L_{\text{pair}}(\boldsymbol{\mu}, \boldsymbol{\Sigma}; \mathbf{y}) = \prod_{j=1}^{p-1} \prod_{k=j+1}^{p} h_2(y_j, y_k; \boldsymbol{\mu}_{[j,k]}, \boldsymbol{\Sigma}_{[j,k]}). \quad (4)$$

Unfortunately, the pairwise maximum likelihood estimator (PMLE) is often inconsistent (Varin et al., 2011). However, for marginally recoverable distributions, the PMLE is consistent because we can estimate the parameters from the two-dimensional marginals.

Note that when $\boldsymbol{\mu}$ and $\sigma_{jj}(j = 1, \ldots, p)$ are known, maximizing $L_{\text{pair}}(\boldsymbol{\mu}, \boldsymbol{\Sigma}; \mathbf{y})$ is equivalent to maximizing $h_2(y_j, y_k; \boldsymbol{\mu}_{[j,k]}, \boldsymbol{\Sigma}_{[j,k]})$ for $1 \leq j < k \leq p$. Motivated by this and to avoid redundant computation of $\boldsymbol{\mu}$ and $\sigma_{jj}$, we first estimate $\mu_j$ and $\sigma_{jj}$ by maximizing the one-dimensional marginal log-likelihood. Then, $\sigma_{jk}$ is estimated by maximizing the two-dimensional marginal log-likelihood. The MMLE $\widehat{\boldsymbol{\Sigma}} = [\hat{\sigma}_{jk}]_{1 \leq j, k \leq p}$ is derived as follows:

$$\begin{aligned}
\hat{\sigma}_{jj} &= \arg\max_{\sigma_{jj}} \sum_{i=1}^{n} \log h_1\left(Y_{ij}; \mu_j, \sigma_{jj}\right), \\
\hat{\sigma}_{jk} &= \arg\max_{\sigma_{jk}} \sum_{i=1}^{n} \log h_2\left(\mathbf{Y}_{i[j,k]}; \boldsymbol{\mu}_{[j,k]}, \boldsymbol{\Sigma}_{[j,k]}\right),
\end{aligned} \quad (5)$$

where in the optimization for $\hat{\sigma}_{jk}$, $\sigma_{jj}$ and $\sigma_{kk}$ in $\boldsymbol{\Sigma}_{[j,k]}$ are fixed to $\hat{\sigma}_{jj}$ and $\hat{\sigma}_{kk}$, respectively.

## 3.2. Mixture for Heterogeneity

The mixture model surpasses the limitations of the single-component model and offers additional flexibility in modeling complex data. In the following, we focus on the mixture model within the marginally recoverable family. Let $\boldsymbol{\pi} = (\pi_1, \ldots, \pi_G)$ denote the mixing proportions, where $\sum_{g=1}^{G} \pi_g = 1$ and $\pi_g > 0$ for $g = 1, \ldots, G$. Suppose

that the distribution $H_p$ is marginally recoverable. A $G$-component mixture of marginally recoverable distributions can be expressed as $H_p^M(\boldsymbol{\pi}, \Omega) = \sum_{g=1}^{G} \pi_g H_p(\boldsymbol{\mu}_g, \boldsymbol{\Sigma}_g)$ where $\Omega = \{\boldsymbol{\mu}_1, \ldots, \boldsymbol{\mu}_G, \boldsymbol{\Sigma}_1, \ldots, \boldsymbol{\Sigma}_G\}$.

However, due to the unknown sample categories, maximizing the marginal log-likelihood of mixed distributions is computationally intractable in practice. Therefore, we propose the EM algorithm to update the MMLE for data from mixed populations and name the estimator as EM-MMLE.

Assume that $\mathbf{Y}_i$, for $i = 1, \ldots, n$, are $n$ $p$-dimensional random vectors generated from $H_p^M(\boldsymbol{\pi}, \Omega)$. To indicate cluster membership, we introduce the indicator variable $\mathbf{Z}_i$ for the $i$-th sample, which follows a multinomial distribution with proportion parameter $\boldsymbol{\pi}$. For $1 \leq j < k \leq p$, let $\Phi_{jk} = \left\{\boldsymbol{\pi}, \boldsymbol{\mu}_{g[j,k]}, \boldsymbol{\Sigma}_{g[j,k]}, g = 1, \ldots, G\right\}$ containing the unknown parameters of $\mathbf{Y}_{[j,k]}$. In the $t$-th iteration, the expected complete log-likelihood $Q(\Phi_{jk}, \Phi_{jk}^{(t-1)})$ is taken as the optimization objective, which is defined as

$$Q(\Phi_{jk}, \Phi_{jk}^{(t-1)}) = \mathrm{E}_{p(\mathbf{Z}|\mathbf{Y}; \Phi_{jk}^{(t-1)})}\left[\log p(\mathbf{Y}, \mathbf{Z}; \Phi_{jk})\right].$$

We update $\hat{\Phi}_{jk}^{(t)} = \arg\max_{\Phi_{jk}} Q(\Phi_{jk}, \hat{\Phi}_{jk}^{(t-1)})$. The iteration terminates when the change in the optimization objective between consecutive steps is negligible. The framework of EM-MMLE is summarized in Algorithm 1.

A key application of the mixture distributions $H_p(\boldsymbol{\pi}, \Omega)$ is to infer networks from its parameters $\{\boldsymbol{\Sigma}_g, g = 1, \ldots, G\}$. Therefore, with EM-MMLE $\widehat{\boldsymbol{\Sigma}}_g$ $(g = 1, \ldots, G)$, we apply the D-trace method to estimate the sparse precision matrix $\boldsymbol{\Theta}_g = \boldsymbol{\Sigma}_g^{-1}$, i.e.,

$$\widehat{\boldsymbol{\Theta}}_g = \arg\min_{\boldsymbol{\Theta}_g \succeq 0} \frac{1}{2} \operatorname{tr}\left(\boldsymbol{\Theta}_g^2 \boldsymbol{\Sigma}_g\right) - \operatorname{tr}(\boldsymbol{\Theta}_g) + \lambda_g \|\boldsymbol{\Theta}_g\|_{1, \text{off}}. \quad (6)$$

where $\boldsymbol{\Theta}_g \succeq 0$ indicates that $\boldsymbol{\Theta}_g$ is positive semi-definite, and $\|\boldsymbol{\Theta}_g\|_{1,\text{off}} = \sum_{i \neq j} |\Theta_{gij}|$. This optimization problem is efficiently solved using an alternating direction method as described in Zhang & Zou (2014). To ensure the convexity of the loss function, the D-trace approach requires the input covariance matrix estimator to be positive semi-definite. To satisfy this requirement, we propose the $\widetilde{\boldsymbol{\Sigma}}_g$:

$$\widetilde{\boldsymbol{\Sigma}}_g = \check{\boldsymbol{\Sigma}}_g + \|\check{\boldsymbol{\Sigma}}_g - \widehat{\boldsymbol{\Sigma}}_g\|_\infty \mathbf{I}, \quad (7)$$

where $\mathbf{I}$ is the identity matrix and $\check{\boldsymbol{\Sigma}}_g = \arg\min_{\mathbf{A} \succeq 0} \|\mathbf{A} - \widehat{\boldsymbol{\Sigma}}_g\|_\infty$ is the projection of $\widehat{\boldsymbol{\Sigma}}_g$ onto the space of positive semi-definite matrices, which can be solved by a splitting conic solver (Fu et al., 2020).

The selection of the tuning parameter in Equation (6) is achieved by minimizing the approximate Bayesian informa-

**Algorithm 1** Framework of EM-MMLE

---

**Input:** Observed data $\mathbf{Y}_1, ..., \mathbf{Y}_n$, the number of populations $G$, the maximum iteration number $T$ and the convergence threshold $\epsilon_L$.

**Output:** $\widehat{\boldsymbol{\Sigma}}_1, ..., \widehat{\boldsymbol{\Sigma}}_G$.

**for** $j = 1$ **to** $p - 1$ **do**

  **for** $k = j + 1$ **to** $p$ **do**

    **while** $t < T$ and $L > \epsilon_L$ **do**

      **for** $i = 1$ **to** $n$ **do**

        **for** $g = 1$ **to** $G$ **do**

$$\hat{P}_{gijk}^{(t)} = h_2 \left( \mathbf{Y}_{i[j,k]}; \hat{\boldsymbol{\mu}}_{g[j,k]}^{(t-1)}, \widehat{\boldsymbol{\Sigma}}_{g[j,k]}^{(t-1)} \right)$$

$$\hat{\gamma}_{gijk}^{(t)} = \frac{\hat{\pi}_g^{(t-1)} \hat{P}_{gijk}^{(t)}}{\sum_{g=1}^{G} \hat{\pi}_g^{(t-1)} \hat{P}_{gijk}^{(t)}}$$

        **end for**

      **end for**

      **for** $g = 1$ **to** $G$ **do**

$$\hat{\pi}_g^{(t)} = n^{-1} \sum_{i=1}^{n} \hat{\gamma}_{gijk}^{(t)}$$

        Update $\hat{\mu}_{gj}^{(t)}$ and $\hat{\sigma}_{gjj}^{(t)}$ by maximizing $\sum_{i=1}^{n} \hat{\gamma}_{gijk}^{(t)} \log h_1(Y_{ij}; \mu_{gj}, \sigma_{gjj})$.

        Update $\hat{\mu}_{gk}^{(t)}$ and $\hat{\sigma}_{gkk}^{(t)}$ by maximizing $\sum_{i=1}^{n} \hat{\gamma}_{gijk}^{(t)} \log h_1(Y_{ik}; \mu_{gk}, \sigma_{gkk})$.

        Update $\hat{\sigma}_{gjk}^{(t)}$ by maximizing $\sum_{i=1}^{n} \hat{\gamma}_{gijk}^{(t)} \log h_2(\mathbf{Y}_{i[j,k]}; \hat{\boldsymbol{\mu}}_{g[j,k]}^{(t)}, \boldsymbol{\Sigma}_{g[j,k]})$.

      **end for**

      $L = \delta\left( Q(\hat{\Phi}_{jk}^{(t)}, \hat{\Phi}_{jk}^{(t)}), Q(\hat{\Phi}_{jk}^{(t-1)}, \hat{\Phi}_{jk}^{(t-1)}) \right)$, where $\delta(x, y) = |x - y|/y$.

    **end while**

  **end for**

**end for**

---

tion criterion

$$\left\| \frac{1}{2}(\widehat{\boldsymbol{\Theta}}_g \widehat{\boldsymbol{\Sigma}}_g + \widehat{\boldsymbol{\Sigma}}_g \widehat{\boldsymbol{\Theta}}_g) - \mathbf{I} \right\|_{\mathrm{F}} + \left( \|\widehat{\boldsymbol{\Theta}}_g\|_0 \log n \right)/n, \quad (8)$$

where $\|\widehat{\boldsymbol{\Theta}}_g\|_0$ denotes the number of nonzero entries in $\widehat{\boldsymbol{\Theta}}_g$.

## 4. Theoretical Properties

In this section, we establish theoretical results concerning the convergence rate of MMLE and the application to network estimation. We focus on the mixture of marginally recoverable distributions $H_p^M(\boldsymbol{\pi}, \Omega)$, which reduces to the single-component model discussed in Section 3.1 when $G = 1$. In theory, under the assumptions that the mixing proportions $\pi_g$ and means $\boldsymbol{\mu}_g$ $(g = 1, \ldots, G)$ are known, we can estimate $\{\boldsymbol{\Sigma}_g, g = 1, ..., G\}$ as $\left\{ \widehat{\boldsymbol{\Sigma}}_g, g = 1, ..., G \right\}$ using the MMLE.

Before presenting the theoretical results, we first provide

the necessary conditions.

Let $h_1^M(\mathbf{y}; \boldsymbol{\omega}_1)$, $h_2^M(\mathbf{y}; \boldsymbol{\omega}_2)$ represent the one-dimensional and two-dimensional marginal density functions of $H_p^M(\boldsymbol{\pi}, \Omega)$, respectively, with parameters $\boldsymbol{\omega}_1 \in \mathcal{O}_1$ and $\boldsymbol{\omega}_2 \in \mathcal{O}_2$, where $\mathcal{O}_1, \mathcal{O}_2$ are parameter spaces in finite-dimensional Euclidean space. The Hellinger distance between two densities $p_1$ and $p_2$ is defined as: $d(p_1, p_2) = \left[ \int \left( p_1^{1/2} - p_2^{1/2} \right)^2 d\nu \right]^{1/2} = \left\| p_1^{1/2} - p_2^{1/2} \right\|_{L_2}$.

**Condition 4.1** (Lower boundedness condition)**.** *For $k = 1, 2$, and for any $\boldsymbol{\omega}_k, \boldsymbol{\omega}_k' \in \mathcal{O}_k$, there exists a positive constant $c$ such that $c\|\boldsymbol{\omega}_k - \boldsymbol{\omega}_k'\|_2 \leq d(h_k^M(\mathbf{y}; \boldsymbol{\omega}_k), h_k^M(\mathbf{y}; \boldsymbol{\omega}_k'))$.*

**Condition 4.2** (Upper boundedness condition)**.** *For $k = 1, 2$, and for any $\boldsymbol{\omega}_k, \boldsymbol{\omega}_k' \in \mathcal{O}_k$, there exist a measurable function $m(\mathbf{y})$ and a constant $c$ such that $\int m^2(\mathbf{y})d\nu = c^2 < \infty$, and $\left| h_k^{M \, 1/2}(\mathbf{y}; \boldsymbol{\omega}_k) - h_k^{M \, 1/2}(\mathbf{y}; \boldsymbol{\omega}_k') \right| \leq m(\mathbf{y}) \|\boldsymbol{\omega}_k - \boldsymbol{\omega}_k'\|_2$.*

Based on these boundedness conditions, we establish a theorem that elucidates the convergence rate of MMLE $\widehat{\boldsymbol{\Sigma}}$.

**Theorem 4.3** (Rate of convergence from MMLE $\widehat{\boldsymbol{\Sigma}}_g$)**.** *For the mixture marginally recoverable model, assume that Conditions 4.1 and 4.2 hold. Then, there exists a constant $c$ such that for any $1 \leq g \leq G$ and $\epsilon > 0$ ,*
$$pr\left( \left\| \widehat{\boldsymbol{\Sigma}}_g - \boldsymbol{\Sigma}_g \right\|_\infty \geq \epsilon \right) \leq 5Gp^2 \exp\left( -cn\epsilon^2 \right).$$

Theorem 4.3 shows that if $p < \exp(c'n)$ for some constant $c'$, or in other words, if $p$ tends to infinity not faster than exponential of $n$, then $\widehat{\boldsymbol{\Sigma}}_g$ is a consistent estimator of $\boldsymbol{\Sigma}_g$.

The precision matrix $\boldsymbol{\Theta}_g = \boldsymbol{\Sigma}_g^{-1}$ encodes the network structure. Specifically, an edge exists between vertices $i$ and $j$ in the $g$-th group if $\Theta_{gij} \neq 0$; otherwise, $\Theta_{gij} = 0$ indicates no edge. With the rate of convergence for $\widehat{\boldsymbol{\Sigma}}_g$, we apply the Theorem 3 in Xiao et al. (2022) to each $\widehat{\boldsymbol{\Sigma}}_g$ and then estabilsh the sign consistency of the estimator $\widehat{\boldsymbol{\Theta}}_g$.

**Theorem 4.4** (Sign consistency for $\widehat{\boldsymbol{\Theta}}_g$)**.** *Assume that for each $g = 1, ..., G$, the true precision matrix $\boldsymbol{\Theta}_g$ satisfies the conditions described in the Appendix A.2. Then, for some $\eta > 2$, $pr\left( \mathrm{vec}(\widehat{\boldsymbol{\Theta}}_g)_{G_g^c} = 0 \right) > 1 - p^{2-\eta}$, if $n$ is sufficiently large (depending on $\eta$, see Appendix A.2), where $\mathrm{vec}(\widehat{\boldsymbol{\Theta}}_g)$ denotes the vector formed by stacking the columns of $\widehat{\boldsymbol{\Theta}}_g$, and $\mathrm{vec}(\widehat{\boldsymbol{\Theta}}_g)_{G_g^c}$ represents the subvector indexed by $G_g^c = \{(i, j) : \Theta_{gij} = 0\}$.*

Theorem 4.4 demonstrates that the estimator $\widehat{\boldsymbol{\Theta}}_g$ recovers all zeros and nonzeros in $\boldsymbol{\Theta}_g$ with probability $1 - p^{2-\eta}$. The proof of Theorem 4.4 is similar to that of Theorem 3 in Xiao et al. (2022) and is omitted here.

Notably, many mixture marginally recoverable distributions

satisfy both Conditions 4.1 and 4.2. A crucial example is the MPLN distribution, well-suited for gene regulatory network inference in scRNA-seq studies. Unlike single-model approaches requiring prior knowledge of cell type labels, the MPLN model handles network inference without such prior knowledge. Additionally, it accounts for overdispersion in scRNA-seq data and supports both positive and negative correlations (Silva et al., 2019; Tunaru, 2002). Tang et al. (2024) proposed VMPLN, a variational inference-based algorithm for estimating the precision matrices of MPLN, but it is time-consuming and lacks theoretical guarantees. In contrast, our estimation method is supported by theoretical guarantees. Under mild conditions, we show the following:

1) The MPLN distribution satisfies Conditions 4.1 and 4.2. This nontrivial proof is provided in Appendix A.3.4.

2) The binary data model in Example 2.6 also satisfies both conditions, with the proof detailed in Appendix A.4.

Thus, Theorem 4.3 and Theorem 4.4 demonstrate broad applicability.

## 5. Simulation

To evaluate the performance of our method, we conduct simulations on mixed count data and binary data.

### 5.1. Simulation for Mixed Count Data

We generate simulation data following the MPLN distribution and compare EM-MMLE with the available network inference methods including PLNet (Xiao et al., 2022), VMPLN (Tang et al., 2024), and Glasso (Friedman et al., 2008). EM-MMLE and VMPLN are developed to directly estimate the precision matrices from the MPLN model, using K-means clustering results as initial values. For PLNet and Glasso, samples are clustered using the K-means algorithm, and then the network is inferred separately for each cluster.

**Simulation Settings.** The number of populations $G$ is set as 3, and the proportion parameter $\pi$ is set as $(1/3, 1/3, 1/3)$. The synthetic datasets vary across different network structures (random, blocked random, banded, and scale-free), dimensions ($p = 100, 300$), sample sizes ($n = 1800, 3000$), population mixing levels (low, middle, and high), and zero-proportion levels that represent the proportion of zeros in the count matrix (low and high). In each scenario, we independently repeat simulations 50 times. Details of the data generation process are provided in Appendix B.1.

**Performance Comparison.** Table 1 shows the area under the precision-recall curve (AUPR) of each estimator for random graphs, calculated by varying the penalty parameters, while results for the other three graph structures

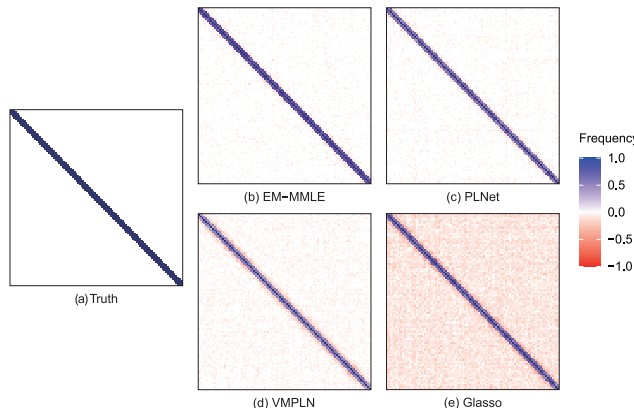

*Figure 1.* Mean networks predicted by EM-MMLE, PLNet, VMPLN, and Glasso for the banded graph with $p = 100$, $n = 3000$, high mixing, and a high zero-proportion rate. False edges are colored in red and true edges are in dark blue.

are provided in Tables 3–5 in the Appendix B.3. As expected, the AUPR decreases with an increase in the number of genes or higher zero-proportion levels. Among the four methods, EM-MMLE proves to be the most robust, consistently outperforming PLNet, VMPLN, and Glasso in AUPR across all simulated scenarios. This advantage is particularly evident in high-dimensional settings, situations with higher population mixing levels, or when zero-proportion rates are high. As the sample size increases, the AUPR of EM-MMLE improves significantly, aligning well with theoretical expectations.

Moreover, the superiority of EM-MMLE over PLNet becomes more evident with increased population mixing. For example, in simulations with a blocked random graph ($n = 3000$, $p = 300$, low zero-proportion), EM-MMLE achieves mean AUPRs of 0.84 and 0.94 in high and low mixing scenarios, respectively, approximatelyy 15% and 8% higher than PLNet's AUPRs (0.73 and 0.87).

Additionally, compared to VMPLN, another network inference method based on the MPLN model, EM-MMLE shows superior performance, especially in higher-dimensional settings. For instance, in random graph simulations ($n = 3000$, low mixing, low zero-proportion), EM-MMLE achieves mean AUPRs of 0.96 ($p = 300$) and 0.99 ($p = 100$), outperforming VMPLN by 28% and 4%, respectively.

To further assess the performance of EM-MMLE, we visualize the average network predicted by the four methods across 50 simulations. We computed the relative frequency matrix $\mathbf{F}$ to capture the accuracy of network recovery. For $\Theta_{ij} \neq 0$, $F_{ij}$ represents the proportion of simulations in which the edge was correctly recovered. Conversely, when $\Theta_{ij} = 0$, $F_{ij}$ denotes the negative proportion of simulations in which an edge between nodes $i$ and $j$ was incorrectly

*Table 1.* Comparisons of EM-MMLE with PLNet, VMPLN and Glasso in terms of AUPR on simulation results for random graphs generated by the MPLN model. The results are averages over 50 replicates with standard deviations in brackets.

| Zero-proportion | | | Low | | | |
|---|---|---|---|---|---|---|
| Dimension | | $p = 100$ | | | $p = 300$ | |
| Mixing degree | Low | Middle | High | Low | Middle | High |
| | | | $n = 1800$ | | | |
| EM-MMLE | **0.95 (0.013)** | **0.94 (0.012)** | **0.91 (0.018)** | **0.86 (0.024)** | **0.81 (0.022)** | **0.72 (0.04)** |
| PLNet | 0.89 (0.03) | 0.86 (0.046) | 0.81 (0.047) | 0.74 (0.061) | 0.67 (0.066) | 0.57 (0.085) |
| VMPLN | 0.9 (0.023) | 0.9 (0.022) | 0.89 (0.037) | 0.67 (0.026) | 0.66 (0.017) | 0.64 (0.022) |
| Glasso | 0.83 (0.036) | 0.8 (0.032) | 0.76 (0.047) | 0.72 (0.028) | 0.69 (0.027) | 0.61 (0.034) |
| | | | $n = 3000$ | | | |
| EM-MMLE | **0.99 (0.006)** | **0.98 (0.006)** | **0.98 (0.008)** | **0.96 (0.008)** | **0.94 (0.011)** | **0.87 (0.019)** |
| PLNet | 0.95 (0.047) | 0.95 (0.033) | 0.93 (0.03) | 0.91 (0.052) | 0.87 (0.063) | 0.75 (0.052) |
| VMPLN | 0.95 (0.028) | 0.95 (0.021) | 0.94 (0.016) | 0.75 (0.021) | 0.74 (0.013) | 0.72 (0.019) |
| Glasso | 0.89 (0.04) | 0.88 (0.033) | 0.85 (0.035) | 0.82 (0.021) | 0.79 (0.019) | 0.72 (0.024) |
| Zero-proportion | | | High | | | |
| Dimension | | $p = 100$ | | | $p = 300$ | |
| Mixing degree | Low | Middle | High | Low | Middle | High |
| | | | $n = 1800$ | | | |
| EM-MMLE | **0.83 (0.026)** | **0.8 (0.037)** | **0.75 (0.03)** | **0.62 (0.041)** | **0.55 (0.028)** | **0.49 (0.03)** |
| PLNet | 0.72 (0.073) | 0.69 (0.059) | 0.61 (0.069) | 0.55 (0.064) | 0.47 (0.04) | 0.41 (0.031) |
| VMPLN | 0.77 (0.037) | 0.75 (0.034) | 0.72 (0.033) | 0.47 (0.034) | 0.45 (0.025) | 0.44 (0.021) |
| Glasso | 0.55 (0.045) | 0.49 (0.046) | 0.45 (0.045) | 0.47 (0.032) | 0.44 (0.028) | 0.41 (0.026) |
| | | | $n = 3000$ | | | |
| EM-MMLE | **0.95 (0.01)** | **0.94 (0.013)** | **0.92 (0.017)** | **0.83 (0.028)** | **0.79 (0.025)** | **0.71 (0.037)** |
| PLNet | 0.89 (0.037) | 0.87 (0.048) | 0.81 (0.065) | 0.77 (0.048) | 0.71 (0.039) | 0.62 (0.048) |
| VMPLN | 0.86 (0.038) | 0.84 (0.042) | 0.81 (0.047) | 0.56 (0.031) | 0.55 (0.026) | 0.54 (0.018) |
| Glasso | 0.63 (0.04) | 0.6 (0.06) | 0.55 (0.063) | 0.58 (0.029) | 0.58 (0.033) | 0.53 (0.043) |

predicted. Figure 2 shows the results for the banded graph with $p = 100$, $n = 3000$, high mixing, and a high zero-proportion rate. EM-MMLE closely aligns with the true network structure, detecting more true edges while maintaining the lowest false positive rate compared to other methods.

### 5.2. Simulation for Binary Data

To evaluate the performance of MMLE in estimating the inverse correlation matrix $\Theta$ for binary data, we follow a similar data-generating procedure as described in Fan et al. (2017). Specifically, we generate simulation data $\mathbf{Y} = (Y_1, \ldots, Y_p)^\top$, where $Y_j = I(X_j > C_j)$ for $j = 1, \ldots, p$, with $\mathbf{X} \sim N_p(\mathbf{0}, \mathbf{\Sigma})$.

We then compare the performance of MMLE with three estimation methods: the rank-based estimator by Fan et al.(2017), AMLE (d'Aspremont et al., 2008), and an Oracle estimator, which utilizes the Pearson correlation of the latent variable $\mathbf{X}$ within the D-trace loss to benchmark the information loss of estimators derived from observed data $\mathbf{Y}$. The rank-based estimator by Fan et al.(2017) is designed to estimate the correlation matrix in a latent Gaussian copula model. To estimate the precision matrix, we apply D-trace to the correlation matrix from this estimator to assess its accuracy in estimating $\Theta$. AMLE is a graphical lasso estimator that takes the modified sample covariance matrix $\mathbf{\Sigma}_A$ as

its input, where $\mathbf{\Sigma}_A = \frac{1}{n} \sum_{i=1}^{n} \left(\mathbf{X}_i - \bar{\mathbf{X}}\right) \left(\mathbf{X}_i - \bar{\mathbf{X}}\right)^\top + \frac{1}{3}$ and $\bar{\mathbf{X}} = \frac{1}{n} \sum_{i=1}^{n} \mathbf{X}_i$.

**Simulation Settings.** We set $p = 50, 200$ and evaluate the performance for three sample sizes: $n = 200, 500, 3000$. Each simulation scenario is repeated 100 times. The data generative process is detailed in Appendix B.2

**Performance Comparison.** Table 2 presents the average estimation errors of $\widehat{\mathbf{\Sigma}} - \mathbf{\Sigma}$ and $\widehat{\mathbf{\Theta}} - \mathbf{\Theta}$ as measured by the Frobenius norm. It is seen that the estimation errors of MMLE and Fan's method remain nearly identical across different dimensions and both are significantly lower than those of AMLE. When $n$ is small, MMLE demonstrates higher accuracy in estimating $\mathbf{\Sigma}$ compared to Fan's method. Notably, Fan's method can only handle model (3), whereas MMLE offers greater generalizability. Compared to the benchmark Oracle estimator, the results in Table 2 indicates that MMLE has almost no information loss at $n = 200$ and $n = 500$, and only moderate information loss in the high-dimensional environment at $n = 3000$.

## 6. Application to a scRNA-seq Dataset

In this section, we evaluate the performance of EM-MMLE for gene regulatory network inference using a real scRNA-seq dataset (Zheng et al., 2017), comprising 6,952 cells

*Table 2.* Average estimation error of MMLE, Fan et al.'s method, Oracle, and AMLE measured by the Frobenius norm on binary data. The results are averages over 100 replicates with standard deviations in brackets.

| $p$ | | 50 | | | 200 | |
|---|---|---|---|---|---|---|
| $n$ | 200 | 500 | 3000 | 200 | 500 | 3000 |
| | | | $\widehat{\Sigma} - \Sigma$ | | | |
| Oracle | **3.599 (0.02)** | **2.173 (0.01)** | **0.909 (0.01)** | **14.153 (0.03)** | **8.911 (0.02)** | **3.629 (0.01)** |
| MMLE | 6.343 (0.15) | 3.978 (0.07) | 1.597 (0.03) | 25.84 (0.14) | 15.869 (0.11) | 6.529 (0.03) |
| Fan et al. | 6.375 (0.15) | 3.986 (0.07) | 1.598 (0.03) | 25.978 (0.14) | 15.901 (0.11) | 6.531 (0.03) |
| AMLE | 31.254 (0.11) | 31.247 (0.06) | 31.036 (0.03) | 125.278 (0.31) | 123.72 (0.36) | 124.367 (0.08) |
| | | | $\widehat{\Theta} - \Theta$ | | | |
| Oracle | **2.327 (0.03)** | 2.335 (0.03) | **1.12 (0.08)** | **2.365 (0.02)** | **2.342 (0.03)** | **1.045 (0.03)** |
| MMLE | 2.328 (0.03) | 2.334 (0.03) | 1.335 (0.06) | 2.365 (0.02) | 2.363 (0.03) | 2.332 (0.04) |
| Fan et al. | 2.327 (0.03) | **2.333 (0.03)** | 1.336 (0.09) | 2.365 (0.02) | 2.363 (0.03) | 2.33 (0.04) |
| AMLE | 2.684 (0.34) | 2.834 (0.03) | 2.198 (0.03) | 2.365 (0.02) | 3.564 (0.02) | 2.871 (0.04) |

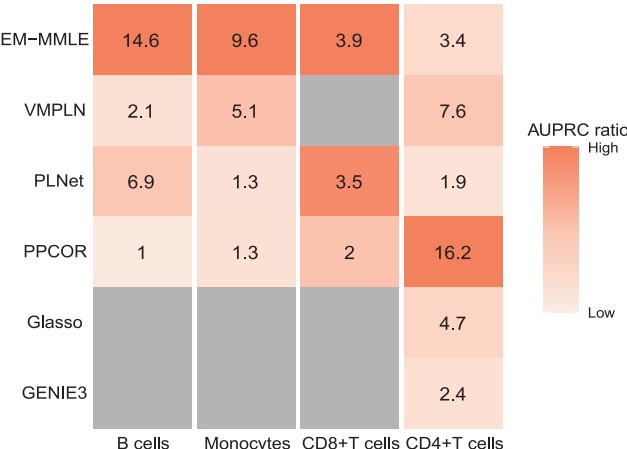

*Figure 2.* AUPRC ratios of network inference algorithms on the scRNA-seq dataset. Algorithms are ordered by decreasing median AUPRC ratios. The color in each cell is proportional to the corresponding value (scaled between 0 and 1, with values below those of a random predictor shown as grey squares). The actual values are displayed in the boxes.

across four cell types. The dataset includes two batches, sequenced by 3' and 5' scRNA-seq technologies. One batch is used to construct silver standards based on public regulatory network databases (Appendix C.1), while the other batch is reserved for algorithm testing. We infer regulatory interactions for the top 300 highly variable genes identified by Seurat (Stuart et al., 2019) and assess the results by comparison to these silver standards.

To compare network predictions fairly, we use the AUPRC ratio, a metric from a previous benchmark (Pratapa et al., 2020) (see Appendix C.2). Algorithms are compared at a fixed network density (5%), with AUPRC ratios calculated relative to the silver standards. Figure 2 presents the

heatmap of AUPRC ratios for six methods (four evaluated in the simulation and two additional single-cell gene regulatory network inference methods: PPCOR (Kim, 2015) and GENIE3 (Huynh-Thu et al., 2010)). EM-MMLE achieves the highest AUPRC ratios in most cases and consistently outperforms the random predictor across all cell types.

From a biological perspective, some interesting association patternss are identified by EM-MMLE (Figure 3). It reveals an association between the genes MYBL2 and TK1, which is predicted across the regulatory networks of three cell types. This finding is supported by the literature (Qiu et al., 2022), which links these genes to N glycan biosynthesis and p53 signaling pathways. Additionally, EM-MMLE predicts an association between the genes MYBL2 and BIRC5, supported by Li et al. (2021). Interestingly, we also identified a cell-type-specific hub gene ID3 in the gene regulatory network of CD4+ T cells. Gene ontology analysis of ID3 target genes (Figure 4) reveals enrichment in terms related to antigen processing and presentation, including exogenous and endogenous peptide antigens via MHC class II. The relationship between CD4+ T cells and MHC class II is central to adaptive immune responses in the immune system.

## 7. Discussion

In this paper, we introduce a new and generic family of graphical models, the marginally recoverable family, which is not limited to specific models or data types. We propose MMLE for parameter estimation with theoretical guarantees, avoiding high-dimensional integration, and introduce EM-MMLE to handle mixture distributions and capture heterogeneous structures. The effectiveness of our method is demonstrated through several specific distributions.

To facilitate intuitive network representation, we impose certain restrictions on the form of the parameters within

the marginally recoverable family and require that the one-dimensional and two-dimensional marginal distributions satisfy specific conditions. However, motivated by the underlying principles of the family, we can provide a more general definition that extends to distributions determined by higher moments.

Let $\mathbf{X}$ be a $p$-dimensional random variable, and let $d < p$ be a fixed integer. We say that $\mathbf{X}$ is $d$-marginally recoverable if any $d$-dimensional marginal distribution of $\mathbf{X}$ belongs to the same distribution family, and the parameters of all $d$-dimensional marginal distributions collectively characterize the parameters of the full distribution.

This generalization improves the flexibility of the marginally recoverable family and broadens its applicability to more general parameter estimation problems. Future work will explore additional distributions within this framework and apply the proposed estimation method to diverse real-world applications.

## Acknowledgements

We thank the anonymous reviewers for their valuable comments and Zihao Chen for helpful discussions. This work was supported by the National Key R&D Program of China (2024YFF0507404 to R.X.), the National Natural Science Foundation of China (12425110, 12371286 to R.X., 12271287 to W.W.), and the Sino-Russian Mathematics Center. Part of the analysis was performed on the high-performance computing platform of the Center for Life Sciences (Peking University).

## Impact Statement

This paper presents work whose goal is to advance the field of machine learning. There are many potential societal consequences of our work, none which we feel must be specifically highlighted here.

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

# A. Theoretical Results and Proofs

## A.1. Proofs for Theorem 4.3

In this section, we first provide some lemmas concerning the Maximum Likelihood Estimator (MLE) and subsequently prove Theorem 4.3.

Recall that $h_1^M(\mathbf{y}; \boldsymbol{\omega}_1)$, $h_2^M(\mathbf{y}; \boldsymbol{\omega}_2)$ denote the one-dimensional and two-dimensional marginal density functions of $H_p^M(\boldsymbol{\pi}, \Omega)$, respectively, with parameters $\boldsymbol{\omega}_1 \in \mathcal{O}_1$ and $\boldsymbol{\omega}_2 \in \mathcal{O}_2$, where $\mathcal{O}_1, \mathcal{O}_2$ are parameter spaces in finite-dimensional Euclidean space. For simplicity, we omit the subscripts and use $h^M(\mathbf{y}; \boldsymbol{\omega})$ to represent either the one-dimensional or two-dimensional marginal density function of $H_p^M(\boldsymbol{\pi}, \Omega)$, where $\boldsymbol{\omega} \in \mathcal{O}$ is a $k$-dimensional parameter vector. Define $\mathcal{F} = \left\{ h^M(\mathbf{y}; \boldsymbol{\omega}) : \boldsymbol{\omega} \in \mathcal{O} \right\}$.

For any $u > 0$, if there exists a finite set $\{(f_j^L, f_j^U), j = 1, \ldots, N\}$ such that $\left\| (f_j^L)^{1/2} - (f_j^U)^{1/2} \right\|_2 \leq u$ for $j = 1, ..., N$ and for any $h^M(\mathbf{y}; \boldsymbol{\omega}) \in \mathcal{F}$, there exists a $j$ such that $f_j^L \leq h^M(\mathbf{y}; \boldsymbol{\omega}) \leq f_j^U$, we say that $\{(f_j^L, f_j^U), j = 1, \ldots, N\}$ is a Hellinger $u$-bracketing of $\mathcal{F}$, and $N$ is the size of the Hellinger $u$-bracketing. Let $\mathcal{N}_u$ be the set of sizes of all Hellinger $u$-bracketings. We define the bracketing Hellinger metric entropy of $\mathcal{F}$ as

$$H(u, \mathcal{F}) = \min_{N \in \mathcal{N}_u} \log N.$$

### A.1.1. LEMMAS

**Lemma A.1.** *There exist positive constants $c_1, c_2, c_3$, such that, for any $\epsilon > 0$, if*

$$\int_{s^2/2^8}^{\sqrt{2}s} H^{1/2}\left( u/c_2, \left\{ p_1 \in \mathcal{F} : d(p_1, h^M(\mathbf{y}; \boldsymbol{\omega})) \leq \sqrt{2}s \right\} \right) du \leq c_3 n^{1/2} s^2 \tag{9}$$

*for all $s \geq \epsilon$, then for the MLE $\hat{\boldsymbol{\omega}}$ of the true parameter $\boldsymbol{\omega}$ using $n$ independent samples, we have*

$$pr\left( \left\| h^{M\,1/2}(\mathbf{y}; \hat{\boldsymbol{\omega}}) - h^{M\,1/2}(\mathbf{y}; \boldsymbol{\omega}) \right\|_{L_2} \geq \epsilon \right) \leq 5 \exp(-c_1 n \epsilon^2)$$

.

Lemma A.1 is a local version of Theorems 1 and 2 from Wong & Shen (1995).

**Lemma A.2.** *Let $\hat{\boldsymbol{\omega}}$ be the MLE of the true parameter $\boldsymbol{\omega}$ restricted on $\mathcal{O}$. Under Condition 4.1 and Condition 4.2, there exists a positive constant $c$ independent with parameters, such that, for any $\epsilon > 0$, we have*

$$pr\left( \| \hat{\boldsymbol{\omega}} - \boldsymbol{\omega} \|_2 \geq \epsilon \right) \leq 5 \exp(-cn\epsilon^2).$$

*Proof.* To prove Lemma A.2, we adopt a proof framework similar to that of Lemma S11 in Xiao et al. (2022). First we will show that there exist postive constants $c_4, c_5$, such that,

$$H\left( u, \left\{ p_1 \in \mathcal{F} : d(p_1, h^M(\mathbf{y}; \boldsymbol{\omega})) \leq \sqrt{2}s \right\} \right) \leq c_4 \log(c_5 s/u). \tag{10}$$

According to the lower boundedness condition 4.1, the parameters whose index in $\left\{ p_1 \in \mathcal{F} : d(p_1, h^M(\mathbf{y}; \boldsymbol{\omega})) \leq \sqrt{2}s \right\}$ can be covered by $\mathcal{F}_{\boldsymbol{\omega},s} = \left\{ \boldsymbol{\omega}' \mid \| \boldsymbol{\omega}' - \boldsymbol{\omega} \|_2 \leq \sqrt{2}s/C_1 \right\}$, where $C_1$ is a constant. Using the upper boundedness condition 4.2, for any $\boldsymbol{\omega}, \boldsymbol{\omega}' \in \mathcal{O}$, there exist a measurable function $m(\mathbf{y})$ and a constant $C_2$ such that $\int m^2(\mathbf{y}) d\nu = C_2^2 < \infty$, and $\left| h^{M\,1/2}(\mathbf{y}; \boldsymbol{\omega}) - h^{M\,1/2}(\mathbf{y}; \boldsymbol{\omega}') \right| \leq m(\mathbf{y}) \| \boldsymbol{\omega} - \boldsymbol{\omega}' \|_2$. It is straightforward to verify that we can cover the set $\mathcal{F}_{\boldsymbol{\omega},s}$ by at most $\left( 2\sqrt{2} C_2 s / C_1 u \right)^k$ balls, while each ball has a radius of $u/2C_2$. For any ball $\mathcal{B}$ with radius $u/2C_2$, we define the centre of $\mathcal{B}$ as $\boldsymbol{\omega}_0$, then $\| \boldsymbol{\omega}' - \boldsymbol{\omega}_0 \|_2 \leq u/2C_2$ for any $\boldsymbol{\omega}' \in \mathcal{B}$. Then, we have $\left| h^{M\,1/2}(\mathbf{y}; \boldsymbol{\omega}') - h^{M\,1/2}(\mathbf{y}; \boldsymbol{\omega}_0) \right| \leq m(\mathbf{y}) u/2C_2$. Therefore, we can select the minimum and maximum density within each ball as follows:

$$f_L^{1/2} = \max\left\{ h^{M\,1/2}(\mathbf{y}; \boldsymbol{\omega}_0) - m(\mathbf{y}) u/2C_2, 0 \right\}, \quad f_U^{1/2} = h^{M\,1/2}(\mathbf{y}; \boldsymbol{\omega}_0) + m(\mathbf{y}) u/2C_2.$$

Consequently, we have $d(f_L, f_U) \leq (\int m^2(\mathbf{y})u^2/C_2^2 d\mathbf{y})^{1/2} = u$, thus we can derive (10).

Next, to apply Lemma A.1, we aim to prove the existence of positive constants $c_2$ and $c'$ such that (9) holds for any $s \geq c'n^{-1/2}$. The case that $s < c'n^{-1/2}$ will be discussed later. According to the Cauchy inequality $\left(\int_a^b f^{1/2} dx\right)^2 \leq (b-a)\int_a^b f\, dx$,

$$
\begin{aligned}
&\left(\int_{s^2/2^8}^{\sqrt{2}s} H^{1/2}\left(u/c_2, \left\{p_1 \in \mathcal{F} : d(p_1, h^M(\mathbf{y};\boldsymbol{\omega})) \leq \sqrt{2}s\right\}\right) du\right)^2 \\
&\leq (\sqrt{2}s - s^2/2^8) \int_{s^2/2^8}^{\sqrt{2}s} H\left(u/c_2, \left\{p_1 \in \mathcal{F} : d(p_1, h^M(\mathbf{y};\boldsymbol{\omega})) \leq \sqrt{2}s\right\}\right) du \\
&\leq \sqrt{2}s \int_{s^2/2^8}^{\sqrt{2}s} H\left(u/c_2, \left\{p_1 \in \mathcal{F} : d(p_1, h^M(\mathbf{y};\boldsymbol{\omega})) \leq \sqrt{2}s\right\}\right) du \\
&\leq \sqrt{2}s \int_{s^2/2^8}^{\sqrt{2}s} c_4 \log(c_2 c_5 s/u) du
\end{aligned}
\tag{11}
$$

Thus, it suffices to show there exist positive constants $c_2$ and $c'$ such that

$$
\int_{s^2/2^8}^{\sqrt{2}s} \log(c_2 c_5 s/u)\, du \leq ns^3,
\tag{12}
$$

for all $s \geq c'n^{-1/2}$.

Let $c_2 \geq c_5^{-1}$. After calculating the left hand of (12), we have

$$
\begin{aligned}
\int_{s^2/2^8}^{\sqrt{2}s} \log(c_2 c_5 s/u) du &= \log(c_2 c_5)(\sqrt{2}s - s^2/2^8) + [u - u\log u]\big|_{s^2/2^8}^{\sqrt{2}s} \\
&\leq -\sqrt{2}s\log(\sqrt{2}) - s^2/2^8(\log(s) - \log(s^2/2^8)) + (\sqrt{2}s - s^2/2^8) + \log(c_2 c_5)\sqrt{2}s \\
&= \sqrt{2}\log(c_2 c_5 e/\sqrt{2})s - s^2/2^8 \log(2^8 e/s).
\end{aligned}
\tag{13}
$$

Note that $\sqrt{2}s \geq s^2/2^8$ implies $\log(2^8 e/s) > 0$. Therefore, if $ns^2 \geq \sqrt{2}\log(c_2 c_5 e/\sqrt{2})$, i.e., $s \geq c'n^{-1/2}$ where $c' = \left(\sqrt{2}\log(c_2 c_5 e/\sqrt{2})\right)^{1/2}$, inequality (12) will hold.

Then using Lemma A.1, there exists a positive constant $c_1$ such that

$$
pr\left(\left\|h^{M\,1/2}(\mathbf{y};\hat{\boldsymbol{\omega}}) - h^{M\,1/2}(\mathbf{y};\boldsymbol{\omega})\right\|_{L_2} \geq \epsilon\right) \leq 5\exp(-c_1 n\epsilon^2),
$$

for any $\epsilon \geq c'n^{-1/2}$. Notice that for $0 < \epsilon < c'n^{-1/2}$, we can have a constant $c_0$ to satisfy

$$
pr\left(\left\|h^{M\,1/2}(\mathbf{y};\hat{\boldsymbol{\omega}}) - h^{M\,1/2}(\mathbf{y};\boldsymbol{\omega})\right\|_{L_2} \geq \epsilon\right) \leq 5\exp(-c_0 n\epsilon^2).
$$

Choosing constant $c = \min\{c_0, c_1\}$, we have for any $\epsilon > 0$,

$$
pr\left(\left\|h^{M\,1/2}(\mathbf{y};\hat{\boldsymbol{\omega}}) - h^{M\,1/2}(\mathbf{y};\boldsymbol{\omega})\right\|_{L_2} \geq \epsilon\right) \leq 5\exp(-cn\epsilon^2).
$$

Noting that $\boldsymbol{\omega}, \hat{\boldsymbol{\omega}} \in \mathcal{O}$, we apply the lower boundedness condition 4.1 to obtain

$$
\left\{d(h^M(\mathbf{y};\hat{\boldsymbol{\omega}}), h^M(\mathbf{y};\boldsymbol{\omega})) < \epsilon\right\} \subseteq \left\{\|\hat{\boldsymbol{\omega}} - \boldsymbol{\omega}\|_2 < \epsilon/C_1\right\},
$$

where $C_1$ is a constant. Thus,

$$
pr\left(\|\hat{\boldsymbol{\omega}} - \boldsymbol{\omega}\|_2 < \epsilon/C_1\right) \geq pr\left(d(h^M(\mathbf{y};\hat{\boldsymbol{\omega}}), h^M(\mathbf{y};\boldsymbol{\omega})) < \epsilon\right) \geq 1 - 5\exp(-cn\epsilon^2),
$$

which implies that

$$
pr\left(\|\hat{\boldsymbol{\omega}} - \boldsymbol{\omega}\|_2 \geq \epsilon'\right) \leq 5\exp(-Cn\epsilon'^2),
$$

for a constant $C$ and all $\epsilon' > 0$. Then we finish our proof. $\qquad\square$

### A.1.2. Proof of Theorem 4.3

*Proof.* Recall the definition of MMLE in Equation (5). Apply Lemma A.2 to $Y_j$ and $Y_{[j,k]}$ for any $1 \leq j < k \leq p$, we have $pr\left(|\hat{\sigma}_{gjk} - \sigma_{gjk}| \geq \epsilon\right) \leq 5 \exp\left(-Cn\epsilon^2\right)$ for any $1 \leq j \leq k \leq p$, $1 \leq g \leq G$ and $\epsilon > 0$. Thus, we have $pr\left(\left\|\widehat{\boldsymbol{\Sigma}}_g - \boldsymbol{\Sigma}_g\right\|_\infty \geq \epsilon\right) \leq 5Gp^2 \exp\left(-cn\epsilon^2\right)$ for any $1 \leq g \leq G$ and $\epsilon > 0$ with a constant $c$ and finish the proof. $\square$

## A.2. Detailed Restatement of Theorem 4.4

For a vector $\mathbf{a}$ with the $i$-th entry $a_i$, let $\|\mathbf{a}\|_1 = \sum_i |a_i|$. For a matrix $\mathbf{A}$, let $\lambda_{\max}(\mathbf{A})$ and $\lambda_{\min}(\mathbf{A})$ be the largest and smallest eigenvalues of $\mathbf{A}$, $\mathrm{vec}(\mathbf{A})$ denote the vector formed by stacking the columns of $\mathbf{A}$, $\|\mathbf{A}\|_0 = \sum_{i,j} I(A_{ij} \neq 0)$ denote the number of nonzero entries, $\|\mathbf{A}\|_{1,\mathrm{off}} = \sum_{i \neq j} |A_{ij}|$ and $\|\mathbf{A}\|_{1,\infty} = \max_i(\sum_j |A_{ij}|)$.

For each $g = 1, ..., G$, we introduce the following notation. Define $G_g = \{(i,j)|\Theta_{gij} \neq 0\}$ as the set of positions corresponding to nonzero elements in $\boldsymbol{\Theta}_g$ and $G_g^c = \{(i,j)|\Theta_{gij} = 0\}$. Let $d$ denote the maximum node degree in $\boldsymbol{\Theta}_g$, $s = \|\boldsymbol{\Theta}_g\|_0$, and $\theta_{\min} = \min_{(i,j) \in G_g} |\Theta_{gij}|$ be the minimal absolute value of nonzero elements of $\boldsymbol{\Theta}_g$. We define $\boldsymbol{\Gamma}^* = \Gamma(\boldsymbol{\Sigma}_g) = (\boldsymbol{\Sigma}_g \otimes \mathbf{I} + \mathbf{I} \otimes \boldsymbol{\Sigma}_g)/2$, where $\mathbf{A} \otimes \mathbf{B}$ is the Kronecker product. For two subsets $T_1$ and $T_2$ of $\{(i,j) \mid 1 \leq i, j \leq p\}$, we define $\Gamma(\mathbf{A})_{T_1, T_2}$ be the submatrix of $\Gamma(\mathbf{A})$ whose rows and columns indexed by $T_1$ and $T_2$, respectively. Other notations are as follows,

$$\gamma = 1 - \max_{(i,j) \in G_g^c} \left\| \boldsymbol{\Gamma}^*_{(i,j), G_g} (\boldsymbol{\Gamma}^*_{G_g, G_g})^{-1} \right\|_1,$$

$$k_\Gamma = \left\| (\boldsymbol{\Gamma}^*_{G_g, G_g})^{-1} \right\|_{1,\infty}, \quad k_\Sigma = \|\boldsymbol{\Sigma}_g\|_{1,\infty}.$$

Recall that the precision matrix estimator $\widehat{\boldsymbol{\Theta}}_g$ is defined as:

$$\widehat{\boldsymbol{\Theta}}_g = \arg \min_{\boldsymbol{\Theta}_g \succeq 0} \frac{1}{2} \mathrm{tr}\left(\boldsymbol{\Theta}_g^2 \boldsymbol{\Sigma}_g\right) - \mathrm{tr}(\boldsymbol{\Theta}_g) + \lambda_g \|\boldsymbol{\Theta}_g\|_{1,\mathrm{off}}. \tag{14}$$

We now provide a detailed restatement of Theorem 4.4, aligning the conditions with those in Theorem 3 of Xiao et al. (2022). For each $g = 1, ..., G$, there exist constants $a_g$ and $b_g$, such that for some $\eta > 2$, if the true precision matrix $\boldsymbol{\Theta}_g$ satisfies the following conditions:

$$n > b_g^{-1}(\eta \log p + \log a_g) \max\left[12 d k_\Gamma, \ 12\gamma^{-1}(k_\Sigma k_\Gamma^2 + k_\Gamma), \ \left\{12\gamma^{-1}\left(k_\Sigma k_\Gamma^3 + k_\Gamma^2\right) + 5 d k_\Gamma^2\right\} \theta_{\min}^{-1},\right.$$

$$\min\left\{s^{1/2}, d+1\right\}\left\{12\gamma^{-1}\left(k_\Sigma k_\Gamma^3 + k_\Gamma^2\right) + 5 d k_\Gamma^2\right\} \lambda_{\min}^{-1}(\boldsymbol{\Theta}_g)\right]^2,$$

and

$$\lambda_g = 12\gamma^{-1}\left(k_\Sigma k_\Gamma^2 + k_\Gamma\right) b_g^{-1/2}(\eta \log p + \log a_g)^{1/2} n^{-1/2},$$

then

$$pr\left(\mathrm{vec}(\widehat{\boldsymbol{\Theta}}_g)_{G_g^c} = 0\right) > 1 - p^{2-\eta},$$

where $\mathrm{vec}(\widehat{\boldsymbol{\Theta}}_g)_{G_g^c}$ represents the subvector indexed by $G_g^c = \{(i,j) : \Theta_{gij} \neq 0\}$.

The proof of Theorem 4.4 is similar to that of Theorem 3 in Xiao et al. (2022) and is omitted here.

## A.3. Theoretical Results and Proofs for the MPLN Model

Recall the definition of the PLN distribution from the manuscript:

$$\mathbf{y} \mid \mathbf{x} \sim \prod_{j=1}^p \mathrm{Poisson}\left\{S \exp\left(x_j\right)\right\},$$

$$\mathbf{x} \sim \mathrm{N}(\boldsymbol{\mu}, \boldsymbol{\Sigma}), \tag{15}$$

where $S$ denotes the total sequencing reads, which can be estimated as the sum of counts per cell or by other methods (Anders & Huber, 2010; Hafemeister & Satija, 2019). Without loss of generality, we assume that $S$ is known and constant. For simplicity, we set $S = 1$ in the proof.

We hereafter represent the PLN distribution as $\mathrm{PLN}(\boldsymbol{\Theta}, \boldsymbol{\mu})$, where $\boldsymbol{\Theta} = \boldsymbol{\Sigma}^{-1}$. For the MPLN distribution, defined as $\sum_{g=1}^{G} \pi_g \mathrm{PLN}\left(\boldsymbol{\Theta}_g, \boldsymbol{\mu}_g\right)$, we assume that the true means $\boldsymbol{\mu}_g$ and proportions $\pi_g$ $(g = 1, \ldots, G)$ are known.

We impose the following conditions:

**Condition A.3.** *There exist positive constants $m$, $M$, and $l$, such that $\max_{1 \le j, k \le p} |\sigma_{gjk}| \le l$ and $m \le \lambda_{\min}\left(\boldsymbol{\Theta}_g\right) \le \lambda_{\max}\left(\boldsymbol{\Theta}_g\right) \le M$ for $g = 1, \ldots, G$.*

**Condition A.4.** *The parameters $(\boldsymbol{\mu}_g^\top, \mathrm{vech}(\boldsymbol{\Theta}_g)^\top)^\top$ $(g = 1, \cdots, G)$ are different from each other.*

Based on these conditions, we establish the following theoretical results for the MPLN model:

**Theorem A.5.** *Under Conditions A.3 and A.4, the MPLN model is identifiable, and its Fisher information matrix is positive definite.*

**Theorem A.6.** *Under Conditions A.3 and A.4, the MPLN model satisfies Conditions 4.1 and 4.2.*

Theorem A.5 establishes the identifiability of the MPLN model and the positive definiteness of its Fisher information matrix, ensuring that the model behaves well under relatively mild conditions. Building on Theorem A.5, we demonstrate Theorem A.6, which represents a key theoretical contribution of this study.

In this section, we first introduce some notations, followed by the presentation of several lemmas. Finally, we provide proofs for Theorem A.5 and Theorem A.6.

### A.3.1. NOTATION

We define two vectorization operators, $\mathrm{vech}$ and $\mathrm{vech}_2$. For a symmetric matrix $\mathbf{A} = [a_{jk}]_{1 \le j, k \le p} \in \mathbb{R}^{p \times p}$, $\mathrm{vech}(\mathbf{A})$ is defined as

$$\mathrm{vech}(\mathbf{A}) = (a_{11}, a_{12}, a_{13}, \ldots, a_{1p}, a_{22}, a_{23}, \ldots, a_{2p}, \ldots, a_{(p-1)(p-1)}, a_{(p-1)p}, a_{pp})^\top,$$

and $\mathrm{vech}_2(\mathbf{A})$ is

$$\mathrm{vech}_2(\mathbf{A}) = (a_{11}, 2a_{12}, 2a_{13}, \ldots, 2a_{1p}, a_{22}, 2a_{23}, \ldots, 2a_{2p}, \ldots, a_{(p-1)(p-1)}, 2a_{(p-1)p}, a_{pp})^\top.$$

Note that $\mathrm{vech}$ and $\mathrm{vech}_2$ only differ at $i \ne j$ elements.

We represent the MPLN distribution as $\mathrm{MPLN}\left(\boldsymbol{\nu}, \{\boldsymbol{\mu}_g\}_{g=1}^{G}\right)$, where $\boldsymbol{\nu} = (\boldsymbol{\nu}_1^\top, \boldsymbol{\nu}_2^\top, \ldots, \boldsymbol{\nu}_G^\top)^\top$ and $\boldsymbol{\nu}_g = \mathrm{vech}(\boldsymbol{\Theta}_g)$.

Define $\boldsymbol{\varphi} = (\boldsymbol{\varphi}_1^\top, \boldsymbol{\varphi}_2^\top, \ldots, \boldsymbol{\varphi}_G^\top)^\top$, where $\boldsymbol{\varphi}_g = \mathrm{vech}(\boldsymbol{\Sigma}_g)$. Since $\boldsymbol{\Sigma}_g$ is positive definite, the mapping between $\boldsymbol{\varphi}$ and $\boldsymbol{\nu}$ is bijective. In the following discussion, defining either $\boldsymbol{\varphi}$ or $\boldsymbol{\nu}$ implicitly determines the other.

Additionally, we define the bounded sets:

$$\mathcal{D}_p^M = \{\boldsymbol{\nu} = \left(\boldsymbol{\nu}_1^\top, \ldots, \boldsymbol{\nu}_G^\top\right)^\top \mid \max_{1 \le j, k \le p} |\Theta_{gjk}| \le l', m \le \lambda_{\min}\left(\boldsymbol{\Theta}_g\right) \le \lambda_{\max}\left(\boldsymbol{\Theta}_g\right) \le M, g = 1, \ldots, G\},$$

and

$$\mathcal{O}_p^M = \{\boldsymbol{\varphi} = \left(\boldsymbol{\varphi}_1^\top, \ldots, \boldsymbol{\varphi}_G^\top\right)^\top \mid \max_{1 \le j, k \le p} |\sigma_{gjk}| \le l, m' \le \lambda_{min}(\boldsymbol{\Sigma}_g) \le \lambda_{max}(\boldsymbol{\Sigma}_g) \le M', g = 1, \ldots, G\}.$$

We assume the true parameter $\boldsymbol{\nu}^*$ is an interior point of $\mathcal{D}_p^M$ and the true parameter $\boldsymbol{\varphi}^*$ is restricted on $\mathcal{O}_p^M$.

We denote the density of the PLN distribution $\mathrm{PLN}(\boldsymbol{\Theta}, \boldsymbol{\mu})$ by $p(\mathbf{y}; \boldsymbol{\Theta}, \boldsymbol{\mu})$ or $p(\mathbf{y}; \boldsymbol{\Sigma}, \boldsymbol{\mu})$, and the density of the MPLN distribution $\mathrm{MPLN}(\boldsymbol{\nu}, \{\boldsymbol{\mu}_g\}_{g=1}^{G})$ by $p\left(\mathbf{y}; \boldsymbol{\nu}, \{\boldsymbol{\mu}_g\}_{g=1}^{G}\right)$ or $p\left(\mathbf{y}; \boldsymbol{\varphi}, \{\boldsymbol{\mu}_g\}_{g=1}^{G}\right)$. In the following sections, we always write $h(\mathbf{y}, \mathbf{x}) = \prod_{j=1}^{p} \exp(x_j y_j) \exp(-\exp(x_j))$. Given a single sample $i$, we write the log-likelihood function of the PLN at $\mathbf{y}_i$ as

$$\ell(\boldsymbol{\Theta}, \mathbf{y}_i) = \log p(\mathbf{y}_i; \boldsymbol{\Theta}, \boldsymbol{\mu})$$
$$= \log \int \det(\boldsymbol{\Theta})^{\frac{1}{2}} \exp\left(-\frac{1}{2}(\mathbf{x} - \boldsymbol{\mu})^\top \boldsymbol{\Theta}(\mathbf{x} - \boldsymbol{\mu})\right) h(\mathbf{y}_i, \mathbf{x}) d\mathbf{x} + C(\mathbf{y}),$$

where $C(\mathbf{y})$ is a term that depends only on $\mathbf{y}$. Also, we define $\ell_n(\boldsymbol{\Theta}) = \sum_{i=1}^{n} \ell(\boldsymbol{\Theta}, \mathbf{y}_i)$ as the log-likelihood in the PLN. For the MPLN, its log-likelihood function at $\mathbf{y}_i$ is

$$
\ell(\boldsymbol{\nu}, \mathbf{y}_i) = \log \left( \sum_{g=1}^{G} \pi_g p(\mathbf{y}_i; \boldsymbol{\Theta}_g, \boldsymbol{\mu}_g) \right)
$$

$$
= \log \left( \sum_{g=1}^{G} \pi_g \int \det(\boldsymbol{\Theta}_g)^{\frac{1}{2}} \exp \left( -\frac{1}{2}(\mathbf{x} - \boldsymbol{\mu}_g)^\top \boldsymbol{\Theta}_g (\mathbf{x} - \boldsymbol{\mu}_g) \right) h(\mathbf{y}_i, \mathbf{x}) d\mathbf{x} \right) + C(\mathbf{y}).
$$

The log-likelihood of the MPLN model is $\ell_n(\boldsymbol{\nu}) = \sum_{i=1}^{n} \ell(\boldsymbol{\nu}, \mathbf{y}_i)$. If we define

$$
L_g(\boldsymbol{\nu}_g, \mathbf{y}) = \int \det(\boldsymbol{\Theta}_g)^{\frac{1}{2}} \exp \left( -\frac{1}{2}(\mathbf{x} - \boldsymbol{\mu}_g)^\top \boldsymbol{\Theta}_g (\mathbf{x} - \boldsymbol{\mu}_g) \right) h(\mathbf{y}, \mathbf{x}) d\mathbf{x},
$$

and $L^M(\boldsymbol{\nu}, \mathbf{y}) = \sum_{g=1}^{G} \pi_g L_g(\boldsymbol{\nu}_g, \mathbf{y})$, then $\ell(\boldsymbol{\nu}, \mathbf{y}_i) = \log \left( L^M(\boldsymbol{\nu}, \mathbf{y}_i) \right) + C(\mathbf{y}_i)$. Note that the function $L_g(\boldsymbol{\nu}_g, \mathbf{y})$ is proportional to the density $p(\mathbf{y}; \boldsymbol{\Theta}_g, \boldsymbol{\mu}_g)$.

For the PLN model, denote the derivative (the score function) and the second order derivative (the Hessian matrix) of its log-likelihood as

$$
S(\boldsymbol{\Theta}, \mathbf{y}) = \frac{\partial \ell(\boldsymbol{\Theta}, \mathbf{y})}{\partial \text{vech}(\boldsymbol{\Theta})}, \mathcal{H}(\boldsymbol{\Theta}, \mathbf{y}) = \frac{\partial^2 \ell(\boldsymbol{\Theta}, \mathbf{y})}{\partial \text{vech}(\boldsymbol{\Theta}) \partial \text{vech}(\boldsymbol{\Theta})^\top}. \tag{16}
$$

For the MPLN model, we can similarly define its score function $\mathcal{S}^M(\boldsymbol{\nu}, \mathbf{y})$, its Hessian matrix $\mathcal{H}^M(\boldsymbol{\nu}, \mathbf{y})$, and its Fisher information matrix $-D(\boldsymbol{\nu}^*)$.

$$
D(\boldsymbol{\nu}) = \mathrm{E}_{\boldsymbol{\nu}^*}(\mathcal{H}^M(\boldsymbol{\nu}, \mathbf{y})). \tag{17}
$$

Finally, we denote $\mathbb{N}^p$ as the set of all $p$-dimensional non-negative integer vector. For a vector $\mathbf{a} = (a_1, \ldots, a_p)^\top$, we denote $||\mathbf{a}||_2 = \sqrt{\sum_{j=1}^{p} a_j^2}$ as its $L_2$-norm and $||\mathbf{a}||_\infty = \max_j |a_j|$ as its $L_\infty$-norm. For a matrix $\mathbf{A}$, we denote $||\mathbf{A}||_2$ as its largest singular value of $\mathbf{A}$. Given $\boldsymbol{\Theta}$ and $\boldsymbol{\mu}$, we define an operator $\mathcal{T}$ that maps functions in $\mathbf{x}$ to functions in $\mathbf{y}$,

$$
\mathcal{T}(f) = \int \exp \left( -\frac{1}{2}(\mathbf{x} - \boldsymbol{\mu})^\top \boldsymbol{\Theta}(\mathbf{x} - \boldsymbol{\mu}) \right) f(\mathbf{x}) h(\mathbf{y}, \mathbf{x}) d\mathbf{x}.
$$

In particular,

$$
\mathcal{T}(1) = \int \exp \left( -\frac{1}{2}(\mathbf{x} - \boldsymbol{\mu})^\top \boldsymbol{\Theta}(\mathbf{x} - \boldsymbol{\mu}) \right) h(\mathbf{y}, \mathbf{x}) d\mathbf{x},
$$

where 1 denotes the constant function $1(x) \equiv 1$.

**Definition A.7** (Good vector). We call a vector $\boldsymbol{\xi} = (\xi_1, \sigma_1, \ldots, \xi_G, \sigma_G)^\top \in \mathbb{R}^{2G}$ as a good vector if there exists $g$ such that $(\xi_{g'}, \sigma_{g'}) \neq (\xi_g, \sigma_g)$ for all $g' \neq g$. We call the index $g$ a good index with respect to $\boldsymbol{\xi}$.

A.3.2. LEMMAS

**Lemma A.8.** *Let* $\mathbf{y} \sim \text{PLN}(\boldsymbol{\Theta}, \boldsymbol{\mu})$. *For any* $n, y \in \mathbb{N}$, *we define*

$$
\phi(n, y) = \begin{cases} 1 & n = 0, \\ y(y-1)\cdots(y-n+1) & n > 0. \end{cases}
$$

*Then, for* $\mathbf{n} = (n_1, \cdots, n_p)^\top$, *we have*

$$
\mathrm{E}\left( \prod_{j=1}^{p} \phi(n_j, y_j) \right) = \exp \left( \mathbf{n}^\top \boldsymbol{\mu} + \mathbf{n}^\top \boldsymbol{\Theta}^{-1} \mathbf{n}/2 \right).
$$

*Proof.* By the property of conditional expectation, we have

$$
E\left(\prod_{j=1}^{p} \phi(n_j, y_j)\right) = E_{\mathbf{x}} E_{\mathbf{y}} \left(\prod_{j=1}^{p} \phi(n_j, y_j) | \mathbf{x}\right).
$$

From the moments of the Poisson distribution, we have

$$
E_{\mathbf{y}}\left(\prod_{j=1}^{p} \phi(n_j, y_j) | \mathbf{x}\right) = \prod_{j=1}^{p} \exp(n_j x_j).
$$

Further, since $\mathbf{x} \sim N(\boldsymbol{\mu}, \boldsymbol{\Theta}^{-1})$, we have

$$
E_{\mathbf{x}}\left(\prod_{j=1}^{p} \exp(n_j x_j)\right) = E_{\mathbf{x}}(\exp(\mathbf{n}^\top \mathbf{x})) = \exp\left(\mathbf{n}^\top \boldsymbol{\mu} + \mathbf{n}^\top \boldsymbol{\Theta}^{-1} \mathbf{n}/2\right),
$$

and the conclusion follows. $\qquad \square$

**Lemma A.9.** *Let $\boldsymbol{\xi} = (\xi_1, \sigma_1, \ldots, \xi_G, \sigma_G)^\top$ be a good vector with a good index $s$, satisfying $\sigma_g > 0$ for $g = 1, \cdots, G$ and $\boldsymbol{\alpha} = (\alpha_1, \cdots, \alpha_G)^\top$. If for any $z \in \mathbb{N}$*

$$
\sum_{g=1}^{G} \alpha_g \exp(\xi_g z + \sigma_g z^2) = 0,
$$

*then $\alpha_s = 0$.*

*Proof.* Without loss of generality, we assume that $(\xi_g, \sigma_g)$ $(g = 1, \cdots, G)$ are first sorted by $\sigma_g$ in increasing order, and for the same $\sigma_g$, they are further sorted by $\xi_g$ in increasing order. We say that $(\xi_g, \sigma_g)$ and $(\xi_s, \sigma_s)$ are equivalent if $(\xi_g, \sigma_g) = (\xi_s, \sigma_s)$. By this equivalence relationship, $\{(\xi_g, \sigma_g)\}_{g=1}^{G}$ can be partitioned into $Q$ groups ($Q \geq 1$). Let $S_q$ be the index set of the $q$-th group. We have

$$
\sum_{q=1}^{Q} \sum_{j \in S_q} \alpha_j \exp(\xi_j z + \sigma_j z^2) = 0
$$

for all $z \in \mathbb{N}$. Dividing $\exp(\xi_G z + \sigma_G z^2)$ on both sides of the above equation, we get

$$
\sum_{q=1}^{Q-1} \sum_{j \in S_q} \alpha_j \exp(\xi_j z + \sigma_j z^2 - \xi_G z - \sigma_G z^2) + \sum_{j \in S_Q} \alpha_j = 0 \tag{18}
$$

for all $z \in \mathbb{N}$. By the choice of $\sigma_G, \xi_G$, the first summation of Equation (18) converges to zero when $z$ goes to infinity. So, we have $\sum_{j \in S_Q} \alpha_j = 0$. By mathematical induction, we have $\sum_{j \in S_q} \alpha_j = 0$ for $q = 1, \ldots, Q$. Since $\boldsymbol{\xi}$ is a good vector with a good index $s$, $(\xi_s, \sigma_s)$ itself forms a group, and hence $\alpha_s = 0$. $\qquad \square$

**Lemma A.10.** *For any $n > 0$, let $\mathcal{M}_i \subset \mathbb{R}^p, i = 1, \cdots, n$ be $n$ linear proper subspaces of $\mathbb{R}^p$. Then, there exists a non-negative integer vector $\boldsymbol{\gamma} \in \mathbb{N}^p$ such that $\boldsymbol{\gamma} \notin \bigcup_{i=1}^{n} \mathcal{M}_i$.*

*Proof.* We prove by mathematical induction. The conclusion clearly holds for $n = 1$. Now we assume that Lemma A.10 holds for $n$ and we aim to prove that it also holds for $n + 1$. Note that by linear algebra, $\bigcup_{i=1}^{n} \mathcal{M}_i$ is a proper subspace of $\mathbb{R}^p$ for all $n$.

By induction hypothesis, we can take $\boldsymbol{\alpha} \in \mathbb{N}^p \setminus \bigcup_{i=1}^{n} \mathcal{M}_i$. If $\boldsymbol{\alpha} \notin \mathcal{M}_{n+1}$, we have $\boldsymbol{\alpha} \notin \bigcup_{i=1}^{n+1} \mathcal{M}_i$, and the proof is finished. Thus, we only need to consider $\boldsymbol{\alpha} \in \mathcal{M}_{n+1}$. Similarly, we can take $\boldsymbol{\beta} \in \mathbb{N}^p \setminus \bigcup_{i=2}^{n+1} \mathcal{M}_i$ and $\boldsymbol{\beta} \in \mathcal{M}_1$. For any $i \neq 1$, we can prove that there is at most one $k_1$ such that $\boldsymbol{\alpha} + k_1 \boldsymbol{\beta} \in \mathcal{M}_i$. In fact, if there are $k_1, k_2$ such that $k_1 \neq k_2$ and $\boldsymbol{\alpha} + k_1 \boldsymbol{\beta} \in \mathcal{M}_i, \boldsymbol{\alpha} + k_2 \boldsymbol{\beta} \in \mathcal{M}_i$, then $\boldsymbol{\beta} \in \mathcal{M}_i$, which is contradictory to the fact that $\boldsymbol{\beta} \in \mathbb{N}^p \setminus \bigcup_{i=2}^{n+1} \mathcal{M}_i$. Furthermore, there is no $k \in \mathbb{N}$ such that $\boldsymbol{\alpha} + k\boldsymbol{\beta} \in \mathcal{M}_1$. Otherwise, there exists a $k \in \mathbb{N}$ such that $\boldsymbol{\alpha} + k\boldsymbol{\beta} \in \mathcal{M}_1$. Then, we have $\boldsymbol{\alpha} \in \mathcal{M}_1$, which is also a contradiction. So we could find at most $n$ positive integers for $k$ such that $\boldsymbol{\alpha} + k\boldsymbol{\beta} \in \bigcup_{i=1}^{n+1} \mathcal{M}_i$. Since there are infinitely many non-negative numbers, we prove that there exists $k \in \mathbb{N}$ such that $\boldsymbol{\alpha} + k\boldsymbol{\beta} \notin \bigcup_{i=1}^{n+1} \mathcal{M}_i$, and Lemma A.10 is proved. $\qquad \square$

**Lemma A.11.** *Assume that* $\mathbf{A}_i \in \mathbb{R}^{p \times p}$ $(i = 1, \ldots, n)$ *are* $n$ *non-zero matrix. For any* $n > 0$, *let* $\mathcal{N}_i = \{\mathbf{x} : \mathbf{x}^\top \mathbf{A}_i \mathbf{x} = 0, \mathbf{x} \in \mathbb{R}^p\}$ $(i = 1, \ldots, n)$ *be* $n$ *proper subspaces of* $\mathbb{R}^p$. *Then, there exists a non-negative integer vector* $\boldsymbol{\gamma}$ *such that* $\boldsymbol{\gamma} \notin \bigcup_{i=1}^{n} \mathcal{N}_i$.

*Proof.* We prove by mathematical induction. The conclusion clearly holds for $n = 1$. Now we assume that Lemma A.11 holds for $n$ and we aim to prove that it also holds for $n + 1$.

By induction hypothesis, we can take $\boldsymbol{\beta} \in \mathbb{N}^p \setminus \bigcup_{i=2}^{n+1} \mathcal{N}_i$. If $\boldsymbol{\beta} \notin \mathcal{N}_1$, we have $\boldsymbol{\beta} \notin \bigcup_{i=1}^{n+1} \mathcal{N}_i$, and the proof is finished. Thus, we only need to consider $\boldsymbol{\beta} \in \mathcal{N}_1$. Next, we can take $\boldsymbol{\alpha}$ such that $\boldsymbol{\alpha} \notin \mathcal{N}_1$. If $i \neq 1$, then there are at most two integers $k \in \mathbb{N}$ satisfying the quadratic equation $(\boldsymbol{\alpha} + k\boldsymbol{\beta})^\top \mathbf{A}_i (\boldsymbol{\alpha} + k\boldsymbol{\beta}) = 0$ $(i \neq 1)$ because of $\boldsymbol{\beta}^\top \mathbf{A}_i \boldsymbol{\beta} \neq 0$. If $i = 1$, then there are at most one integers $k \in \mathbb{N}$ satisfying the quadratic equation $(\boldsymbol{\alpha} + k\boldsymbol{\beta})^\top \mathbf{A}_1 (\boldsymbol{\alpha} + k\boldsymbol{\beta}) = 0$ because of $\boldsymbol{\beta}^\top \mathbf{A}_1 \boldsymbol{\beta} = 0$ and $\boldsymbol{\alpha}^\top \mathbf{A}_1 \boldsymbol{\alpha} \neq 0$. It follows that there are at most $2n + 1$ positive integers $k \in \mathbb{N}$ such that $\boldsymbol{\alpha} + k\boldsymbol{\beta} \in \bigcup_{i=1}^{n+1} \mathcal{N}_i$. Since there are infinitely many non-negative numbers, we prove that there exists $k \in \mathbb{N}$ such that $\boldsymbol{\alpha} + k\boldsymbol{\beta} \notin \bigcup_{i=1}^{n+1} \mathcal{N}_i$, and Lemma A.11 is proved. $\square$

**Lemma A.12.** *Assume that* $\mathbf{A}_i \in \mathbb{R}^{p \times p}$ $(i = 1, \ldots, n)$ *are* $n$ *non-zero matrices. For any* $n > 0$, *let* $\mathcal{N}_i = \{\mathbf{x} : \mathbf{x}^\top \mathbf{A}_i \mathbf{x} = 0, \mathbf{x} \in \mathbb{R}^p\}$ $(i = 1, \ldots, n)$ *be* $n$ *proper subspaces of* $\mathbb{R}^p$. *For any* $m > 0$, *let* $\mathcal{M}_i \subset \mathbb{R}^p, i = 1, \cdots, m$ *be* $m$ *linear proper subspaces. Then, there exists a non-negative integer vector* $\boldsymbol{\gamma}$ *such that* $\boldsymbol{\gamma} \notin (\bigcup_{i=1}^{n} \mathcal{M}_i) \cup (\bigcup_{i=1}^{n} \mathcal{N}_i)$.

*Proof.* By the proof of Lemma A.10, there exists $\boldsymbol{\alpha} \notin \bigcup_{i=1}^{n} \mathcal{M}_i$. We can assume $\boldsymbol{\alpha} \in \bigcup_{i=1}^{n} \mathcal{N}_i$. Otherwise, we complete the proof. By the proof of Lemma A.11, there exists $\boldsymbol{\gamma} \notin \bigcup_{i=1}^{m} \mathcal{N}_i$. On the one hand, since $\boldsymbol{\gamma} \in \mathbb{N}^p \setminus \bigcup_{i=1}^{m} \mathcal{N}_i$, there are at most two integers $k \in \mathbb{N}$ satisfying the quadratic equation $(\boldsymbol{\alpha} + k\boldsymbol{\gamma})^\top \mathbf{A}_i (\boldsymbol{\alpha} + k\boldsymbol{\gamma}) = 0$. On the other hand, for any $\mathcal{M}_i, i = 1, \ldots, n$, there is at most one integer $k$ such that $\boldsymbol{\alpha} + k\boldsymbol{\beta} \in \mathcal{M}_i$. Otherwise, if there exist $k_1 \neq k_2$ and $i$ satisfying

$$\boldsymbol{\alpha} + k_1 \boldsymbol{\gamma} \in \mathcal{M}_i, \boldsymbol{\alpha} + k_2 \boldsymbol{\gamma} \in \mathcal{M}_i,$$

then we have $(k_2 - k_1)\boldsymbol{\gamma} \in \mathcal{M}_i$. It follows that $\boldsymbol{\alpha} \in \mathcal{M}_i$, which is contradictory to the fact that $\boldsymbol{\alpha} \notin \bigcup_{i=1}^{n} \mathcal{M}_i$. Hence, there are at most $2m + n$ positive integers $k \in \mathbb{N}$ such that $\boldsymbol{\alpha} + k\boldsymbol{\gamma} \in (\bigcup_{i=1}^{n} \mathcal{M}_i) \cup (\bigcup_{i=1}^{n} \mathcal{N}_i)$. Since there are infinitely many non-negative integers, there exists an integer $k$ such that $\boldsymbol{\alpha} + k\boldsymbol{\gamma} \notin (\bigcup_{i=1}^{n} \mathcal{M}_i) \cup (\bigcup_{i=1}^{m} \mathcal{N}_i)$, and Lemma A.12 is proved. $\square$

**Lemma A.13.** *Let* $\mathbf{y}$ *and* $\mathbf{x}$ *be random variables as defined in* (15) *for the PLN model, and let* $\phi(n, y)$ *be the same as in Lemma A.8. Let* $\mathbf{n} = (n_1, \cdots, n_p)^\top$ *and* $\mathbf{T} \in \mathbb{R}^{p \times p}$ *be a* $p \times p$ *matrix. We have*

$$\mathrm{E}\left( \prod_{j=1}^{p} \phi(n_j, y_j) \, \mathrm{tr}\left( \mathbf{T}\left( \boldsymbol{\Theta}^{-1} - (\mathbf{x} - \boldsymbol{\mu})(\mathbf{x} - \boldsymbol{\mu})^\top \right) \right) \right)$$
$$= \left( \mathbf{n}^\top \boldsymbol{\Theta}^{-1} \mathbf{T} \boldsymbol{\Theta}^{-1} \mathbf{n} \right) \exp\left( \mathbf{n}^\top \boldsymbol{\mu} + \mathbf{n}^\top \boldsymbol{\Theta}^{-1} \mathbf{n}/2 \right).$$

*Proof.* By Lemma A.8

$$\mathrm{E}\left( \prod_{j=1}^{p} \phi(n_j, y_j) \, \mathrm{tr}\left( \mathbf{T} \boldsymbol{\Theta}^{-1} \right) \right) = \mathrm{tr}\left( \mathbf{T} \boldsymbol{\Theta}^{-1} \right) \exp\left( \mathbf{n}^\top \boldsymbol{\mu} + \mathbf{n}^\top \boldsymbol{\Theta}^{-1} \mathbf{n}/2 \right).$$

Similar to the proof of Lemma A.8, by the moment generating function of the normal distribution, we have

$$\mathrm{E}\left( \prod_{j=1}^{p} \phi(n_j, y_j) \, \mathrm{tr}\left( \mathbf{T}(\mathbf{x} - \boldsymbol{\mu})(\mathbf{x} - \boldsymbol{\mu})^\top \right) \right)$$
$$= \mathrm{E}_{\mathbf{x}}\left( \exp(\mathbf{n}^\top \mathbf{x}) \, \mathrm{tr}\left( \mathbf{T}(\mathbf{x} - \boldsymbol{\mu})(\mathbf{x} - \boldsymbol{\mu})^\top \right) \right)$$
$$= \left\{ \mathrm{tr}\left( \mathbf{T} \boldsymbol{\Theta}^{-1} \right) + \mathbf{n}^\top \boldsymbol{\Theta}^{-1} \mathbf{T} \boldsymbol{\Theta}^{-1} \mathbf{n} \right\} \exp\left( \mathbf{n}^\top \boldsymbol{\mu} + \mathbf{n}^\top \boldsymbol{\Theta}^{-1} \mathbf{n}/2 \right).$$

Lemma A.13 follows from the above two equations.

$\square$

**Lemma A.14.** *For any $(\boldsymbol{\mu}_g^\top, \text{vech}(\boldsymbol{\Theta}_g)^\top)^\top$ $(g = 1, \cdots, G)$ that are bounded and different from each other, $p(\mathbf{y}; \boldsymbol{\Theta}_1, \boldsymbol{\mu}_1), \cdots, p(\mathbf{y}; \boldsymbol{\Theta}_G, \boldsymbol{\mu}_G)$ are linearly independent.*

*Proof.* We prove this by mathematical induction. The independence for $G = 1$ is trivial. Now we assume that Lemma A.14 holds for $G - 1$. For any $(\boldsymbol{\mu}_g^\top, \text{vech}(\boldsymbol{\Theta}_g)^\top)^\top$ $(g = 1, \cdots, G)$ that are bounded and different from each other, if we can prove that there exist $\boldsymbol{\alpha} = (\alpha_1, \ldots, \alpha_G)^\top$ and an index $s \in \{1, \ldots, G\}$ such that

$$\sum_{g=1}^{G} \alpha_g p(\mathbf{y}; \boldsymbol{\Theta}_g, \boldsymbol{\mu}_g) = 0 \text{ and } \alpha_s = 0, \tag{19}$$

then by induction, we have $\boldsymbol{\alpha} = 0$ and hence $p(\mathbf{y}; \boldsymbol{\Theta}_g, \boldsymbol{\mu}_g)$ $(g = 1, \ldots, G)$ are linearly independent. So our goal is to prove that if $\sum_{g=1}^{G} \alpha_g p(\mathbf{y}; \boldsymbol{\Theta}_g, \boldsymbol{\mu}_g) = 0$, then we can always find an index $s$ such that $\alpha_s = 0$.

Let $\mathbf{n} = (n_1, \cdots, n_p)^\top$ be any non-negative integer vector. Then, for any positive integer $z$, by Lemma A.8, there exists a polynomial function $p_z(\mathbf{y}) = \prod_{j=1}^{p} \phi(z n_j, y_j)$ such that, if (19) holds, then

$$\sum_{g=1}^{G} \alpha_g \text{E}_g(p_z(\mathbf{y})) = 0 \text{ and } \text{E}_g(p_z(\mathbf{y})) = \exp\left(z\mathbf{n}^\top \boldsymbol{\mu}_g + z^2 \mathbf{n}^\top \boldsymbol{\Theta}_g^{-1} \mathbf{n}/2\right),$$

where $\text{E}_g$ represents taking expectation with respect to $\text{PLN}(\boldsymbol{\Theta}_g, \boldsymbol{\mu}_g)$. Let

$$\boldsymbol{\xi} = \left(\mathbf{n}^\top \boldsymbol{\mu}_1, \mathbf{n}^\top \boldsymbol{\Theta}_1^{-1} \mathbf{n}/2, \ldots, \mathbf{n}^\top \boldsymbol{\mu}_G, \mathbf{n}^\top \boldsymbol{\Theta}_G^{-1} \mathbf{n}/2\right).$$

By Lemma A.9, if there exists an $\mathbf{n}$ such that $\boldsymbol{\xi}$ is a good vector with good index $s$, then $\alpha_s = 0$ and we complete the proof. If, on the other hand, $\boldsymbol{\xi} = \left(\mathbf{n}^\top \boldsymbol{\mu}_1, \mathbf{n}^\top \boldsymbol{\Theta}_1^{-1} \mathbf{n}/2, \ldots, \mathbf{n}^\top \boldsymbol{\mu}_G, \mathbf{n}^\top \boldsymbol{\Theta}_G^{-1} \mathbf{n}/2\right)$ is not a good vector for any non-negative integer vector $\mathbf{n}$. Therefore, for any $\mathbf{n}$, there exists $s \neq 1$ such that $\mathbf{n}^\top \boldsymbol{\mu}_1 = \mathbf{n}^\top \boldsymbol{\mu}_s$ and $\mathbf{n}^\top \boldsymbol{\Theta}_1^{-1} \mathbf{n} = \mathbf{n}^\top \boldsymbol{\Theta}_s^{-1} \mathbf{n}$. Thus, $\mathbf{n}$ is the solution to the linear equation $\mathbf{x}^\top (\boldsymbol{\mu}_1 - \boldsymbol{\mu}_s) = 0$ and the quadratic equation $\mathbf{x}^\top \left(\boldsymbol{\Theta}_1^{-1} - \boldsymbol{\Theta}_s^{-1}\right) \mathbf{x} = 0$. We define $\mathcal{M}_g$ as the linear space consisting of solutions to the linear equation $\mathbf{x}^\top (\boldsymbol{\mu}_1 - \boldsymbol{\mu}_g) = 0$ $(g \neq 1)$. We define $\mathcal{N}_g$ as the space consisting of solutions to the quadratic equation $\mathbf{x}^\top \left(\boldsymbol{\Theta}_1^{-1} - \boldsymbol{\Theta}_g^{-1}\right) \mathbf{x} = 0$ $(g \neq 1)$. Define $\mathcal{Q} = \bigcup_{g=2}^{G} (\mathcal{M}_g \cap \mathcal{N}_g)$. Thus, for any non-negative integer vector $\mathbf{n}$, we have $\mathbf{n} \in \mathcal{Q}$. Since $(\boldsymbol{\mu}_g^\top, \text{vech}(\boldsymbol{\Theta}_g)^\top)^\top$, $g = 1, \ldots, G$, are different from each other, $\mathcal{M}_g \cap \mathcal{N}_g$ is a proper subspace of $\mathbb{R}^p$. More precisely, if $\boldsymbol{\mu}_1 = \boldsymbol{\mu}_s$, then we have $\boldsymbol{\Theta}_1 \neq \boldsymbol{\Theta}_s$, which means $\mathcal{M}_g \cap \mathcal{N}_g = \mathcal{N}_g$. Let $\mathcal{I} = \{g : \boldsymbol{\mu}_1 = \boldsymbol{\mu}_g\}$. Then, we have

$$\mathbf{n} \in \mathcal{Q} = \bigcup_{g=2}^{G} (\mathcal{M}_g \cap \mathcal{N}_g) \subset \bigcup_{g \in \mathcal{I}} \mathcal{N}_g \cup \bigcup_{g \notin \mathcal{I}} \mathcal{M}_g.$$

This contradicts Lemma A.12. Therefore, there exists a vector $\mathbf{n}$ such that $\boldsymbol{\xi}$ is a good vector. Consequently, for any $(\boldsymbol{\mu}_g^\top, \text{vech}(\boldsymbol{\Theta}_g)^\top)^\top$ $(g = 1, \ldots, G)$ that are different from each other, $p(\mathbf{y}; \boldsymbol{\Theta}_1, \boldsymbol{\mu}_1), \ldots, p(\mathbf{y}; \boldsymbol{\Theta}_G, \boldsymbol{\mu}_G)$ are linearly independent. $\square$

**Lemma A.15.** *For any $\boldsymbol{\varphi} \in \mathcal{O}_p^M$, there exist two functions $f_{low}^M(\cdot)$ and $f_{up}^M(\cdot)$ of $\mathbf{y}$ such that $f_{low}^M(\mathbf{y}) \leq p(\mathbf{y}; \boldsymbol{\varphi}, \{\boldsymbol{\mu}_g\}_{g=1}^G) \leq f_{up}^M(\mathbf{y})$, and $\int f_{low}^M(\mathbf{y}) K(\mathbf{y}) d\mathbf{y} < \infty$, $\int f_{up}^M(\mathbf{y}) K(\mathbf{y}) d\mathbf{y} < \infty$ for any polynomial $K(\mathbf{y})$ of $\mathbf{y}$.*

*Proof.* By Remark 1 of Xiao et al. (2022). $\square$

A.3.3. PROOF OF THEOREM A.5

*Proof.* By Yakowitz & Spragins (1968), under Conditions A.3 and A.4, the identifiability of the MPLN model is equivalent to the linear independence of the PLN components. By Lemma A.14, the first conclusion holds.

To establish the second conclusion of Theorem A.5, it is necessary to derive the explicit formulas for the score functions and Fisher information matrices of the PLN and MPLN models. It is clear that the densities of the PLN and MPLN satisfy the

regularity conditions in the literature (Shao, 2003). Then, we can calculate the score function and the Fisher information as follows.

The Hessian matrix $\mathcal{H}(\boldsymbol{\Theta}, \mathbf{y})$ of the PLN model is a $p(p+1)/2 \times p(p+1)/2$ matrix. For notational convenience, let $(i, j)$ be the index of the element at position $(2p - i + 1)i/2 - p + j$ of the score function $\mathcal{S}$ and let $\mathcal{H}_{(i,j)(i',j')} = \frac{\partial^2 \ell(\boldsymbol{\Theta}, \mathbf{y})}{\partial \boldsymbol{\Theta}_{i'j'} \partial \boldsymbol{\Theta}_{ij}}$, $(i \leq j, i' \leq j')$ be the element at row $(2p - i + 1)i/2 - p + j$ and column $(2p - i' + 1)i'/2 - p + j'$ of the Hessian matrix.

Recall the Equation (16) above. The score function of the PLN model can be written as

$$\mathcal{S}(\boldsymbol{\Theta}, \mathbf{y}) = \frac{1}{2}\text{vech}_2\left(\boldsymbol{\Theta}^{-1}\right) - \frac{1}{2}\frac{\int \exp\left(-\frac{1}{2}(\mathbf{x} - \boldsymbol{\mu})^\top \boldsymbol{\Theta}(\mathbf{x} - \boldsymbol{\mu})\right) \text{vech}_2\left((\mathbf{x} - \boldsymbol{\mu})(\mathbf{x} - \boldsymbol{\mu})^\top\right) h(\mathbf{y}, \mathbf{x}) d\mathbf{x}}{\int \exp\left(-\frac{1}{2}(\mathbf{x} - \boldsymbol{\mu})^\top \boldsymbol{\Theta}(\mathbf{x} - \boldsymbol{\mu})\right) h(\mathbf{y}, \mathbf{x}) d\mathbf{x}}.$$

Especially, at the true parameter $\boldsymbol{\Theta}^*$, we have $\mathcal{S}(\boldsymbol{\Theta}^*, \mathbf{y}) = \frac{1}{2}\mathbb{E}_\mathbf{x}\left(\text{vech}_2\left(\boldsymbol{\Theta}^{*-1} - (\mathbf{x} - \boldsymbol{\mu})(\mathbf{x} - \boldsymbol{\mu})^\top\right) | \mathbf{y}\right)$. By applying the operator $\mathcal{T}$, the score function element at position $(2p - i + 1)i/2 - p + j$ can be rewritten as

$$\mathcal{S}_{(i,j)}(\boldsymbol{\Theta}, \mathbf{y}) = \frac{1}{2}\text{vech}_2\left(\boldsymbol{\Theta}^{-1}\right)_{(i,j)} - \frac{1}{2}\frac{\mathcal{T}\left(\text{vech}_2\left((\mathbf{x} - \boldsymbol{\mu})(\mathbf{x} - \boldsymbol{\mu})^\top\right)_{(i,j)}\right)}{\mathcal{T}(1)}.$$

Using the operator $\mathcal{T}$, the Hessian matrix can be written as follows. Let $\boldsymbol{\Sigma} = \boldsymbol{\Theta}^{-1}$.

When $i = j, i' = j'$,

$$\mathcal{H}_{(i,i)(i',i')}(\boldsymbol{\Theta}, \mathbf{y}) = -\frac{1}{2}\Sigma_{ii'}\Sigma_{i'i} + \frac{1}{4}\frac{\mathcal{T}\left((\mathbf{x} - \boldsymbol{\mu})_i^2(\mathbf{x} - \boldsymbol{\mu})_{i'}^2\right)}{\mathcal{T}(1)} - \frac{1}{4}\frac{\mathcal{T}\left((\mathbf{x} - \boldsymbol{\mu})_{i'}^2\right)\mathcal{T}\left((\mathbf{x} - \boldsymbol{\mu})_i^2\right)}{\mathcal{T}^2(1)}.$$

When $i \neq j, i' = j'$,

$$\mathcal{H}_{(i,j)(i',i')}(\boldsymbol{\Theta}, \mathbf{y}) = -\Sigma_{ii'}\Sigma_{i'j} + \frac{1}{2}\frac{\mathcal{T}\left((\mathbf{x} - \boldsymbol{\mu})_i(\mathbf{x} - \boldsymbol{\mu})_j(\mathbf{x} - \boldsymbol{\mu})_{i'}^2\right)}{\mathcal{T}(1)} - \frac{1}{2}\frac{\mathcal{T}\left((\mathbf{x} - \boldsymbol{\mu})_{i'}^2\right)\mathcal{T}\left((\mathbf{x} - \boldsymbol{\mu})_i(\mathbf{x} - \boldsymbol{\mu})_j\right)}{\mathcal{T}^2(1)}.$$

When $i = j, i' \neq j'$,

$$\mathcal{H}_{(i,i)(i',j')}(\boldsymbol{\Theta}, \mathbf{y}) = -\Sigma_{ii'}\Sigma_{j'i} + \frac{1}{2}\frac{\mathcal{T}\left((\mathbf{x} - \boldsymbol{\mu})_{i'}(\mathbf{x} - \boldsymbol{\mu})_{j'}(\mathbf{x} - \boldsymbol{\mu})_i^2\right)}{\mathcal{T}(1))}$$
$$- \frac{1}{2}\frac{\mathcal{T}\left((\mathbf{x} - \boldsymbol{\mu})_i^2\right)\mathcal{T}\left((\mathbf{x} - \boldsymbol{\mu})_{i'}(\mathbf{x} - \boldsymbol{\mu})_{j'}\right)}{\mathcal{T}^2(1)}.$$

When $i \neq j, i' \neq j'$,

$$\mathcal{H}_{(i,j)(i',j')}(\boldsymbol{\Theta}, \mathbf{y}) = -(\Sigma_{ii'}\Sigma_{j'j} + \Sigma_{ij'}\Sigma_{i'j}) + \frac{\mathcal{T}\left((\mathbf{x} - \boldsymbol{\mu})_{i'}(\mathbf{x} - \boldsymbol{\mu})_{j'}(\mathbf{x} - \boldsymbol{\mu})_i(\mathbf{x} - \boldsymbol{\mu})_j\right)}{\mathcal{T}(1)}$$
$$- \frac{\mathcal{T}\left((\mathbf{x} - \boldsymbol{\mu})_i(\mathbf{x} - \boldsymbol{\mu})_j\right)\mathcal{T}\left((\mathbf{x} - \boldsymbol{\mu})_{i'}(\mathbf{x} - \boldsymbol{\mu})_{j'}\right)}{\mathcal{T}^2(1)}.$$

Now we consider the score function and the Fisher information matrix of the MPLN model. The score function of the MPLN model can be written as

$$\mathcal{S}^M(\boldsymbol{\nu}, \mathbf{y}) = L^M(\boldsymbol{\nu}, \mathbf{y})^{-1}\left(\pi_1\frac{\partial L_1(\boldsymbol{\nu}_1, \mathbf{y})}{\partial \boldsymbol{\nu}_1}, \ldots, \pi_G\frac{\partial L_G(\boldsymbol{\nu}_G, \mathbf{y})}{\partial \boldsymbol{\nu}_G}\right) := \left(\mathcal{S}_1^M(\boldsymbol{\nu}, \mathbf{y}), \cdots, \mathcal{S}_G^M(\boldsymbol{\nu}, \mathbf{y})\right).$$

Similarly, we use the triplet $(g, i, j)$, with $i \leq j$, to index $\mathcal{S}^M$. Recall that $\boldsymbol{\nu}_g = \text{vech}(\boldsymbol{\Theta}_g)$. Let $(g, i, j), (g', i', j'), i \leq j, i' \leq j'$ be the index of the element at the $(g - 1)p(p + 1)/2 + (2p - i + 1)i/2 - p + j$ row and $(g' - 1)p(p + 1)/2 + (2p - i' + 1)i'/2 - p + j'$ column of $\mathcal{H}^M(\boldsymbol{\nu}, \mathbf{y})$, respectively.

When $g = g'$,

$$
\begin{aligned}
\mathcal{H}^M_{(g,i,j),(g,i',j')}(\boldsymbol{\nu}, \mathbf{y}) &= \frac{\partial}{\partial \Theta_{gi'j'}} \mathcal{S}^M_{(g,i,j)}(\boldsymbol{\nu}, \mathbf{y}) \\
&= \frac{\pi_g L_g(\boldsymbol{\Theta}_g, \mathbf{y})}{L^M(\boldsymbol{\nu}, \mathbf{y})} \left( \mathcal{H}_{(i,j)(i',j')}(\boldsymbol{\Theta}_g, \mathbf{y}) + \mathcal{S}_{(i',j')}(\boldsymbol{\Theta}_g, \mathbf{y}) \mathcal{S}_{(i,j)}(\boldsymbol{\Theta}_g, \mathbf{y}) \right) \\
&\quad - \frac{(\pi_g L_g(\boldsymbol{\Theta}_g, \mathbf{y}))^2}{L^M(\boldsymbol{\nu}, \mathbf{y})^2} \mathcal{S}_{(i',j')}(\boldsymbol{\Theta}_g, \mathbf{y}) \mathcal{S}_{(i,j)}(\boldsymbol{\Theta}_g, \mathbf{y}).
\end{aligned}
\tag{20}
$$

When $g \neq g'$, we have

$$
\begin{aligned}
\mathcal{H}^M_{(g,i,j),(g',i',j')}(\boldsymbol{\nu}, \mathbf{y}) &= \frac{\partial}{\partial \Theta_{g'i'j'}} \mathcal{S}^M_{(g,i,j)}(\boldsymbol{\nu}, \mathbf{y}) \\
&= -\frac{\pi_g \pi'_g L_g(\boldsymbol{\Theta}_g, \mathbf{y}) L_{g'}(\boldsymbol{\Theta}_{g'}, \mathbf{y})}{L^M(\boldsymbol{\nu}, \mathbf{y})^2} \mathcal{S}_{(i',j')}(\boldsymbol{\Theta}_{g'}, \mathbf{y}) \mathcal{S}_{(i,j)}(\boldsymbol{\Theta}_g, \mathbf{y}).
\end{aligned}
\tag{21}
$$

Recall the definition (17) of $D(\boldsymbol{\nu})$. Then, $-D(\boldsymbol{\nu}^*) = \mathrm{E}\left(\mathcal{S}^M(\boldsymbol{\nu}^*, \mathbf{y}) \mathcal{S}^M(\boldsymbol{\nu}^*, \mathbf{y})^\top\right)$ is the Fisher information matrix of the MPLN at $\boldsymbol{\nu}^*$.

To establish positive definiteness, we show that there exists no non-zero vector $\mathbf{t} = (\mathbf{t}_1, \cdots, \mathbf{t}_G)^\top$ such that $\mathrm{E}(\mathbf{t}^\top \mathcal{S}^M(\boldsymbol{\nu}^*, \mathbf{y}) \mathcal{S}^M(\boldsymbol{\nu}^*, \mathbf{y})^\top \mathbf{t}) = 0$. Assuming that there exists a vector $\mathbf{t} = (\mathbf{t}_1, \cdots, \mathbf{t}_G)^\top$ satisfying $\mathrm{E}(\mathbf{t}^\top \mathcal{S}^M(\boldsymbol{\nu}^*, \mathbf{y}) \mathcal{S}^M(\boldsymbol{\nu}^*, \mathbf{y})^\top \mathbf{t}) = 0$, we aim to prove that $\mathbf{t} = 0$. Since

$$
\mathrm{E}(\mathbf{t}^\top \mathcal{S}^M(\boldsymbol{\nu}^*, \mathbf{y}) \mathcal{S}^M(\boldsymbol{\nu}^*, \mathbf{y})^\top \mathbf{t}) = \sum_{\mathbf{y}} p\left(\mathbf{y}; \boldsymbol{\nu}^*, \{\boldsymbol{\mu}_g\}_{g=1}^G\right) (\mathbf{t}^\top \mathcal{S}^M(\boldsymbol{\nu}^*, \mathbf{y}))^2 = 0,
$$

and under the assumption that $p\left(\mathbf{y}; \boldsymbol{\nu}^*, \{\boldsymbol{\mu}_g\}_{g=1}^G\right) > 0$ for all $\mathbf{y}$, it follows that $\mathbf{t}^\top \mathcal{S}^M(\boldsymbol{\nu}^*, \mathbf{y}) = 0$ for any $\mathbf{y}$.

Then, when $\boldsymbol{\nu} = \boldsymbol{\nu}^*$, we have

$$
L^M(\boldsymbol{\nu}, \mathbf{y})^{-1} \sum_{g=1}^G \mathbf{t}_g^\top \pi_g \frac{\partial L_g(\boldsymbol{\nu}_g, \mathbf{y})}{\partial \boldsymbol{\nu}_g} = 0.
$$

Since $L^M(\boldsymbol{\nu}, \mathbf{y}) \neq 0$, we have $\sum_{g=1}^G \mathbf{t}_g^\top \pi_g \frac{\partial L_g(\boldsymbol{\nu}_g, \mathbf{y})}{\partial \boldsymbol{\nu}_g} = 0$. Let $\mathbf{n} = (n_1, \cdots, n_p)^\top$ be any non-negative integer vector and $\psi(\mathbf{y}) = \prod_{j=1}^p \phi(z n_j, y_j), z \in \mathbb{N}$. Then

$$
\sum_{g=1}^G \mathbf{t}_g^\top \pi_g \frac{\partial \log L_g(\boldsymbol{\nu}_g, \mathbf{y})}{\partial \boldsymbol{\nu}_g} L_g(\boldsymbol{\nu}_g, \mathbf{y}) \psi(\mathbf{y}) = 0.
\tag{22}
$$

Since $L_g(\boldsymbol{\nu}_g, \mathbf{y})$ is proportional to the density of the PLN with parameters $\boldsymbol{\Theta}_g$ and $\boldsymbol{\mu}_g$, (22) can be rewritten as

$$
\sum_{g=1}^G \mathbf{t}_g^\top \pi_g \frac{\partial \log L_g(\boldsymbol{\nu}_g, \mathbf{y})}{\partial \boldsymbol{\nu}_g} p(\mathbf{y}; \boldsymbol{\Theta}_g, \boldsymbol{\mu}_g) \psi(\mathbf{y}) = 0.
$$

Summing over $\mathbf{y}$, we get

$$
\sum_{\mathbf{y}} \sum_{g=1}^G \mathbf{t}_g^\top \pi_g \frac{\partial \log L_g(\boldsymbol{\nu}_g, \mathbf{y})}{\partial \boldsymbol{\nu}_g} p(\mathbf{y}; \boldsymbol{\Theta}_g, \boldsymbol{\mu}_g) \psi(\mathbf{y}) = 0.
$$

By Fubini 's Theorem, we get

$$
\sum_{g=1}^G \sum_{\mathbf{y}} \mathbf{t}_g^\top \pi_g \frac{\partial \log L_g(\boldsymbol{\nu}_g, \mathbf{y})}{\partial \boldsymbol{\nu}_g} p(\mathbf{y}; \boldsymbol{\Theta}_g, \boldsymbol{\mu}_g) \psi(\mathbf{y}) = 0.
\tag{23}
$$

Then, let $\mathbf{T}_g$ be the symmetric matrix such that $\mathrm{vech}(\mathbf{T}_g) = \mathbf{t}_g$. For a fixed $g$, we have

$$\sum_{\mathbf{y}} \mathbf{t}_g^{\top} \frac{\partial \log L_g(\boldsymbol{\nu}_g, \mathbf{y})}{\partial \boldsymbol{\nu}_g} p(\mathbf{y}; \boldsymbol{\Theta}_g, \boldsymbol{\mu}_g) \psi(\mathbf{y}) = \mathrm{E}_{\mathbf{y}_g} \left( \psi(\mathbf{y}_g) \mathbf{t}_g^{\top} \frac{\partial \log L_g(\boldsymbol{\nu}_g, \mathbf{y}_g)}{\partial \boldsymbol{\nu}_g} \right)$$

$$= \frac{1}{2} \mathrm{E}_{\mathbf{x}_g, \mathbf{y}_g} \left( \psi(\mathbf{y}_g) \operatorname{tr} \left\{ \mathbf{T}_g \boldsymbol{\Theta}_g^{*-1} - \mathbf{T}_g (\mathbf{x}_g - \boldsymbol{\mu})(\mathbf{x}_g - \boldsymbol{\mu})^{\top} \right\} \right),$$

where $\mathbf{y}_g$ follows the PLN distribution with parameters $\boldsymbol{\Theta}_g$ and $\boldsymbol{\mu}_g$, and $\mathbf{x}_g \sim \mathrm{N}\left(\boldsymbol{\mu}_g, \boldsymbol{\Theta}_g^{-1}\right)$ is the corresponding latent variable. By Lemma A.13, we get

$$\mathrm{E}_{\mathbf{y}_g} \left( \psi(\mathbf{y}_g) \mathbf{t}_g^{\top} \frac{\partial \log L_g(\boldsymbol{\nu}_g, \mathbf{y}_g)}{\partial \boldsymbol{\nu}_g} \right) = \frac{1}{2} z^2 \left( \mathbf{n}^{\top} \boldsymbol{\Theta}_g^{-1} \mathbf{T}_g \boldsymbol{\Theta}_g^{-1} \mathbf{n} \right) \exp\left( z \mathbf{n}^{\top} \boldsymbol{\mu}_g + z^2 \mathbf{n}^{\top} \boldsymbol{\Theta}_g^{-1} \mathbf{n}/2 \right).$$

Then, Equation (23) can be rewritten as, for all $z \in \mathbb{N}$,

$$\sum_{g=1}^{G} \pi_g z^2 \left( \mathbf{n}^{\top} \boldsymbol{\Theta}_g^{-1} \mathbf{T}_g \boldsymbol{\Theta}_g^{-1} \mathbf{n} \right) \exp\left( z \mathbf{n}^{\top} \boldsymbol{\mu}_g + z^2 \mathbf{n}^{\top} \boldsymbol{\Theta}_g^{-1} \mathbf{n}/2 \right) = 0.$$

In order to show $\mathbf{T}_1 = 0$, similar to the proof of the first conclusion, We define $\mathcal{M}_g$ as the linear space consisting of solutions to the linear equation $\mathbf{x}^{\top} \left( \boldsymbol{\mu}_1 - \boldsymbol{\mu}_g \right) = 0$ $(g \neq 1)$. We define $\mathcal{N}_g$ as the space consisting of solutions to the quadratic equation $\mathbf{x}^{\top} \left( \boldsymbol{\Theta}_1^{-1} - \boldsymbol{\Theta}_s^{-1} \right) \mathbf{x} = 0$ $(g \neq 1)$. Define $\mathcal{Q} = \bigcup_{g=2}^{G} (\mathcal{M}_g \cap \mathcal{N}_g)$. For any $\mathbf{n} \notin \mathcal{Q}$, $\left( \mathbf{n}^{\top} \boldsymbol{\mu}_1, \mathbf{n}^{\top} \boldsymbol{\Theta}_1^{-1} \mathbf{n}/2, \ldots, \mathbf{n}^{\top} \boldsymbol{\mu}_G, \mathbf{n}^{\top} \boldsymbol{\Theta}_G^{-1} \mathbf{n}/2 \right)$ is a good vector with a good index 1. Since $\pi_1 > 0$, we must have

$$\mathbf{n}^{\top} \boldsymbol{\Theta}_1^{-1} \mathbf{T}_1 \boldsymbol{\Theta}_1^{-1} \mathbf{n} = 0. \tag{24}$$

By Lemma A.12, if $\boldsymbol{\Theta}_1^{-1} \mathbf{T}_1 \boldsymbol{\Theta}_1^{-1}$ is not a zero matrix, then there exists an $\mathbf{n}$ such that $\mathbf{n} \notin \mathcal{Q}$ and $\mathbf{n}^{\top} \boldsymbol{\Theta}_1^{-1} \mathbf{T}_1 \boldsymbol{\Theta}_1^{-1} \mathbf{n} \neq 0$, which is contradictory to (24). Hence, we must have $\boldsymbol{\Theta}_1^{-1} \mathbf{T}_1 \boldsymbol{\Theta}_1^{-1} = \mathbf{0}$ and thus $\mathbf{T}_1 = \mathbf{0}$. Similarly, we get $\mathbf{T}_g = \mathbf{0}$ for all $g = 1, \ldots, G$. It follows that $\mathbf{t} = \mathbf{0}$, and we compete the proof. $\qquad\square$

### A.3.4. PROOF OF THEOREM A.6

*Proof.* For the MPLN model, the marginal distribution is also an MPLN distribution. Therefore, it is unnecessary to separately verify the one-dimensional and two-dimensional marginal density functions. Instead, it suffices to verify that the $p$-dimensional density function of the MPLN distribution satisfies the inequalities stated in Conditions 4.1 and 4.2. Specifically, there exist a measurable function $m(\mathbf{y})$ and constants $C_1$, $C_2$, such that $\int m^2(\mathbf{y}) \, d\nu = C_1^2 < \infty$, for any $\boldsymbol{\varphi}_1, \boldsymbol{\varphi}_2 \in \mathcal{O}_p^M$, let $p_1 = p\left(\mathbf{y}; \boldsymbol{\nu}_1, \{\boldsymbol{\mu}_g\}_{g=1}^{G}\right)$ and $p_2 = p\left(\mathbf{y}; \boldsymbol{\nu}_2, \{\boldsymbol{\mu}_g\}_{g=1}^{G}\right)$, we have

(i) $C_2 \|\boldsymbol{\varphi}_1 - \boldsymbol{\varphi}_2\|_2 \leq d(p_1, p_2)$.

(ii) $|p_1^{1/2} - p_2^{1/2}| \leq m(\mathbf{y}) \|\boldsymbol{\varphi}_1 - \boldsymbol{\varphi}_2\|_2$.

To establish this result, we follow a proof framework similar to that of Lemma S9 in Xiao et al. (2022). Here we denote the score function and Hessian matrix of the log-likelihood in the PLN model with respect to $\boldsymbol{\Sigma}$ as

$$\mathcal{S}(\boldsymbol{\Sigma}, \mathbf{y}) = \frac{\partial \ell(\boldsymbol{\Theta}, \mathbf{y})}{\partial \mathrm{vech}(\boldsymbol{\Sigma})}, \mathcal{H}(\boldsymbol{\Sigma}, \mathbf{y}) = \frac{\partial^2 \ell(\boldsymbol{\Theta}, \mathbf{y})}{\partial \mathrm{vech}(\boldsymbol{\Sigma}) \partial \mathrm{vech}(\boldsymbol{\Sigma})^{\top}}.$$

For the MPLN model, we can similarly denote the score function and Hessian matrix with respect to $\boldsymbol{\varphi}$ as $\mathcal{S}^M(\boldsymbol{\varphi}, \mathbf{y})$ and $\mathcal{H}^M(\boldsymbol{\varphi}, \mathbf{y})$.

First, we verify that (i) is satisfied. We get the Taylor expansion of $p_2^{1/2}$ on $\boldsymbol{\varphi}_1$,

$$p_2^{1/2} = p_1^{1/2} + p_1^{-1/2} \left( \left. \frac{\partial p\left(\mathbf{y}; \boldsymbol{\varphi}, \{\boldsymbol{\mu}_g\}_{g=1}^{G}\right)}{\partial \boldsymbol{\varphi}} \right|_{\boldsymbol{\varphi} = \boldsymbol{\varphi}_1} \right)^{\top} (\boldsymbol{\varphi}_2 - \boldsymbol{\varphi}_1)/2 + (\boldsymbol{\varphi}_2 - \boldsymbol{\varphi}_1)^{\top} R(\boldsymbol{\varphi}_0, \mathbf{y}) (\boldsymbol{\varphi}_2 - \boldsymbol{\varphi}_1)/2$$

$$= p_1^{1/2} + p_1^{1/2} \mathcal{S}^M(\boldsymbol{\varphi}_1, \mathbf{y})^{\top} (\boldsymbol{\varphi}_2 - \boldsymbol{\varphi}_1)/2 + (\boldsymbol{\varphi}_2 - \boldsymbol{\varphi}_1)^{\top} R(\boldsymbol{\varphi}_0, \mathbf{y}) (\boldsymbol{\varphi}_2 - \boldsymbol{\varphi}_1)/2, \tag{25}$$

where $R(\boldsymbol{\varphi}_0, \mathbf{y}) = p^{1/2}(\mathbf{y}; \boldsymbol{\varphi}_0, \{\boldsymbol{\mu}_g\}_{g=1}^G) \left[\mathcal{S}^M(\boldsymbol{\varphi}_0, \mathbf{y})\mathcal{S}^M(\boldsymbol{\varphi}_0, \mathbf{y})^\top/4 + \mathcal{H}^M(\boldsymbol{\varphi}_0, \mathbf{y})/2\right]$ and $\boldsymbol{\varphi}_0$ is between $\boldsymbol{\varphi}_1$ and $\boldsymbol{\varphi}_2$. We define $p_* = p\left(\mathbf{y}; \boldsymbol{\varphi}_0, \{\boldsymbol{\mu}_g\}_{g=1}^G\right)$ and $\boldsymbol{\delta} = \boldsymbol{\varphi}_2 - \boldsymbol{\varphi}_1$, then we have

$$
\begin{aligned}
d(p_1, p_2) &= \left[\int \left(p_1^{1/2} - p_2^{1/2}\right)^2 d\nu\right]^{1/2} \\
&= \left[\int \left(p_1^{1/2}\mathcal{S}^M(\boldsymbol{\varphi}_1, \mathbf{y})^\top\boldsymbol{\delta} + p_*^{1/2}\boldsymbol{\delta}^\top R(\boldsymbol{\varphi}_0, \mathbf{y})\boldsymbol{\delta}\right)^2 d\nu\right]^{1/2}/2 \\
&= \left[\int p_1\boldsymbol{\delta}^\top\mathcal{S}^M(\boldsymbol{\varphi}_1, \mathbf{y})\mathcal{S}^M(\boldsymbol{\varphi}_1, \mathbf{y})^\top\boldsymbol{\delta}d\nu + \right. \\
&\qquad \int 2p_1^{1/2}p_*^{1/2}\mathcal{S}^M(\boldsymbol{\varphi}_1, \mathbf{y})^\top\boldsymbol{\delta}\boldsymbol{\delta}^\top R(\boldsymbol{\varphi}_0, \mathbf{y})\boldsymbol{\delta}d\nu + \\
&\qquad \left.\int p_*\boldsymbol{\delta}^\top R(\boldsymbol{\varphi}_0, \mathbf{y})\boldsymbol{\delta}\boldsymbol{\delta}^\top R(\boldsymbol{\varphi}_0, \mathbf{y})\boldsymbol{\delta}d\nu\right]^{1/2}/2 \\
&=: (\mathrm{I} + \mathrm{II} + \mathrm{III})^{1/2}/2,
\end{aligned}
\tag{26}
$$

where I, II, and III are defined in an obvious way.

Define the minimum eigenvalue of $\mathrm{E}_{\boldsymbol{\varphi}_1}\left(\mathcal{S}^M(\boldsymbol{\varphi}_1, \mathbf{y})\mathcal{S}^M(\boldsymbol{\varphi}_1, \mathbf{y})^\top\right)$ is $\lambda_{min}(\boldsymbol{\varphi}_1)$, then

$$
\mathrm{I} = \boldsymbol{\delta}^\top\mathrm{E}_{\boldsymbol{\varphi}_1}\left(\mathcal{S}^M(\boldsymbol{\varphi}_1, \mathbf{y})\mathcal{S}^M(\boldsymbol{\varphi}_1, \mathbf{y})^\top\right)\boldsymbol{\delta} \geq \lambda_{min}(\boldsymbol{\varphi}_1)\|\boldsymbol{\delta}\|_2^2.
\tag{27}
$$

Since $\boldsymbol{\varphi}_1 \in \mathcal{O}_p^M$ is defined on a compact set, and $\mathrm{E}_{\boldsymbol{\varphi}_1}\left(\mathcal{S}^M(\boldsymbol{\varphi}_1, \mathbf{y})\mathcal{S}^M(\boldsymbol{\varphi}_1, \mathbf{y})^\top\right)$ is positive definite and continuous for $\boldsymbol{\varphi}_1$, there exists a positive constant $C_{low} > 0$, such that $C_{low} \leq \lambda_{min}(\boldsymbol{\varphi}_1)$ for any $\boldsymbol{\varphi}_1 \in \mathcal{O}_p^M$, then we have

$$
\mathrm{I} \geq C_{low}\|\boldsymbol{\delta}\|_2^2.
\tag{28}
$$

For part II, we have

$$
\begin{aligned}
|\mathrm{II}| &\leq \int p_1\left|\mathcal{S}^M(\boldsymbol{\varphi}_1, \mathbf{y})^\top\boldsymbol{\delta}\boldsymbol{\delta}^\top R(\boldsymbol{\varphi}_0, \mathbf{y})\boldsymbol{\delta}\right|d\nu + \int p_*\left|\mathcal{S}^M(\boldsymbol{\varphi}_1, \mathbf{y})^\top\boldsymbol{\delta}\boldsymbol{\delta}^\top R(\boldsymbol{\varphi}_0, \mathbf{y})\boldsymbol{\delta}\right|d\nu \\
&= \mathrm{E}_{\boldsymbol{\varphi}_1}\left(\left|\mathcal{S}^M(\boldsymbol{\varphi}_1, \mathbf{y})^\top\boldsymbol{\delta}\boldsymbol{\delta}^\top R(\boldsymbol{\varphi}_0, \mathbf{y})\boldsymbol{\delta}\right|\right) + \int p_*\left|\mathcal{S}^M(\boldsymbol{\varphi}_1, \mathbf{y})^\top\boldsymbol{\delta}\boldsymbol{\delta}^\top R(\boldsymbol{\varphi}_0, \mathbf{y})\boldsymbol{\delta}\right|d\nu.
\end{aligned}
\tag{29}
$$

For notational convenience, let $(i, j)$ be the index of the element at position $(2p - i + 1)i/2 - p + j$ in $\mathcal{S}$, while $(g, i, j)$ is similar index for $\mathcal{S}^M$ with $i \leq j$. Similarly, for $i \leq j, i' \leq j', (i, j)(i', j')$ refers to the element at row $(2p - i + 1)i/2 - p + j$ and column $(2p - i' + 1)i'/2 - p + j'$ in $\mathcal{H}$ and $(g, i, j), (g', i', j')$ indexes the element at row $(g - 1)p(p + 1)/2 + (2p - i + 1)i/2 - p + j$ and column $(g' - 1)p(p + 1)/2 + (2p - i' + 1)i'/2 - p + j'$ in $\mathcal{H}^M$.

We now claim that for $\mathbf{y} \sim \mathrm{MPLN}(\boldsymbol{\nu}, \{\boldsymbol{\mu}_g\}_{g=1}^G)$, there exist two polynomial functions $K_1(\mathbf{y})$ and $K_2(\mathbf{y})$ satisfying $\mathrm{E}[K_1(\mathbf{y})] < \infty$ and $\mathrm{E}[K_2(\mathbf{y})] < \infty$, such that for any $g, i, j, g', i', j', |\mathcal{S}_{(g,i,j)}^M(\boldsymbol{\varphi}, \mathbf{y})| \leq K_1(\mathbf{y})$ and $|\mathcal{H}_{(g,i,j),(g',i',j')}^M(\boldsymbol{\varphi}, \mathbf{y})| \leq K_2(\mathbf{y})$.

To prove this, first consider the case $g = g'$. Since $\pi_g L_g(\boldsymbol{\nu}, \mathbf{y})/L^M(\boldsymbol{\nu}, \mathbf{y}) \leq 1$, then we have,

$$
|\mathcal{S}_{(g,i,j)}^M(\boldsymbol{\varphi}, \mathbf{y})| \leq |\mathcal{S}_{(i,j)}(\boldsymbol{\Sigma}_g, \mathbf{y})|
$$

and

$$
|\mathcal{H}_{(g,i,j),(g,i',j')}^M(\boldsymbol{\varphi}, \mathbf{y})| \leq |\mathcal{H}_{(i,j)(i',j')}(\boldsymbol{\Sigma}_g, \mathbf{y})| + 2|\mathcal{S}_{(i,j)}(\boldsymbol{\Sigma}_g, \mathbf{y})\mathcal{S}_{(i',j')}(\boldsymbol{\Sigma}_g, \mathbf{y})|.
$$

Then, by applying Remark 3 in Xiao et al. (2022), we know that, there exist two polynomial functions $K_1(\mathbf{y}), K_2(\mathbf{y})$ such that $|\mathcal{S}_{(g,i,j)}^M(\boldsymbol{\varphi}, \mathbf{y})| \leq K_1(\mathbf{y})$ and $|\mathcal{H}_{(g,i,j),(g,i',j')}^M(\boldsymbol{\varphi}, \mathbf{y})| \leq K_2(\mathbf{y})$ with $\mathrm{E}(K_1(\mathbf{y})) < \infty$ and $\mathrm{E}(K_2(\mathbf{y})) < \infty$. The same proof can be applied to the $g \neq g'$ case. This completes the proof of the claim.

Therefore, there exist two polynomial functions $K_1(\mathbf{y})$ and $K_2(\mathbf{y})$, such that

$$\left|\mathcal{S}^M(\boldsymbol{\varphi}_1, \mathbf{y})^\top \boldsymbol{\delta}\right| \leq \left\|\mathcal{S}^M(\boldsymbol{\varphi}_1, \mathbf{y})\right\|_1 \|\boldsymbol{\delta}\|_2 \leq K_1(\mathbf{y}) \|\boldsymbol{\delta}\|_2 \tag{30}$$

and

$$\left|\boldsymbol{\delta}^\top R(\boldsymbol{\varphi}_0, \mathbf{y})\boldsymbol{\delta}\right| \leq \|R(\boldsymbol{\varphi}_0, \mathbf{y})\|_2 \|\boldsymbol{\delta}\|_2^2 \leq \|R(\boldsymbol{\varphi}_0, \mathbf{y})\|_F \|\boldsymbol{\delta}\|_2^2 \leq K_2(\mathbf{y}) \|\boldsymbol{\delta}\|_2^2. \tag{31}$$

Using the fact that any MPLN distribution have any finite moments, we can derive that the first part of Equation (29) can be bounded by $C \|\boldsymbol{\delta}\|_2^3$ with a constant $C$. Then using Lemma A.15, we know the second part of Equation (29) satisfies

$$\int p_* \left|\mathcal{S}^M(\boldsymbol{\varphi}_1, \mathbf{y})^\top \boldsymbol{\delta}\boldsymbol{\delta}^\top R(\boldsymbol{\varphi}_0, \mathbf{y})\boldsymbol{\delta}\right| d\nu \leq \int f_{up}^M(\mathbf{y}) K_1(\mathbf{y}) K_2(\mathbf{y}) \|\boldsymbol{\delta}\|_2^3 \, d\nu \leq C' \|\boldsymbol{\delta}\|_2^3$$

with a constant $C'$. Then we can derive

$$|\mathrm{II}| \leq (C + C') \|\boldsymbol{\delta}\|_2^3. \tag{32}$$

For $\mathrm{III} = \int p_* \left[\boldsymbol{\delta}^\top R(\boldsymbol{\varphi}_0, \mathbf{y})\boldsymbol{\delta}\boldsymbol{\delta}^\top R(\boldsymbol{\varphi}_0, \mathbf{y})\boldsymbol{\delta}\right] d\nu$, using (31) and similar technique in II, we have

$$|\mathrm{III}| \leq C'' \|\boldsymbol{\delta}\|_2^4 \tag{33}$$

with a constant $C''$.

Since the dominating functions $f_{up}^M, K_1, K_2$ are all independent from parameters, then the constants $C, C', C''$ are all independent from parameters. Thus, there exist positive constants $K$ and $C_2'$, both independent of the parameters, such that for any $\|\boldsymbol{\varphi}_1 - \boldsymbol{\varphi}_2\|_2 \leq K$, we have $C_2' \|\boldsymbol{\varphi}_1 - \boldsymbol{\varphi}_2\|_2 \leq d(p_1, p_2)$.

Next, we prove that (i) holds for any $\boldsymbol{\varphi}_1, \boldsymbol{\varphi}_2 \in \mathcal{O}_p^M$. Given that $\max_{1 \leq j, k \leq p} |\sigma_{jk}| \leq l$, there exists a positive constant $K'$ such that $\|\boldsymbol{\varphi}_1 - \boldsymbol{\varphi}_2\|_2 \leq K'$ for any $\boldsymbol{\varphi}_1, \boldsymbol{\varphi}_2 \in \mathcal{O}_p^M$. Define the set $\mathcal{W} = \left\{(\boldsymbol{\varphi}_1, \boldsymbol{\varphi}_2) \mid \|\boldsymbol{\varphi}_1 - \boldsymbol{\varphi}_2\|_2 \geq K, \boldsymbol{\varphi}_1, \boldsymbol{\varphi}_2 \in \mathcal{O}_p^M\right\}$. By the identifiability of the MPLN model, we have $d(p_1, p_2) \neq 0$ when $p_1 \neq p_2$. Since $d(p_1, p_2)$ is a continuous function and $\mathcal{W}$ is a closed and bounded domain, $d(p_1, p_2)$ has a minimum value $k_1 > 0$ on $\mathcal{W}$. For any $(\boldsymbol{\varphi}_1, \boldsymbol{\varphi}_2) \in \mathcal{W}$, we have $k_1 \|\boldsymbol{\varphi}_1 - \boldsymbol{\varphi}_2\|_2 / K' \leq d(p_1, p_2)$. Let $C_2 = \min\{C_2', k_1/K'\}$. Then, we have (i) holds for any $\boldsymbol{\varphi}_1, \boldsymbol{\varphi}_2 \in \mathcal{O}_p^M$.

Finally, we verify (ii). Let $p_1 = \sum_{g=1}^G \pi_g p_{1g}$ and $p_2 = \sum_{g=1}^G \pi_g p_{2g}$, where $p_{1g} = p(\mathbf{y}; \boldsymbol{\Sigma}_g, \boldsymbol{\mu}_g)$ and $p_{2i} = p(\mathbf{y}; \boldsymbol{\Sigma}_g', \boldsymbol{\mu}_g)$. According to the Lemma S10 in Xiao et al. (2022), there exist measurable functions $m_g(\mathbf{y})$ and constants $C_g$, such that $\int m_g^2(\mathbf{y})d\nu = C_g^2 < \infty$ and $\left|p_{1g}^{1/2} - p_{2g}^{1/2}\right| \leq m_g(\mathbf{y}) \|\boldsymbol{\varphi}_g - \boldsymbol{\varphi}_g'\|_2$ for $1 \leq g \leq G$. Using Cauchy-Schwarz inequality, we have

$$\begin{aligned}
\left|p_1^{1/2} - p_2^{1/2}\right|^2 &= \sum_{g=1}^G \pi_g (p_{1g} + p_{2g}) - 2\sqrt{\sum_{g=1}^G \pi_g p_{1g}} \sqrt{\sum_{g=1}^G \pi_g p_{2g}} \\
&\leq \sum_{g=1}^G \pi_g (p_{1g} + p_{2g}) - 2\sum_{g=1}^G \pi_g p_{1g}^{1/2} p_{2g}^{1/2} \\
&= \sum_{g=1}^G \pi_g \left|p_{1g}^{\frac{1}{2}} - p_2^{\frac{1}{2}}\right|^2 \\
&\leq \sum_{g=1}^G \pi_g m_g^2(\mathbf{y}) \|\boldsymbol{\varphi}_g - \boldsymbol{\varphi}_g'\|_2^2.
\end{aligned} \tag{34}$$

Let $m^2(\mathbf{y}) = \max_{1 \leq g \leq G} \pi_g m_g^2(\mathbf{y})$, then $\int m^2(\mathbf{y})d\boldsymbol{\nu} = C_1^2 < \infty$ for a constant $C_1$ and

$$\left|p_1^{1/2} - p_2^{1/2}\right|^2 \leq m^2(\mathbf{y}) \sum_{g=1}^G \|\boldsymbol{\varphi}_g - \boldsymbol{\varphi}_g'\|_2^2 = m^2(\mathbf{y}) \|\boldsymbol{\varphi} - \boldsymbol{\varphi}'\|_2^2.$$

So $\left|p_1^{1/2} - p_2^{1/2}\right| \leq m(\mathbf{y}) \|\boldsymbol{\varphi} - \boldsymbol{\varphi}'\|_2$. $\qquad\square$

## A.4. Theoretical Results and Proofs for the Binary Data Model in Example 2.6

Let $\mathbf{y} = (y_1, \ldots, y_p)^\top$ represent $p$-dimensional binary variables and $\mathbf{x} = (x_1, \ldots, x_p)^\top$ be a $p$-dimensional random vector. Assume that $\mathbf{y}$ follows the latent Gaussian copula model for binary data defined in Example 2.6, hereafter referred to as the binary model, then we have

$$
\begin{aligned}
y_j &= I\left(x_j > C_j\right) \\
\mathbf{x} &\sim \text{NPN}(\mathbf{0}, \mathbf{\Sigma}, f)
\end{aligned}
\tag{35}
$$

where $I(\cdot)$ is the indicator function and $\Sigma_{jj} = 1$ for any $1 \le j \le p$.

Let $\mathbf{C} = (C_1, \ldots, C_p)^\top$. The joint density function of $\mathbf{y} \in \{0, 1\}^p$ is given by:

$$
p(\mathbf{y}; \mathbf{\Sigma}, \mathbf{C}) = \frac{1}{(2\pi)^{p/2} \det(\mathbf{\Sigma})^{1/2}} \int_{\mathbf{x} \in U(\mathbf{y})} \exp\left(-\frac{1}{2}\mathbf{x}^\top \mathbf{\Sigma}^{-1} \mathbf{x}\right) d\mathbf{x}
$$

where $U(\mathbf{y}) = U_1(y_1) \times \cdots \times U_p(y_p)$, with:

$$
U_j(y_j) = \begin{cases} [f(C_j), \infty) & \text{if } y_j = 1 \\ (-\infty, f(C_j)) & \text{if } y_j = 0 \end{cases}
$$

for $1 \le j \le p$.

Given $\mathbf{\Sigma}$ and $\mathbf{C}$, we define an operator $\mathcal{K}$ that maps functions in $\mathbf{x}$ to functions in $\mathbf{y}$,

$$
\mathcal{K}(g) = \int_{\mathbf{x} \in U(\mathbf{y})} \exp\left(-\frac{1}{2}\mathbf{x}^\top \mathbf{\Sigma}^{-1} \mathbf{x}\right) g(\mathbf{x}) d\mathbf{x}.
$$

In particular,

$$
\mathcal{K}(1) = \int_{\mathbf{x} \in U(\mathbf{y})} \exp\left(-\frac{1}{2}\mathbf{x}^\top \mathbf{\Sigma}^{-1} \mathbf{x}\right) d\mathbf{x},
$$

where 1 denotes the constant function $1(x) \equiv 1$.

Let $\mathbf{\Theta} = \mathbf{\Sigma}^{-1}$ denote the precision matrix. Assuming that $\mathbf{C}$ is known, we establish the following theoretical results for the binary model:

**Condition A.16.** *There exist positive constants $m$ and $M$, such that $m \le \lambda_{\min}(\mathbf{\Theta}) \le \lambda_{\max}(\mathbf{\Theta}) \le M$.*

**Theorem A.17.** *Under Condition A.16, the binary model satisfies Conditions 4.1 and 4.2.*

*Proof.* For the binary model, marginal distributions of any dimension remain consistent with the binary model framework. One-dimensional marginal density functions do not depend on any parameters. Thus, we focus on the two-dimensional case, where $\mathbf{y} = (y_1, y_2)^\top$.

$$
\begin{aligned}
y_j &= I\left(x_j > C_j\right), j = 1, 2 \\
\mathbf{x} &\sim \text{NPN}(\mathbf{0}, \mathbf{\Sigma}, f),
\end{aligned}
\tag{36}
$$

where $\mathbf{x} = (x_1, x_2)$ and

$$
\mathbf{\Sigma} = \begin{pmatrix} 1 & \sigma \\ \sigma & 1 \end{pmatrix}.
\tag{37}
$$

For notational simplicity, we denote the density function as $h_2(\mathbf{y}; \sigma)$. According to the Condition A.16, we have $-1 < c \le \sigma \le C < 1$ for some constants $c$ and $C$. Define the bounded set $\mathcal{D} = \{\sigma \mid -1 < c \le \sigma \le C < 1\}$.

We only need to prove that there exist a measurable function $m(\mathbf{y})$ and constants $C_1, C_2$, such that $\int m^2(\mathbf{y}) d\nu = C_1^2 < \infty$, and for any $\sigma_1, \sigma_2 \in \mathcal{D}$, let $p_1 = h_2(\mathbf{y}; \sigma_1)$ and $p_2 = h_2(\mathbf{y}; \sigma_2)$, we have:

(i) $C_2|\sigma_1 - \sigma_2| \le d(p_1, p_2)$.

(ii) $|p_1^{1/2} - p_2^{1/2}| \leq m(\mathbf{y})|\sigma_1 - \sigma_2|.$

First, we demonstrate that (i) is satisfied, following the proof of Theorem A.6.

We denote the score function and Hessian matrix of the log-likelihood. Straightforward calculation shows that

$$
\begin{aligned}
\mathcal{S}(\sigma, \mathbf{y}) =& \frac{\log h_2(\mathbf{y};\sigma)}{\partial \sigma} \\
=& \frac{(1+\sigma^2)\mathcal{K}(x_1 x_2) - \sigma\mathcal{K}(x_1^2 + x_2^2)}{(1-\sigma^2)^2\mathcal{K}(1)} + \frac{\sigma}{1-\sigma^2}, \\
\mathcal{H}(\sigma, \mathbf{y}) =& \frac{\partial^2 \log h_2(\mathbf{y};\sigma)}{\partial \sigma^2} \\
=& \frac{(1+\sigma^2)^2\mathcal{K}(x_1^2 x_2^2) + \sigma^2\mathcal{K}((x_1^2 + x_2^2)^2) - 2\sigma(1+\sigma^2)\mathcal{K}(x_1 x_2(x_1^2 + x_2^2))}{(1-\sigma^2)^4\mathcal{K}(1)} \\
& - \frac{((1+\sigma^2)\mathcal{K}(x_1 x_2) - \sigma\mathcal{K}(x_1^2 + x_2^2))^2}{(1-\sigma^2)^4\mathcal{K}^2(1)} + \frac{1+\sigma^2}{(1-\sigma^2)^2} \\
& + \frac{(6\sigma + 2\sigma^3)\mathcal{K}(x_1 x_2) - (1+3\sigma^2)\mathcal{K}(x_1^2 + x_2^2)}{(1-\sigma^2)^3\mathcal{K}(1)}.
\end{aligned}
\tag{38}
$$

We get the Taylor expansion of $p_2^{1/2}$ on $\sigma_1$,

$$
p_2^{1/2} = p_1^{1/2} + p_1^{1/2}\mathcal{S}(\sigma_1, \mathbf{y})(\sigma_2 - \sigma_1)/2 + R(\sigma_0, \mathbf{y})(\sigma_2 - \sigma_1)^2/2,
\tag{39}
$$

where $R(\sigma_0, \mathbf{y}) = p^{1/2}(\mathbf{y};\sigma_0)\left[\mathcal{S}(\sigma_0, \mathbf{y})^2/4 + \mathcal{H}(\sigma_0, \mathbf{y})/2\right]$ and $\sigma_0$ is between $\sigma_1$ and $\sigma_2$. We define $p_0 = h_2(\mathbf{y};\sigma_0)$ and $\delta = \sigma_2 - \sigma_1$, then we have

$$
\begin{aligned}
d(p_1, p_2) =& \left[\int p_1 \mathcal{S}(\sigma_1, \mathbf{y})^2 \delta^2 d\nu + \int 2p_1^{1/2} p_0^{1/2} \mathcal{S}(\sigma_1, \mathbf{y}) R(\sigma_0, \mathbf{y}) \delta^3 d\nu + \int p_0 R(\sigma_0, \mathbf{y})^2 \delta^4 d\nu\right]^{1/2}/2 \\
=:& (I + II + III)^{1/2}/2,
\end{aligned}
\tag{40}
$$

where I, II, and III are defined in an obvious way.

Since $\sigma_1 \in \mathcal{D}$ is defined on a compact set, and $\mathrm{E}_{\sigma_1}\left(\mathcal{S}(\sigma_1, \mathbf{y})^2\right)$ is positive definite and continuous for $\sigma_1$, there exists a positive constant $C_{low} > 0$, such that for any $\sigma_1 \in \mathcal{D}$,

$$
I \geq C_{low}|\delta|^2.
\tag{41}
$$

For part II, we have

$$
\begin{aligned}
|II| \leq& \int p_1 \left|\mathcal{S}(\sigma_1, \mathbf{y}) R(\sigma_0, \mathbf{y}) \delta^3\right| d\nu + \int p_0 \left|\mathcal{S}(\sigma_1, \mathbf{y}) R(\sigma_0, \mathbf{y}) \delta^3\right| d\nu. \\
=& \mathrm{E}_{\sigma_1}\left(\left|\mathcal{S}(\sigma_1, \mathbf{y}) R(\sigma_0, \mathbf{y}) \delta^3\right|\right) + \int p_0 \left|\mathcal{S}(\sigma_1, \mathbf{y}) R(\sigma_0, \mathbf{y}) \delta^3\right| d\nu.
\end{aligned}
\tag{42}
$$

Since $\mathbf{y}$ takes a finite number of values, according to Equation (38), there exist constants $K_1$ and $K_2$, independent of the parameters, such that $|\mathcal{S}(\sigma, \mathbf{y})| \leq K_1$ and $|\mathcal{H}(\sigma, \mathbf{y})| \leq K_2$ for any $\sigma \in \mathcal{D}$. Similarly, we have $h_2(\mathbf{y};\sigma) \leq K_3$, where $K_3$ is a constant.

Then we can derive

$$
|II| \leq C'|\delta|^3
\tag{43}
$$

with a constant $C'$.

For part III $= \int p_0 R(\sigma_0, \mathbf{y})^2 \delta^4 d\nu$, using similar technique in II, we have

$$
|III| \leq C''|\delta|^4
\tag{44}
$$

with a constant $C''$.

Since the constants $K_1, K_2, C', C''$ are all independent from parameters, there exist positive constants $K$ and $C_2'$, both independent of the parameters, such that $C_2'|\sigma_1 - \sigma_2| \leq d(p_1, p_2)$ for any $|\sigma_1 - \sigma_2| \leq K$.

Next, we prove that (i) holds for any $\sigma_1, \sigma_2 \in \mathcal{D}$. Given that $-1 < c \leq \sigma \leq C < 1$, there exists a positive constant $K'$ such that $|\sigma_1 - \sigma_2| \leq K'$ for any $\sigma_1, \sigma_2 \in \mathcal{D}$. Define $\mathcal{W} = \{(\sigma_1, \sigma_2) \mid |\sigma_1 - \sigma_2| \geq K, \sigma_1, \sigma_2 \in \mathcal{D}\}$. By the identifiability of the binary model, $d(p_1, p_2) \neq 0$ when $p_1 \neq p_2$. Since $d(p_1, p_2)$ is continuous and $\mathcal{W}$ is compact, it attains a minimum value $k_1 > 0$ on $\mathcal{W}$. For any $(\sigma_1, \sigma_2) \in \mathcal{W}$, $k_1|\sigma_1 - \sigma_2|/K' \leq d(p_1, p_2)$. Let $C_2 = \min\{C_2', k_1/K'\}$. Thus, (i) holds for all $\sigma_1, \sigma_2 \in \mathcal{D}$.

Finally, we verify (ii). For any $\sigma_1, \sigma_2 \in \mathcal{D}$, let $p_1 = h_2(\mathbf{y}; \sigma_1)$ and $p_2 = h_2(\mathbf{y}; \sigma_2)$. Similarly, we have Taylor expansion of $p_2^{1/2}$ on $\sigma_1$,

$$p_2^{1/2} = p_1^{1/2} + p_*^{-1/2}\left(\frac{\partial p_*}{\partial \sigma^*}\right)(\sigma_2 - \sigma_1)/2$$

where $\sigma^*$ is between $\sigma_1$ and $\sigma_2$, and $p_* = h_2(\mathbf{y}; \sigma^*)$. Let $\eta = \sigma_2 - \sigma_1$, we have,

$$\left|p_2^{1/2} - p_1^{1/2}\right| = \frac{1}{2}p_*^{1/2}|\mathcal{S}(\sigma^*, \mathbf{y})\eta| \leq \frac{1}{2}K_3^{1/2}K_1|\eta|$$

So we finish the proof.

$\square$

# B. Simulation

### B.1. Details of the Data Generation Process for the MPLN Model

The network structures are generated based on the following procedures.

- Random Graph: Pairs of nodes are connected with probability 0.016 and the nonzero edges are randomly set as 0.3 or -0.3.

- Blocked Random Graph: The blocked random graph comprises 5 independent groups of randomly connected nodes. Pairs of nodes within the same group are connected with probability 0.016, while nodes belonging to separate groups are not related. The nonzero edges are randomly set as 0.3 or -0.3.

- Banded Graph: Pairs $(i, j)$ of nodes are connected if $|i - j| \leq 2, i = j$. All nonzero edges are set as 0.3.

- Scale-free Graph: A scale-free network follows a power law which suggests that the central node have more connections. The scale-free graphs are generated with power 1 and the nonzero edges are randomly set as 0.3 or -0.3.

For each simulation dataset, we independently generate the precision matrix for each of the three latent normal distributions based on one of four graph structures. To ensure positive definiteness, the diagonal elements of the precision matrix are set to 1 plus a small positive value. The mean vectors $\boldsymbol{\mu}_g = (\mu_{g1}, \ldots, \mu_{gp})^\top$ for each latent normal distribution ($g = 1, 2, 3$) are then generated as follows: the first $p_d$ elements of $\boldsymbol{\mu}_g$ are independently sampled from $\{v_1, (v_1 + v_2)/2, v_2\}$, and the remaining $p - p_d$ elements are shared across $\boldsymbol{\mu}_1, \boldsymbol{\mu}_2, \boldsymbol{\mu}_3$, independently sampled from $\{v_3, v_4\}$. The values of $(v_1, v_2, v_3, v_4)$ are set to $(1.9, -0.6, 0.4, -0.6)$ for the low zero-proportion case (about 20% zeros) and $(1.1, -1.4, -0.4, -1.4)$ for the high zero-proportion case (about 40% zeros). We vary $p_d$ to control the mixing degree of the three populations.

The scaling factors $\mathbf{S} = (S_1, \ldots, S_n)^\top$ are independently sampled from a distribution defined as $S_i = \exp(Z_i)$, where $Z_i \sim N(\log(\log 10), 0.05^2)$ for $i = 1, \ldots, n$. Using these model parameters, the observed expression values $\mathbf{Y}_1, \ldots, \mathbf{Y}_n$ are generated from the MPLN model. We then compute the Adjusted Rand Index (ARI) between the true population labels and the labels obtained from K-means clustering of the normalized data

$$\widetilde{\mathbf{Y}}_i = \log(\frac{\mathbf{Y}_i + 1}{\widehat{S}_i}),$$

*Table 3.* Comparisons of EM-MMLE with PLNet, VMPLN and Glasso in terms of AUPR on simulation results for blocked random graphs generated by the MPLN model. The results are averages over 50 replicates with standard deviations in brackets.

| Zero-proportion | Low | | | | | |
|---|---|---|---|---|---|---|
| Dimension | $p = 100$ | | | $p = 300$ | | |
| Mixing degree | Low | Middle | High | Low | Middle | High |
| | | | $n = 1800$ | | | |
| EM-MMLE | **0.95 (0.011)** | **0.93 (0.017)** | **0.9 (0.016)** | **0.82 (0.028)** | **0.78 (0.023)** | **0.67 (0.043)** |
| PLNet | 0.86 (0.05) | 0.79 (0.084) | 0.79 (0.074) | 0.7 (0.033) | 0.65 (0.056) | 0.52 (0.054) |
| VMPLN | 0.91 (0.028) | 0.88 (0.037) | 0.89 (0.024) | 0.62 (0.024) | 0.62 (0.019) | 0.6 (0.025) |
| Glasso | 0.82 (0.038) | 0.78 (0.038) | 0.75 (0.05) | 0.66 (0.032) | 0.64 (0.027) | 0.56 (0.037) |
| | | | $n = 3000$ | | | |
| EM-MMLE | **0.99 (0.005)** | **0.98 (0.005)** | **0.98 (0.006)** | **0.94 (0.014)** | **0.91 (0.013)** | **0.84 (0.047)** |
| PLNet | 0.96 (0.02) | 0.95 (0.041) | 0.91 (0.057) | 0.87 (0.062) | 0.83 (0.048) | 0.73 (0.068) |
| VMPLN | 0.95 (0.03) | 0.93 (0.036) | 0.94 (0.021) | 0.7 (0.021) | 0.7 (0.02) | 0.7 (0.024) |
| Glasso | 0.89 (0.037) | 0.87 (0.043) | 0.85 (0.041) | 0.75 (0.03) | 0.74 (0.021) | 0.68 (0.039) |
| Zero-proportion | High | | | | | |
| Dimension | $p = 100$ | | | $p = 300$ | | |
| Mixing degree | Low | Middle | High | Low | Middle | High |
| | | | $n = 1800$ | | | |
| EM-MMLE | **0.8 (0.029)** | **0.78 (0.028)** | **0.74 (0.035)** | **0.58 (0.048)** | **0.53 (0.048)** | **0.45 (0.045)** |
| PLNet | 0.7 (0.064) | 0.67 (0.049) | 0.63 (0.045) | 0.49 (0.054) | 0.44 (0.049) | 0.35 (0.055) |
| VMPLN | 0.76 (0.054) | 0.74 (0.034) | 0.71 (0.036) | 0.43 (0.034) | 0.42 (0.031) | 0.41 (0.036) |
| Glasso | 0.52 (0.045) | 0.47 (0.036) | 0.45 (0.048) | 0.44 (0.037) | 0.42 (0.031) | 0.36 (0.038) |
| | | | $n = 3000$ | | | |
| EM-MMLE | **0.95 (0.012)** | **0.93 (0.014)** | **0.91 (0.018)** | **0.79 (0.037)** | **0.75 (0.037)** | **0.68 (0.039)** |
| PLNet | 0.87 (0.061) | 0.86 (0.034) | 0.8 (0.062) | 0.73 (0.057) | 0.67 (0.057) | 0.6 (0.047) |
| VMPLN | 0.86 (0.048) | 0.84 (0.043) | 0.83 (0.046) | 0.53 (0.028) | 0.53 (0.023) | 0.51 (0.019) |
| Glasso | 0.62 (0.058) | 0.59 (0.041) | 0.54 (0.067) | 0.55 (0.034) | 0.54 (0.028) | 0.49 (0.029) |

where $\widehat{S}_i = \sum_{j=1}^p Y_{ij}/10^3$ for $i = 1, 2, \ldots, n$. The ARI values are adjusted as follows: low-level mixing data correspond to an ARI in $(0.9, 1]$, middle-level mixing data to an ARI in $(0.75, 0.85]$, and high-level mixing data to an ARI in $(0.6, 0.7]$.

## B.2. Details of the Data Generation Process for Binary Data

The threshold parameter $C_j$ is sampled from uniform distribution on $[-1, 1]$. We construct the inverse correlation matrix $\Theta$ such that $\Theta_{jj} = 1$ and $\Theta_{jk} = \alpha_1 z_{jk}$ for $j \neq k$. Here, we set $\alpha_1 = 0.15$ to ensure the positive definiteness of $\Theta$, and $z_{jk}$ is a Bernoulli random variable with success probability

$$p_{jk} = \frac{400}{p(p-1)} \exp\left(-\frac{\|\mathbf{t}_j - \mathbf{t}_k\|_2}{2\alpha_2}\right),$$

where $\mathbf{t}_j$ and $\mathbf{t}_k$ are independently drawn from a bivariate uniform distribution on $[0, 1]$. The constant $\alpha_2$ is adjusted to generate approximately 200 edges in the graph, and the covariance matrix $\Sigma$ is rescaled so that all diagonal elements equal 1.

## B.3. Additional Results

Tables 3–5 present the AUPR performance of EM-MMLE, PLNet, VMPLN, and Glasso for blocked random graphs, banded graphs, and scale-free graphs, respectively.

## C. Real Data Analysis

### C.1. Silver Standard Construction for Benchmarking on scRNA-seq Data

The databases utilized in this study include STRING (Szklarczyk et al., 2019), HumanTFDB (Hu et al., 2019), hTFtarget (Zhang et al., 2020), ChEA (Lachmann et al., 2010), ChIP-Atlas (Oki et al., 2018), ChIPBase (Zhou et al., 2016), ESCAPE (Xu et al., 2013), TRRUST (Han et al., 2018), and RegNetwork (Liu et al., 2015).

*Table 4.* Comparisons of EM-MMLE with PLNet, VMPLN and Glasso in terms of AUPR on simulation results for banded graphs generated by the MPLN model. The results are averages over 50 replicates with standard deviations in brackets.

| Zero-proportion | | Low | | | | | |
|---|---|---|---|---|---|---|
| Dimension | | $p = 100$ | | | | $p = 300$ | |
| Mixing degree | Low | Middle | High | Low | Middle | High |
| | | | $n = 1800$ | | | |
| EM-MMLE | **0.87 (0.017)** | **0.88 (0.017)** | **0.85 (0.022)** | **0.87 (0.01)** | **0.83 (0.017)** | **0.76 (0.021)** |
| PLNet | 0.8 (0.019) | 0.77 (0.03) | 0.72 (0.017) | 0.73 (0.027) | 0.67 (0.052) | 0.6 (0.031) |
| VMPLN | 0.71 (0.02) | 0.71 (0.021) | 0.71 (0.019) | 0.71 (0.012) | 0.71 (0.014) | 0.69 (0.016) |
| Glasso | 0.65 (0.037) | 0.63 (0.034) | 0.6 (0.03) | 0.72 (0.018) | 0.7 (0.018) | 0.65 (0.021) |
| | | | $n = 3000$ | | | |
| EM-MMLE | **0.93 (0.015)** | **0.93 (0.016)** | **0.93 (0.017)** | **0.95 (0.008)** | **0.93 (0.01)** | **0.88 (0.019)** |
| PLNet | 0.88 (0.014) | 0.85 (0.023) | 0.81 (0.03) | 0.85 (0.037) | 0.82 (0.025) | 0.75 (0.052) |
| VMPLN | 0.76 (0.024) | 0.75 (0.024) | 0.76 (0.025) | 0.77 (0.014) | 0.77 (0.015) | 0.77 (0.012) |
| Glasso | 0.69 (0.028) | 0.67 (0.034) | 0.66 (0.044) | 0.79 (0.018) | 0.78 (0.019) | 0.74 (0.02) |
| Zero-proportion | | High | | | | | |
| Dimension | | $p = 100$ | | | | $p = 300$ | |
| Mixing degree | Low | Middle | High | Low | Middle | High |
| | | | $n = 1800$ | | | |
| EM-MMLE | **0.79 (0.019)** | **0.79 (0.023)** | **0.77 (0.025)** | **0.74 (0.015)** | **0.7 (0.018)** | **0.64 (0.022)** |
| PLNet | 0.72 (0.028) | 0.69 (0.031) | 0.65 (0.035) | 0.62 (0.033) | 0.59 (0.031) | 0.53 (0.03) |
| VMPLN | 0.62 (0.021) | 0.61 (0.034) | 0.61 (0.028) | 0.59 (0.019) | 0.57 (0.017) | 0.57 (0.021) |
| Glasso | 0.5 (0.023) | 0.48 (0.035) | 0.46 (0.037) | 0.55 (0.015) | 0.54 (0.019) | 0.5 (0.025) |
| | | | $n = 3000$ | | | |
| EM-MMLE | **0.88 (0.012)** | **0.89 (0.022)** | **0.87 (0.019)** | **0.87 (0.01)** | **0.84 (0.015)** | **0.79 (0.015)** |
| PLNet | 0.81 (0.023) | 0.79 (0.033) | 0.75 (0.03) | 0.79 (0.034) | 0.74 (0.039) | 0.69 (0.036) |
| VMPLN | 0.67 (0.025) | 0.67 (0.036) | 0.66 (0.029) | 0.66 (0.015) | 0.65 (0.019) | 0.65 (0.017) |
| Glasso | 0.53 (0.028) | 0.52 (0.042) | 0.49 (0.037) | 0.64 (0.018) | 0.62 (0.024) | 0.59 (0.024) |

*Table 5.* Comparisons of EM-MMLE with PLNet, VMPLN and Glasso in terms of AUPR on simulation results for scale-free graphs generated by the MPLN model. The results are averages over 50 replicates with standard deviations in brackets.

| Zero-proportion | | Low | | | | | |
|---|---|---|---|---|---|---|
| Dimension | | $p = 100$ | | | | $p = 300$ | |
| Mixing degree | Low | Middle | High | Low | Middle | High |
| | | | $n = 1800$ | | | |
| EM-MMLE | **0.74 (0.036)** | **0.71 (0.035)** | **0.64 (0.05)** | **0.64 (0.034)** | **0.58 (0.04)** | **0.5 (0.04)** |
| PLNet | 0.68 (0.044) | 0.6 (0.042) | 0.54 (0.043) | 0.57 (0.049) | 0.49 (0.046) | 0.43 (0.045) |
| VMPLN | 0.54 (0.025) | 0.53 (0.03) | 0.52 (0.029) | 0.48 (0.028) | 0.48 (0.029) | 0.43 (0.042) |
| Glasso | 0.52 (0.038) | 0.48 (0.037) | 0.44 (0.04) | 0.53 (0.035) | 0.49 (0.037) | 0.42 (0.033) |
| | | | $n = 3000$ | | | |
| EM-MMLE | **0.84 (0.029)** | **0.85 (0.035)** | **0.78 (0.045)** | **0.83 (0.025)** | **0.78 (0.031)** | **0.68 (0.033)** |
| PLNet | 0.78 (0.045) | 0.75 (0.053) | 0.67 (0.042) | 0.78 (0.042) | 0.71 (0.05) | 0.63 (0.04) |
| VMPLN | 0.59 (0.027) | 0.59 (0.025) | 0.57 (0.034) | 0.59 (0.021) | 0.58 (0.018) | 0.56 (0.023) |
| Glasso | 0.56 (0.044) | 0.56 (0.037) | 0.5 (0.038) | 0.66 (0.031) | 0.62 (0.03) | 0.54 (0.03) |
| Zero-proportion | | High | | | | | |
| Dimension | | $p = 100$ | | | | $p = 300$ | |
| Mixing degree | Low | Middle | High | Low | Middle | High |
| | | | $n = 1800$ | | | |
| EM-MMLE | **0.56 (0.037)** | **0.55 (0.033)** | **0.49 (0.047)** | **0.37 (0.039)** | **0.32 (0.04)** | **0.28 (0.043)** |
| PLNet | 0.5 (0.04) | 0.47 (0.035) | 0.41 (0.032) | 0.33 (0.04) | 0.29 (0.039) | 0.25 (0.043) |
| VMPLN | 0.42 (0.026) | 0.42 (0.025) | 0.4 (0.031) | 0.3 (0.026) | 0.28 (0.029) | 0.27 (0.036) |
| Glasso | 0.36 (0.032) | 0.35 (0.033) | 0.31 (0.031) | 0.31 (0.032) | 0.29 (0.034) | 0.25 (0.035) |
| | | | $n = 3000$ | | | |
| EM-MMLE | **0.7 (0.032)** | **0.7 (0.031)** | **0.65 (0.052)** | **0.58 (0.046)** | **0.53 (0.05)** | **0.47 (0.073)** |
| PLNet | 0.63 (0.035) | 0.61 (0.034) | 0.53 (0.043) | 0.54 (0.042) | 0.49 (0.047) | 0.42 (0.067) |
| VMPLN | 0.46 (0.02) | 0.46 (0.023) | 0.44 (0.02) | 0.39 (0.025) | 0.38 (0.028) | 0.37 (0.043) |
| Glasso | 0.4 (0.038) | 0.39 (0.038) | 0.35 (0.032) | 0.42 (0.034) | 0.41 (0.037) | 0.35 (0.056) |

Silver standards are derived from the 3' batch data, with gene pairs from public gene regulatory network databases considered as potential regulatory relationships. Each identified regulatory relationship involves at least one transcription factor. For each cell type in the 3' batch, Spearman's $\rho$ correlation is calculated between gene pairs with potential regulatory connections. Gene pairs showing significant Spearman's $\rho$ correlations are designated as true regulatory relationships and included in the silver standard edge set for the respective cell type.

## C.2. Description of the AUPRC Ratio

First, given a network estimation $\widehat{\Theta}$ from an algorithm, we define an edge score for each edge. For EM-MMLE, VMPLN, PLNet, and Glasso, the edge score for the edge $(i, j)$ is defined as:

$$\left| -\left(\widehat{\Theta}_{ii}\widehat{\Theta}_{jj}\right)^{-\frac{1}{2}}\widehat{\Theta}_{ij}\right|.$$

For PPCOR and GENIE3, the edge score is defined as the estimated connected weight for each edge.

Since the network inferred by the method contains connected edges with varying scores and unconnected edges, we calculate the area under the partial precision-recall curve (AUPRC) by applying different thresholds to the edge scores. To mitigate the impact of varying network densities across methods, we define the AUPRC ratio as the ratio between the AUPRC of a given method and the expected AUPRC of a random network prediction. The precision of the random predictor is the edge density of the ground-truth network.

## C.3. Supporting Figures

Figure 3 shows the gene regulatory networks inferred by EM-MMLE for the top 300 highly variable genes. ID3 is identified as a cell-type-specific hub gene in the CD4+ T cell network. Figure 4 presents the top 10 biological processes from a gene ontology analysis of ID3 target genes, highlighting the most significant biological processes associated with these genes.

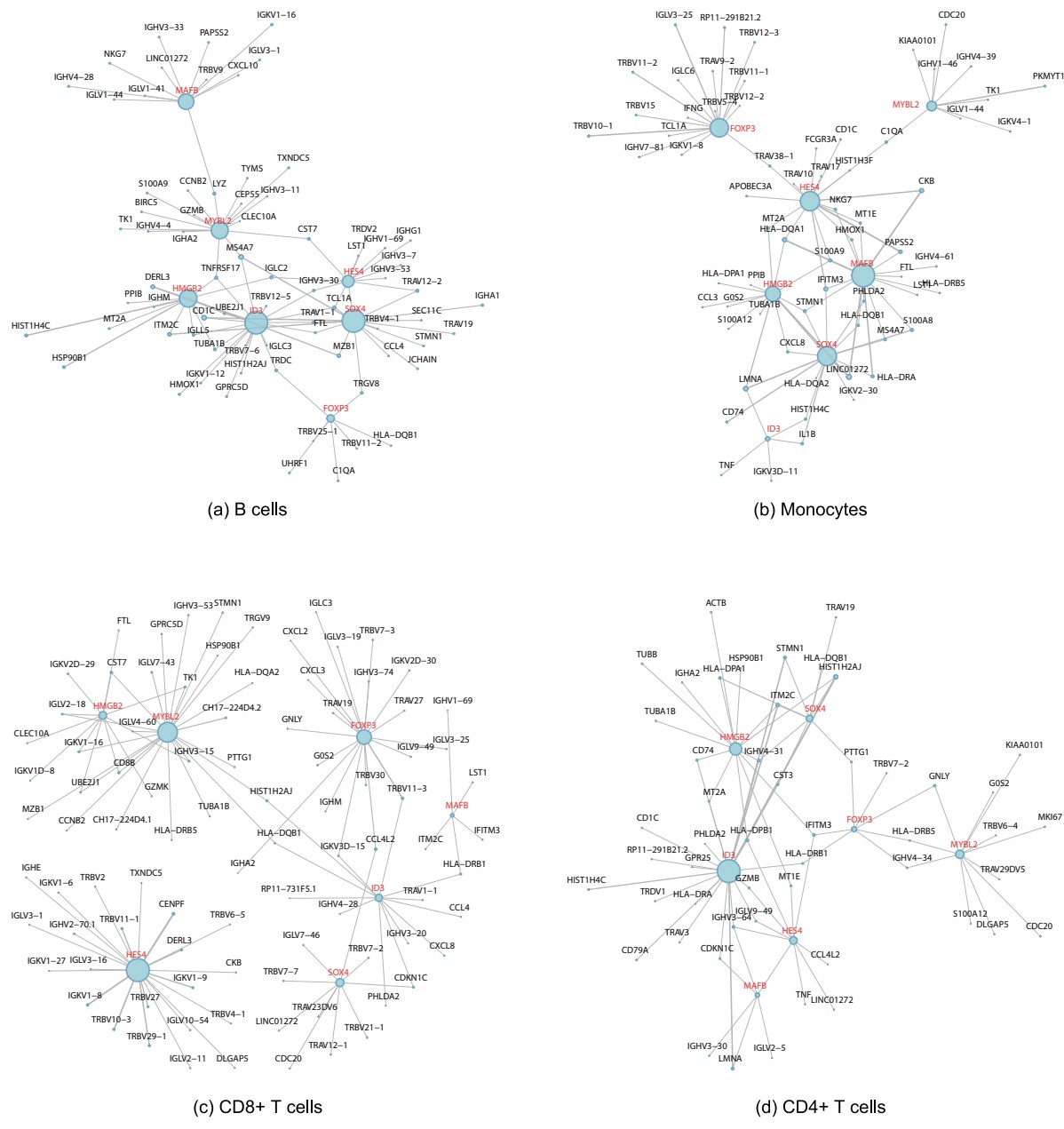

*Figure 3.* Inferred gene regulatory networks for four cell types. The size of each node represents its weighted node degree, while the edge width indicates the correlation weight. Genes highlighted in red represent the transcription factors of interest.

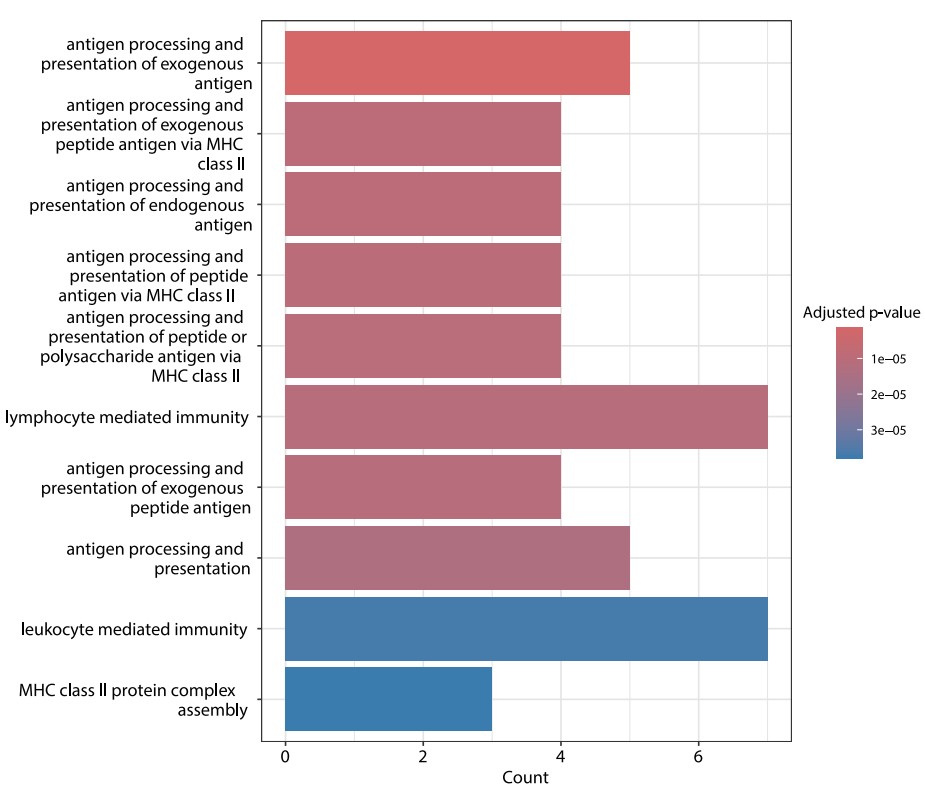

*Figure 4.* Gene ontology enrichment analysis of ID3 target genes within the gene regulatory network of CD4+ T cells.

