# OpenReview forum: "A Generic Family of Graphical Models: Diversity, Efficiency, and Heterogeneity"
_ICML.cc/2025/Conference — ICML 2025 poster_

### Official Review · Reviewer_oABz · 2025-02-16

**Overall Recommendation:** 3

**Summary:**

To infer high dimensional graphical models with a variety of data types, this paper introduces a marginally recoverable parametric family. The family is very flexible, and it includes Gaussian, PLN, and latent Gaussian copula for count and binary data. Within this family, the joint distribution can be characterized by all pairwise marginal distributions, and hence the dimensionality is reduced significantly. To further capture heterogeneous structures, they extend the method to mixture models. The proposed methods are evaluated via simulations and single-cell RNA sequencing data.

**Claims And Evidence:**

Yes, the claims are supported by clear and convincing evidence.

**Essential References Not Discussed:**

NA

**Experimental Designs Or Analyses:**

Yes. All the simulations and application to single-cell RNA data is solid. For application in singel-cell RNA data, they further provide biological perspective with supporting biological literature, which make the analyses more meaningful.

**Methods And Evaluation Criteria:**

Yes, the author study their methods by both simulations and single-cell RNA data. They evaluate the model performance using diverse criteria for different data types, e.g., AUPR for count data and 3 estimation methods for binary data. All these methods and criteria are standard and valid.

**Other Comments Or Suggestions:**

Figure 1 is blank.

**Other Strengths And Weaknesses:**

Strength: the writing is clear and easy to follow. The application part provides biological insights.

Weakness: The major contribution for this paper is proposing a new marginally recoverable family. The author(s) write the family in a very general way, but the examples are more or less all Gaussian related. Therefore, there are some concerns about novolties, as the defined family may be some extensions of Gaussian and we can directly use results. For mixture part, overlaiding the MMLE with EM is standard.

**Questions For Authors:**

NA

**Relation To Broader Scientific Literature:**

The family defined in this paper is general, which contains many widely-used distributions. Therefore, we can use the technique proposed here to models (not limited to graphical models) within this family, to deal with high dimensionality issue.

**Theoretical Claims:**

No, I didn't, since these are well-established statistical methods and I believe the authors did it correctly.

---

> ### Author Rebuttal · Authors · 2025-03-31
>
> We sincerely appreciate your time in reviewing our paper and your insightful comments. In the following response, we would like to address your major concern and provide additional clarification.
>
> For weakness:
>
> > The author(s) write the family in a very general way, but the examples are more or less all Gaussian related. Therefore, there are some concerns about novolties, as the defined family may be some extensions of Gaussian and we can directly use results.
>
> Thanks for your comment. First, since the focus of this work is primarily on graphical network inference, with the application scenario being gene regulatory network inference using biological genomics data, the examples provided here are mainly based on Gaussian distribution models in hierarchical settings. However, the new concept introduced—the marginally recoverable family—includes distributions beyond just Gaussian-related distributions. For instance, the multinomial distribution belongs to this family but does not Gaussian-related. Let $\mathbf{X}=\left(X_{1}, \ldots, X_{p}\right)^\top \sim Multinomial(n,\boldsymbol{\theta})$, where $\boldsymbol{\theta}=(\theta_1,\dots,\theta_p)^\top$. Then, for $1 \leq j < k \leq p $, we have $X_j \sim Multinomial(n, \theta_{j}) $ and $\left(X_j , X_k \right)^\top \sim Multinomial(n, (\theta_j, \theta_k)^\top) $. This demonstrates that the multinomial distribution satisfies the definition of the family.
>
> Secondly, our defined family is not an extension of the Gaussian distribution that can be addressed by existing methods. Even when the latent layer is modeled using a Gaussian distribution, distributions with hierarchical structures still pose significant challenges for parameter estimation. For example, as mentioned in the paper, there are few methods developed for estimating the precision matrix of the MPLN model in high-dimensional settings. Compared to Gaussian mixture models, the MPLN model presents a greater challenge during the parameter estimation process using the EM algorithm, as its expected complete log-likelihood involves high-dimensional integrals that are difficult to compute. Therefore, for such models, we cannot directly apply existing methods or results.
>
> For other comments or suggestions:
>
> > Figure 1 is blank.
>
> Thank you for your comment. We appreciate your attention to Figure 1. We would like to clarify that Figure 1 is not blank. It is possible that the white areas in the figure were confused with the background, which might have led to the impression that the figure is empty. To avoid this misunderstanding, we will add borders around each subplot in Figure 1 in the final version. We would like to clarify that in subplot (a), the white areas represent positions where the true network does not have edges. In subplots (b) to (e), the white areas at position $(i,j)$ indicate that the frequency of false positives (i.e., incorrectly predicting an edge where there is none) for that position is zero. We hope this explanation clarifies the meaning of the white areas.

---

### Official Review · Reviewer_HnXb · 2025-03-09

**Overall Recommendation:** 4

**Summary:**

The paper introduces a novel family of graphical models, termed the marginally recoverable parametric family, to address diversity, efficiency, and heterogeneity in high-dimensional network inference. The paper proposes a Maximum Marginal Likelihood Estimator (MMLE) for efficient parameter estimation and extend it to EM-MMLE to handle heterogeneous membership via mixture modelling. The approach is demonstrated through theoretical consistency, simulations, and application to real single-cell RNA sequencing data.

**Claims And Evidence:**

Most of the claims are theoretical statements and are verified via proofs.

**Essential References Not Discussed:**

Not sure.

**Experimental Designs Or Analyses:**

The simulation studies are well-structured, covering different aspects of experiment settings and data heterogeneity. The real-world application (scRNA-seq) is relevant and well-motivated.

**Methods And Evaluation Criteria:**

The proposed marginally recoverable parametric family is well-motivated and mathematically rigorous. It is evaluated via theoretical consistency, simulation benchmarks (e.g. AUPRC) and application to real RNA-seq datasets.

**Other Comments Or Suggestions:**

- Notation $d$ is used for both dimension and Hellinger distance in Condition 4.1 and 4.2.
- It is very hard to interpret the sample size requirement in Theorem 4.4. Some discussion of sample complexity depending on the dimension / sparsity wil be appreciated.
- One motivation of this paper is for mixed type data analysis. It would be helpful to provide an example with both continuous and discrete variables with explicit modelling formula.

**Other Strengths And Weaknesses:**

**Strength**:
- This paper introduces a flexble class of graphical models with rigorous mathematical justification.
- The advantage of this class is to avoid high-dimensional integration via marginal likelihood estimation.
- The experiments show the proposal outperforms existing methods on benchmark datasets.

**Weakness**:
- Marginal recoverability may not extend to distributions with higher moments.

**Questions For Authors:**

- How do we justify Condition 4.1 and 4.2 intuitively? I can find parametric examples in Line 275-278.
- Is there any theoretical guarantee for the consistency of membership assignment in the mixture modelling?

**Relation To Broader Scientific Literature:**

Graphical modelling for network analysis is widely used in scientific research, especially in genetics and epidemiology. Developing flexible and consistent methods is significant for the exploratory analysis.

**Theoretical Claims:**

The consistency proofs for MMLE (Theorem 4.3) and EM-MMLE (Theorem 4.4) appear mathematically sound. The proofs rely on Conditions 4.1 and 4.2, which ensure proper convergence.

---

> ### Author Rebuttal · Authors · 2025-03-31
>
> We sincerely appreciate your time in reviewing our paper and your positive feedback. Here we address your comments as follows.
>
> For weakness:
> >Marginal recoverability may not extend to distributions with higher moments.
>
> Thanks for your comment. To facilitate an intuitive network representation, we impose certain restrictions on the parameters of the family, typically related to the first and second moments. However, motivated by the core ideas behind this family, we can provide a more general definition that extends to distributions determined by higher moments. Specifically, we introduce the definition of a $d$-marginally recoverable family. Let $\mathbf{X}$ be a $p$-dimensional random variable, and let $d$ be a constant such that $d<p$. We say that $\mathbf{X}$ is $d$-marginally recoverable if any $d$-dimensional marginal distribution of $\mathbf{X}$ belongs to the same distribution family, and the parameters of all $d$-dimensional marginal distributions collectively characterize the parameters of the full distribution. We will introduce this general definition in the Discussion section of our revision and explore additional distribution examples that align with it in our future work.
>
> For suggestions:
> >1. Notation $d$ is used for both dimension and Hellinger distance in Condition 4.1 and 4.2.
>
> Thank you for your helpful suggestions. We have corrected the duplicate symbols in the manuscript by replacing $d$ (used for dimension in Conditions 4.1 and 4.2) with $k$.
>
> >2. It is very hard to interpret the sample size requirement in Theorem 4.4. Some discussion of sample complexity depending on the dimension / sparsity will be appreciated.
>
> For some $\eta >2$, the result in Theorem 4.4 holds under the sample size condition $n>O(\eta s^{1/2}\log p)$, where $n$ is the sample size, $s$ denotes the sparsity level of the network (i.e., the number of nonzero entries), and $p$ denotes the dimension. Since $\widehat{\Theta}_g$ recovers all the zeros and nonzeros in $\Theta_g$ with probability $1-p^{2-\eta}$, increasing $\eta$ raises this probability towards 1, but requires a larger sample size.
>
> >3. It would be helpful to provide an example with both continuous and discrete variables with explicit modeling formula.
>
> Here, we provide a specific example involving both continuous and discrete variables, with an explicit modeling formula. Assume that $\mathbf{X}=(\mathbf{X}_1,\mathbf{X}_2)$, where $\mathbf{X}_1$ is a random $a$-vector and $\mathbf{X}_2$ is a random $b$-vector. Suppose that there is a random vector $\mathbf{Z}_1=(Z_1,...,Z_a)$ such that $(\mathbf{Z}_1,\mathbf{X}_2)\sim N(\mu,\Sigma)$ and $X_j=I(Z_j>C_j)$ for all $j=1, ..., a$, where $C_j$ is a constant. We can easily verify that $\mathbf{X}$ is marginally recoverable. While this paper does not primarily focus on mixed-type data, our approach can be applied to parameter estimation as long as the variables meet the marginal recoverability definition.
>
> For questions:
> >1. How do we justify Condition 4.1 and 4.2 intuitively?
>
> Conditions 4.1 and 4.2 establish a relationship between the parameter distance and the distance between the marginal density functions. Intuitively, for any two parameters within the parameter space, if their Euclidean distance is small, the corresponding Hellinger distance between their marginal density functions will also be small. When verifying that specific distributions satisfy Conditions 1 and 2, we found that this is equivalent to verifying the following conditions:
>
> (A1) The Fisher information matrix is positive definite and continuous with respect to the parameters.
>
> (A2) Both the score function and the Hessian matrix are dominated by integrable functions.
>
> (A3) The parameter space is compact, and the model is identifiable.
>
> Among these, (A1)-(A3) are standard regularity conditions. However, expressing the original conditions as (A1)-(A3) would appear somewhat redundant. Conditions 4.1 and 4.2, on the other hand, provide a more intuitive link between the distances of densities and parameters, making the formulation clearer and more accessible. Therefore, we use these boundedness conditions to present the core conditions of our theorem instead of (A1)-(A3).
>
> >2. Is there any theoretical guarantee for the consistency of membership assignment in the mixture modeling?
>
> For the Gaussian mixture model, Chen and Zhang [1] derive minimax lower bounds for clustering with respect to the misclustering error rate. To the best of our knowledge, establishing theoretical guarantees for membership assignment remains a challenging problem. Due to time constraints, we have not addressed the consistency of membership assignment in general mixture models in this paper, but this could be explored in future work.
>
> Reference
>
> [1] Chen, X. and Zhang, A. Y. Achieving optimal clustering in gaussian mixture models with anisotropic covariance structures. In The Thirty-eighth Annual
> Conference on Neural Information Processing Systems, 2024.

---

> > ### Comment · Reviewer_HnXb · 2025-04-03
> >
> > Thank you for the detailed resposne. The response is helpful for my understanding of the paper. I would keep my score unchanged.

---

> > > ### Author Response · Authors · 2025-04-03
> > >
> > > Thank you for your feedback and support. We will add the rebuttal contents to the main paper in the final version following your valuable suggestions.

---

### Official Review · Reviewer_fTvr · 2025-03-21

**Overall Recommendation:** 3

**Summary:**

This article introduces a new class of graphical models, referred to as the marginally recoverable parametric family, which aims to tackle challenges related to efficiency and heterogeneity in high-dimensional network inference tasks. The proposed family is rather flexible, with the joint distribution characterized by pairwise marginal distributions. To facilitate efficient parameter estimation, the paper presents a Maximum Marginal Likelihood Estimator, which is further extended to handle heterogeneous scenarios through mixture modeling. The method’s effectiveness is supported by theoretical consistency, simulations, and applications to real-world sequencing data.

**Claims And Evidence:**

The theoretical results are well-supported, though not particularly surprising, as they follow directly from well-established theorems. The evidence provided is both clear and convincing.
While the marginally recoverable parametric family introduced is intriguing, I remain somewhat skeptical that it encompasses distributions beyond elliptical copulas and simple hierarchical compositions that do not alter the dependency structure.

**Essential References Not Discussed:**

No.

**Experimental Designs Or Analyses:**

Experimental results show that the proposed method performs well on benchmark datasets. The simulation studies are carefully designed, addressing diverse experimental settings and data heterogeneity. The real-world applications are both relevant and well-motivated, with analyses supported by biological literature.

**Methods And Evaluation Criteria:**

On the experimental side, the model's performance is assessed through various simulation benchmarks and large-scale real-world applications. All evaluation procedures appear to be appropriate and well-executed.

**Other Comments Or Suggestions:**

No.

**Other Strengths And Weaknesses:**

The primary contribution of the paper lies in the introduction of the marginally recoverable family. While the family is presented in a general form, it appears to only include elliptical copula distributions (and simple compositions thereof). This raises some concerns about the novelty of the approach, as elliptical copula families are already well-established in the literature. Additionally, elliptical copulas are quite limited in their expressive power, particularly when it comes to modeling tail dependencies, which either vanish asymptotically or are entirely absent in the special case of the Gaussian copula. In many applications of copulas, this limitation has led to the development of more complex copula families. Therefore, the claim on page 2 that "The marginally recoverable parametric family includes many of the most common distributions" might be somewhat misleading.

**Questions For Authors:**

Please comment on my concerns regarding the "real" flexibility of the model beyond elliptical copulas.

**Relation To Broader Scientific Literature:**

The family introduced in this paper is versatile, making the proposed approach applicable to several problems related to graphical models. However, it is also relatively limited in its ability to model tail dependencies, which might be problematic in some applications.

**Theoretical Claims:**

The consistency proofs are mathematically sound and based on widely recognized statistical methods. However, I believe that neither Theorem 4.2 nor Theorem 4.3 should be labeled as "Theorems". Instead, I would suggest calling them "Lemmas," as they directly follow from other established theorems.

---

> ### Author Rebuttal · Authors · 2025-03-31
>
> We sincerely appreciate your time in reviewing our paper and your insightful comments. In the following response, we would like to address your major concern and provide additional clarification.
>
> For theoretical claims:
>
> > I believe that neither Theorem 4.2 nor Theorem 4.3 should be labeled as "Theorems". Instead, I would suggest calling them "Lemmas," as they directly follow from other established theorems.
>
> Thank you for your valuable suggestion. Since these two theorems are derived from established results, we acknowledge that referring to them as "Lemmas" is more appropriate. Nevertheless, we would like to emphasize that these results establish that once a connection between the parameter distance and the density distance is made (as specified in Conditions 4.1 and 4.2), we can obtain tail probability control for high-dimensional parameters. We believe that this provides new insights and a novel proof strategy. Furthermore, verifying that a specific model satisfies the necessary conditions (Conditions 4.1 and 4.2) is nontrivial, especially for complex models such as MPLN. In fact, a substantial portion of the appendix (see Appendix A.3) is dedicated to rigorously demonstrating that the MPLN model satisfies these conditions.
>
> For weaknesses:
>
> > While the family is presented in a general form, it appears to only include elliptical copula distributions (and simple compositions thereof). This raises some concerns about the novelty of the approach, as elliptical copula families are already well-established in the literature. Additionally, elliptical copulas are quite limited in their expressive power, particularly when it comes to modeling tail dependencies, which either vanish asymptotically or are entirely absent in the special case of the Gaussian copula. In many applications of copulas, this limitation has led to the development of more complex copula families. Therefore, the claim on page 2 that "The marginally recoverable parametric family includes many of the most common distributions" might be somewhat misleading.
>
> Thanks for your comment. In fact, the family includes distributions beyond elliptical copula distributions (and simple compositions thereof). For instance, the multinomial distribution belongs to the family but does not fall within the class of elliptical copula distributions. Specifically, let $\mathbf{X}=\left(X_{1}, \ldots, X_{p}\right)^\top \sim Multinomial(n,\boldsymbol{\theta})$, where $\boldsymbol{\theta}=(\theta_1,\dots,\theta_p)^\top$. Then, for $1 \leq j < k \leq p $, we have $X_j \sim Multinomial(n, \theta_{j}) $ and $\left(X_j, X_k\right)^\top \sim Multinomial(n, (\theta_j,\theta_k)^\top ) $. This demonstrates that the multinomial distribution satisfies the definition of the family.
>
> Furthermore, the concept of the marginally recoverable family can be extended to a more general framework. We introduce the definition of a $d$-marginally recoverable family as follows:
>
> Let $\mathbf{X}$ be a $p$-dimensional random variable, and let $d$ be a constant such that $d<p$. We say that $\mathbf{X}$ is $d$-marginally recoverable if any $d$-dimensional marginal distribution of $\mathbf{X}$ belongs to the same distribution family, and the parameters of all $d$-dimensional marginal distributions collectively characterize the parameters of the full distribution.
>
> In this way, the marginally recoverable family becomes more flexible. Under this definition, the multinomial distribution mentioned earlier is actually 1-marginally recoverable, as only the parameters of the 1-dimensional marginal distributions are required to recover all the parameters of the full distribution. We will introduce this general definition in the Discussion section of a future revision and explore additional examples that align with this broader definition in our future work.
>
> For modeling tail dependencies, we acknowledge that elliptical copulas do have limitations in this regard. That said, this concern is primarily relevant in financial applications. Our work, on the other hand, focuses on network inference and its applications to biological genomics data, where the elliptical copula family is commonly used, and tail dependencies are generally not a central issue. Exploring tail dependencies within our framework is an interesting direction for future research. The claim on page 2 could be revised to: "The marginally recoverable parametric family includes distributions commonly used in the field of bioinformatics," which would be a more precise statement. We sincerely appreciate your constructive suggestion.

---

### Official Review · Reviewer_7ySj · 2025-03-25

**Overall Recommendation:** 3

**Summary:**

The paper proposes a new, unified class of graphical models—called the marginally recoverable parametric family—designed to handle diverse data types (e.g., Gaussian, Poisson log‑normal, and latent Gaussian copula models) and heterogeneous structures via mixture modeling. The authors introduce an efficient maximum marginal likelihood estimator (MMLE) that avoids full high‑dimensional integration by leveraging low‑dimensional marginal computations, and extend it to EM‑MMLE for mixture contexts. The work is supported by extensive theory, simulations, and an application to single‑cell RNA‑seq gene regulatory network inference.

**Claims And Evidence:**

### Main Claims:

- The proposed framework unifies and generalizes a range of graphical models while reducing computational complexity.

- MMLE (and its EM extension) achieves consistent parameter estimation and sign recovery with nearly optimal convergence rates.

### Major Concern:

The theoretical guarantees hinge on two stringent conditions (the lower‑boundedness and upper‑boundedness conditions in Conditions 4.1 and 4.2). Their restrictive nature may limit applicability in practical scenarios where the data do not closely conform to these idealized requirements.

**Essential References Not Discussed:**

N/A

**Experimental Designs Or Analyses:**

### Simulation Studies:

Simulations are performed on mixed count and binary data generated from realistic distributions (e.g., MPLN for count data and latent Gaussian copula for binary data).

### Real‑Data Application:

The method is applied to scRNA‑seq data to infer gene regulatory networks, compared against silver standards from public databases.

**Methods And Evaluation Criteria:**

The estimation strategy first maximizes one‑dimensional marginal likelihoods to estimate location and scale parameters, then refines covariance estimation by using two‑dimensional marginal likelihoods. This decomposition reduces the burden of high‑dimensional integration. The extension to mixture models is handled via an EM algorithm updating the MMLE.

Performance is measured using standard network inference metrics (e.g., AUPR, Frobenius norm error) and clustering accuracy (Adjusted Rand Index). The study covers various simulation settings—including different network structures, dimensions, mixing levels, and zero proportions—to assess robustness.

**Other Comments Or Suggestions:**

N/A

**Other Strengths And Weaknesses:**

### Strengths:

- Presents a versatile, unified framework that handles multiple data types and heterogeneous populations.

- Develops an innovative estimation strategy (MMLE/EM‑MMLE) that significantly reduces computational complexity.

- Offers comprehensive theory and extensive simulations alongside a compelling real‑data application in genomics.

### Weaknesses:

- The two core conditions (Conditions 4.1 and 4.2) are quite strong, potentially restricting the method’s applicability.

- More experiments showing robustness under deviations from the assumed conditions would be valuable.

**Questions For Authors:**

See my previous comments.

**Relation To Broader Scientific Literature:**

The work builds on several key strands of graphical model research—extending classical Gaussian approaches to handle count and binary data—and connects with recent work on latent Gaussian copula models.

**Theoretical Claims:**

The paper provides detailed proofs—offered in the supplementary material—for key results such as convergence rates (Theorem 4.3) and sign consistency (Theorem 4.4).

Concern: The reliance on Conditions 4.1 and 4.2 is a potential weakness. These assumptions, pivotal to the derived guarantees, appear rather strong and may not hold in many real‑world applications.

---

> ### Author Rebuttal · Authors · 2025-03-31
>
> We sincerely appreciate your time in reviewing our paper and your insightful comments. In the following response, we would like to address your major concern and provide additional clarification.
>
> For your major concern in the claims and evidence section, as well as the similar concern raised in the theoretical claims and weaknesses sections:
>
> > The theoretical guarantees hinge on two stringent conditions (the lower-boundedness and upper-boundedness conditions in Conditions 4.1 and 4.2). Their restrictive nature may limit applicability in practical scenarios where the data do not closely conform to these idealized requirements.
>
> Thank you for your comment. We appreciate your concern about the potential restrictiveness of Conditions 4.1 and 4.2 and their applicability to real-world data. We would like to clarify that Conditions 4.1 and 4.2 are, in fact, relatively mild. As demonstrated in Appendix A.3.4 and Appendix A.4, both the MPLN distribution and the latent Gaussian copula model for binary data satisfy these conditions.
>
> More specifically, our proof establishes that a distribution satisfies Conditions 4.1 and 4.2 as long as the following assumptions hold:
>
> (A1) The Fisher information matrix is positive definite and continuous with respect to the parameters.
>
> (A2) Both the score function and the Hessian matrix are dominated by integrable functions.
>
> (A3) The parameter space is compact, and the model is identifiable.
>
> Among these, (A1)-(A3) are standard regularity conditions, which are widely used to ensure the asymptotic efficiency of the maximum likelihood estimator, as shown in Shao's book [1].
>
> For the other weakness:
>
> > More experiments showing robustness under deviations from the assumed conditions would be valuable.
>
> Thank you for your comment. Due to rebuttal time constraints and the fact that we have not identified any marginally recoverable distributions that do not satisfy the conditions, we have not conducted additional experiments to demonstrate robustness under deviations from the assumed conditions at this time.
>
> Reference
>
> [1] Shao, J. Mathematical statistics. Springer Science \& Business Media, 2008.

---

### Decision · Program_Chairs · 2025-05-01

**Decision:**

Accept (poster)

**Comment:**

All reviewers assigned (mostly weakly) positive scores. They all agreed that the underlying research question of designing flexible models for handling heterogeneous data is interesting and relevant, that  the proposed framework of the "marginally recoverable family" of copula distributions has contains some novel inference ideas, and that the experiments conducted are rather convincing. On the more negative side, reviewers mentioned potential limitations of the marginally recoverable family concerning the restriction to Gaussian-like distributions and problems with handling tail dependencies in data, and some missing experiments that would  indicate how robust the proposed framework is when some of the theoretical assumptions are violated. Some of these concerns, however,  could be addressed (at least to some degree) in the rebuttal. In summary, I think that for this paper, the positive aspects outweigh the negative ones.